# Batched Energy-Entropy acquisition for Bayesian Optimization

**Felix Teufel**[12]    **Carsten Stahlhut**[1]    **Jesper Ferkinghoff-Borg**[1]

[1]Machine Intelligence, Novo Nordisk A/S    [2]Department of Biology, University of Copenhagen

`{fegt,ctqs,jfgb}@novonordisk.com`

## Abstract

Bayesian optimization (BO) is an attractive machine learning framework for performing sample-efficient global optimization of black-box functions. The optimization process is guided by an acquisition function that selects points to acquire in each round of BO. In batched BO, when multiple points are acquired in parallel, commonly used acquisition functions are often high-dimensional and intractable, leading to the use of sampling-based alternatives. We propose a statistical physics inspired acquisition function for BO with Gaussian processes that can natively handle batches. Batched Energy-Entropy acquisition for BO (BEEBO) enables tight control of the explore-exploit trade-off of the optimization process and generalizes to heteroskedastic black-box problems. We demonstrate the applicability of BEEBO on a range of problems, showing competitive performance to existing methods.

## 1 Introduction

Bayesian Optimization (BO) has since its inception [1, 2] made a profound contribution to the realm of global optimization of black-box functions through the usage of Bayesian statistics. For global optimization problems pursuing $x_* = \text{argmax}_{x \in \mathcal{X}} f_{\text{true}}(x)$, BO has surfaced as a premier strategy for efficiently handling especially complex and costly unknown functions, $f_{\text{true}}(x)$. While BO is traditionally formulated in a single-point scenario, where individual points are queried and results are observed sequentially, there are situations where *batched* acquisition is needed. Such situations arise when $f_{\text{true}}(x)$ is expensive to evaluate in either time or cost, but can be effectively evaluated in parallel by dispatching multiple experiments, reducing the overall optimization time. This is often the case in e.g. drug discovery, materials design or hyper-parameter tuning for deep models [3, 4, 5, 6, 7].

The realization that BO could be employed for the training of deep neural networks, as suggested by [3], sparked renewed research interest, with advancements encompassing a variety of areas, including the generalization to accommo-

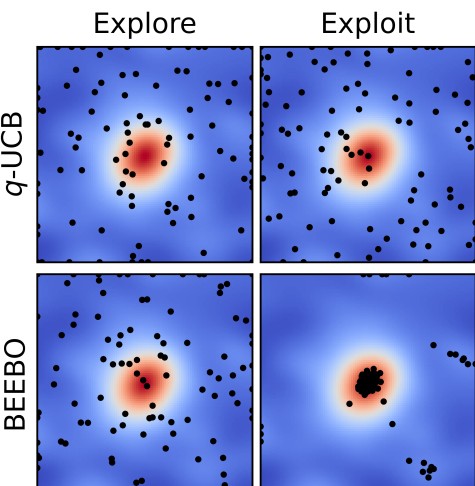

Figure 1: $q$-UCB does not allow for controlling its explore-exploit trade-off with large batches. A GP surrogate (background) was initialized with 100 random points of the Ackley function. $q$-UCB was run with $\kappa = 0.1$ and $\kappa = 100$, BEEBO with $T' = 0.05$ and $T' = 50$. Batch size $Q = 100$.

38th Conference on Neural Information Processing Systems (NeurIPS 2024).

date noisy inputs [8, 9], heteroskedastic noise [10, 11], multi-task problems [12], multi-fidelity [13], high-dimensional input spaces [14], and parallel methods with batch queries [15, 16]. Generally, these desired properties are addressed by customizing one of the two key components in BO, either the *surrogate model* or the *acquisition function*. The surrogate model $f$ approximates the black-box function $f_{\text{true}}$ using the available data. In BO, the surrogate is formulated from a Bayesian perspective, allowing us to quantify the model's uncertainty when evaluating new points. Typically, the model of choice is a Gaussian Process (GP) [17]. The acquisition function is responsible for guiding the selection of new input point(s) to evaluate at each optimization step, utilizing the surrogate model to identify promising regions in the input domain and exploring the unknown function further.

Any acquisition process needs to trade off exploration (reducing uncertainty to learn a better surrogate model) against exploitation (selecting points with a high expected $f_{\text{true}}(x)$ based on the current surrogate). In this work, we are particularly interested in acquisition processes that make this trade-off controllable using a hyperparameter. Controllability can be a desirable property if e.g. domain knowledge relating to the difficulty of the optimization process and the quality of the surrogate model is available, or if the strategy needs to be adjusted depending on future experimental budgets. Similarly, it can be desirable to acquire multiple $x$ with high $f_{\text{true}}(x)$ in a batch (as opposed to just finding the optimum $x_*$, with the remaining $x$ being considered explorative). This is useful when optima identified in BO can be subject to constraints that are unknown at optimization time, but may render $x_*$ intractable [18]. Such constraints arise when the $f_{\text{true}}$ explored in BO is a necessary simplification of the actual objective. Practical examples include e.g. the synthesizability of a material at larger scale, when BO experiments are performed at lab scale; or the *in vivo* activity of a molecule with BO experiments performed *in vitro*.

A wide range of batch mode acquisition functions has been proposed, with approaches often leveraging random sampling strategies or Monte Carlo (MC) integration, which can adversely affect controllability for large batches (Figure 1). In contrast, we here introduce BEEBO (Batched Energy-Entropy acquisition for BO), a statistical physics inspired acquisition function for BO with GP surrogate models that natively generalizes to batched acquisition. BEEBO enables

- Parallel gradient-based optimization of the inputs, **without requiring sampling or Monte Carlo integrals**.
- **Tight control of the explore-exploit trade-off** in batch mode using a single temperature hyperparameter.
- Risk-averse BO under **heteroskedastic noise**.

We demonstrate the application of BEEBO on a wide range of test problems, and investigate its behaviour under heteroskedastic noise.

## 2  Related works

**Batch variants of traditional strategies**  Parallel acquisition in BO has seen a variety of approaches, often starting from established single-point acquisition functions like probability of improvement (PI), expected improvement (EI), knowledge gradient (KG) or upper confidence bound (UCB) [2, 19, 20, 21, 22]. Reformulating these to batch mode with $Q$ query points, we obtain $q$-PI, $q$-EI, and $q$-UCB [23, 24]. While the single-point specifications provide an analytical form and enable gradient-based optimization, batch expressions are more challenging and require different optimization strategies, typically involving greedy algorithms [25] or deriving an integral expression over multiple points.

For instance, in the popular EI acquisition function, a single point is selected by maximizing the expression $a_{\text{EI}}(x) = \mathbb{E}[\max(0, f(x) - f_t^*)] = \int \max(0, f(x) - f_t^*) P(f|x) df$. Here $f_t^*$ represents the best observed evaluation of $f_{\text{true}}$ so far. With a surrogate model in the form of a GP, the acquisition function depends only on the predictive mean and variance functions, $\mu(x)$ and $C(x)$. Effectively, we need to evaluate the cumulative normal distribution, which quickly becomes intractable for large batch sizes and approximating the gradient of the $q$-EI acquisition function typically requires MC estimation [26, 27]. However, proper MC integration can be laborious and is sensitive to both the dimension of the problem and the choice of batch size $Q$. Specifically, MC methods face the *curse of dimensionality* problem when applied to high-dimensional integrals, as they require an exponentially increasing number of sample points to maintain accuracy, making them

computationally impractical for such tasks [28, 29]. Of particular interest is Wilson et al. [24], in which they adopt the reparameterization trick [30, 31] on acquisition functions integrals, enabling gradient based approaches to the optimization of PI, EI, and UCB. This demonstrates particular usefulness in modest to higher dimensions.

While EI trades off exploration and exploitation, users do not have a direct control over the balance. To alleviate this, Sobester et al. [32] proposed a weighted EI formulation. An alternative strategy with an explicit explore-exploit trade-off is offered by the UCB acquisition function, $a_{\text{UCB}}(x) = \mu(x) + \sqrt{\kappa} \cdot \sqrt{C(x)}$, which directly expresses exploration and exploitation as two terms, traded off by the parameter $\kappa$. As we are particularly interested in enabling this direct user control, we focus our primary comparison on $q$-UCB in the main text, while a more extensive comparison with alternative methods can be found in Appendix B, both theoretically and experimentally.

**Greedy strategies**   As mentioned, a popular approach for leveraging single-point acquisition functions is devising batch filling strategies that score candidate points sequentially. Kriging Believer (KB) [33] uses EI to select points and iteratively updates the GP by fantasizing an observation with the posterior mean. Likewise, GP-BUCB [34] uses fantasized observations to update $\sqrt{C(x)}$ at each step. Local penalization (LP) [35] introduces a penalization function that repulses selection away from already selected points. Contal et al. [36] propose selecting a single point using UCB and dedicating the remainder of the batch budget for exploration in a restricted region around the believed optimum. GLASSES [37] treats batch selection as a multi-step lookahead problem to overcome the myopia of only considering the immediate effect of selecting a point.

**Entropy based strategies**   From an information theory perspective, BO can be interpreted as seeking to reduce uncertainty over the location of optima of the unknown function. This has given rise to entropy-based acquisition functions such as entropy search (ES) [38], predictive entropy search (PES) [39] and max-value entropy search (MES) [13, 40, 41]. MES is distinct in that it seeks to quantify the mutual information between the unknown $f_{\text{true}}(x_*)$ and the observations $y|D$, rather than the location of $x_*$. General-purpose Information-Based Bayesian OptimizatioN (GIBBON) [42] provides an extension of MES that enables application to batched acquisition as well as other challenges such as multi-fidelity BO. GIBBON proposes a lower bound formulation for the intractable batch MES criterion, which is then optimized using greedy selection. Despite being formulated to handle a large degree of parallelism, Moss et al. [42] reported that GIBBON fails in practice for large batches with $Q > 50$. Potentially, this behaviour is a consequence of the accuracy of the lower bound approximation. A heuristic scaling of the batch diversity was proposed to improve performance with large batches. GIBBON may also be interpreted as a determinantal point process (DPP) [4, 43]. In Appendix B we provide a detailed discussion of the relationship of the BEEBO acquisition function to GIBBON and DPPs. Note that while we will also make use of the term *entropy* in BEEBO, the quantity is distinct from the ones leveraged by the aforementioned approaches in the sense that it does not relate to an unknown optimum.

**Thompson sampling**   Given the challenges of generalizing acquisition functions to batch mode, Thompson sampling (TS), which was originally adopted from bandit problems [44, 45, 46, 47, 48], is a popular alternative strategy for guiding batched BO. While being an attractive approach in general, it has been demonstrated that default TS can become too exploitative, motivating the use of alternatives such as Bayesian Quadrature [49], or advanced strategies on top of TS that ensure diversity [18]. Eriksson et al. [50] demonstrate that overexploration also can be problematic in higher dimensions, and alleviate this using local trust regions in TuRBO. Maintaining such regions with high precision discretization can be memory-expensive, as indicated by [51], who suggest using MCMC-BO with adaptive local optimization to address this by transitioning a set of candidate points towards more promising positions.

## 3   The BEEBO acquisition function

Assume $f_{\text{true}} : X \rightarrow \mathbb{R}$ is some real output associated with the input and a set of data be given $D = \{(x_i, y_i)\}_{i=1}^N$ where $y_i \in \mathbb{R}$ represent some noisy observations of $f_{\text{true}}(x_i)$, say

$$y_i = f_{\text{true}}(x_i) + \epsilon_i \tag{1}$$

with $\epsilon_i$ denoting the measurement noise. Let $\mathbf{x} = (x_1, \cdots, x_Q) \in X^Q$ represent a collection of test points we wish to assign an acquisition value to. In keeping with the BO framework, we assume a given posterior probability distribution over the surrogate function $\mathbf{f}$ evaluated at $\mathbf{x}$,

$$\mathbf{f}(\mathbf{x}) \sim P(\mathbf{f} \,|\, D, \mathbf{x}) \tag{2}$$

The lack of knowledge we have of the surrogate function at $\mathbf{x}$ is quantified by the differential *entropy* $H$:

$$H\big(\mathbf{f} \,|\, D, \mathbf{x}\big) = - \int P\big(\mathbf{f} \,|\, D, \mathbf{x}\big) \ln\Big(P\big(\mathbf{f} \,|\, D, \mathbf{x}\big)\Big) d\mathbf{f} \tag{3}$$

This entropy can be contrasted with the expected entropy of the surrogate function if $Q$ observations $\mathbf{y} = (y_1, \cdots, y_Q)$ were acquired at $\mathbf{x}$, i.e. if the training data $D$ would be augmented with $D'(\mathbf{y}) = \{(x_q, y_q)\}_{q=1}^Q$ to form the joint data set $D_{\text{aug}}(\mathbf{y}) = \big(D, D'(\mathbf{y})\big)$. We refer to this entropy as $H_{\text{aug}}$:

$$H_{\text{aug}}\big(\mathbf{f} \,|\, D, \mathbf{x}\big) = \int P(\mathbf{y} \,|\, D, \mathbf{x}) H\big(\mathbf{f} \,|\, D_{\text{aug}}(\mathbf{y})\big) d\mathbf{y}, \tag{4}$$

where $P(\mathbf{y} \,|\, D, \mathbf{x})$ represent the posterior predictive distribution at $\mathbf{x}$. The expected *information gain*, $I(\mathbf{x})$, from acquiring observations at $\mathbf{x}$ is given by the expected reduction of entropy from this process:

$$I(\mathbf{x}) = H\big(\mathbf{f} \,|\, D, \mathbf{x}\big) - H_{\text{aug}}\big(\mathbf{f} \,|\, D, \mathbf{x}\big) \tag{5}$$

We propose to represent the explore component of the acquisition function, $a_{\text{BEEBO}}$, by $I(\mathbf{x})$. The information gain $I(\mathbf{x})$ is distinct from the quantities exploited by entropy search approaches, as it quantifies global uncertainty reduction, rather than estimating the information over an unknown $x_*$. The information gain is directly applicable to multivariate functions and to heteroskedastic settings where $\sigma^2 = \sigma^2(\mathbf{x})$. Since large measurement uncertainties imply smaller information gain, $a_{\text{BEEBO}}$ exhibits risk-averse behaviour [11] by automatically prioritizing regions of small uncertainties from where more precise information of $f_{\text{true}}$ can be obtained, everything else being equal.

The exploit component of BEEBO relies on taking expectation values of a scalar function of the random variable $\mathbf{f}(\mathbf{x})$, $\tilde{E} : \mathbb{R}^Q \to \mathbb{R}$, that summarizes the optimality properties of a given batch $\mathbf{x}$. Natural choices would be the mean or the maximum of $\mathbf{f}(\mathbf{x})$. Of particular interest is expressing the optimality as a softmax-weighted sum over $\mathbf{f}(\mathbf{x})$, as this allows us to smoothly interpolate between the two regimes:

$$E(\mathbf{x}) = -\mathbb{E}[\tilde{E}(\mathbf{x})] \cdot Q = -\mathbb{E}\left[\sum_{q=1}^{Q} \text{softmax}(\beta\mathbf{f})_q f_q\right] \cdot Q, \tag{6}$$

where $\beta$ is the softmax inverse temperature. At $\beta = 0$, we recover the mean. We scale the expectation with $Q$ so that both $I$ and $E$ scale linearly with increasing batch size. While the mean provides a closed form expression for its expectation, this is not the case for the general softmax-weighted sum of a multivariate normal. Using Taylor expansion, we introduce an approximation of the expectation of the softmax-weighted sum that is fully differentiable and can be computed in closed form. A detailed derivation is provided in Appendix A. At $\beta = 0$, all $Q$ points contribute equally to $E(\mathbf{x})$, whereas at $\beta > 0$, points that do not compete for optimality are dynamically *released*. This effect can be quantified as the *effective number of points* via the entropy of the softmax weights. In the following, we will refer to the (exact) $\beta = 0$ limit as meanBEEBO, and the (approximated) general case as maxBEEBO.

The BEEBO acquisition function then takes the form

$$a_{\text{BEEBO}}(\mathbf{x}) = -E(\mathbf{x}) + T \cdot I(\mathbf{x}), \tag{7}$$

where $T$ sets the balance between exploitation (small $T$) and exploration (large $T$). As both $E$ and $I$ scale with the batch size $Q$, a given choice of $T$ would set the explore-exploit balance in an approximately $Q$-independent manner. This acquisition function bears a strong similarity to the definition of (negative) *free energies* in statistical physics, where $E$ and $I$ correspond to respectively the thermodynamic energy and entropy of the system and $T$ corresponds to the temperature.

## 3.1 BEEBO with Gaussian processes

Gaussian processes offer a particular convenient framework for BO, due to the availability of closed-form expressions for the inference step [17]. Specifically

$$
\begin{aligned}
P(\mathbf{f} \mid D, \mathbf{x}) &= \mathcal{N}(\mathbf{f} \mid \mu(\mathbf{x}), C(\mathbf{x})) \\
\mu(\mathbf{x}) &= K(\mathbf{x}, \mathbf{x}_D) \cdot M_D^{-1} \cdot \mathbf{y}_D \\
C(\mathbf{x}) &= K(\mathbf{x}, \mathbf{x}) - K(\mathbf{x}, \mathbf{x}_D) \cdot M_D^{-1} \cdot K(\mathbf{x}_D, \mathbf{x}) \\
M_D &= K(\mathbf{x}_D, \mathbf{x}_D) + \sigma^2(\mathbf{x}_D)
\end{aligned} \tag{8}
$$

where $\mathcal{N}(\cdot \mid \mu, C)$ is the multivariate Gaussian distribution with mean $\mu$, and covariance $C$, $\mathbf{x}_D$ and $\mathbf{y}_D$ are the $x$ and $y$ values of the acquired data, $\sigma(\mathbf{x}_D) = \text{diag}\left(\sigma_1^2, \cdots, \sigma_N^2\right)$ is a diagonal matrix with the measurement uncertainties in the diagonal and $K(\cdot, \cdot)$ are matrices derived from the GP-kernel, $k(\cdot, \cdot)$, i.e. $K(\mathbf{x}, \mathbf{x}')_{ij} = k(x_i, x'_j)$. It is worth noting that $C(\mathbf{x})$ only depends on the input location of the test points $\mathbf{x}$ and the data points $\mathbf{x}_D$ with their corresponding measurement uncertainties, $\sigma^2(\mathbf{x}_D)$, but *not* on the actual observations, $\mathbf{y}_D$. Consequently, the entropy of the posterior distribution

$$
H\big(\mathbf{f} \mid D, \mathbf{x}\big) = \frac{Q}{2} \ln(2\pi e) + \frac{1}{2} \ln \det(C(\mathbf{x})) \tag{9}
$$

is independent of $\mathbf{y}_D$ as well, with $\ln \det$ denoting the log determinant. Similarly, the expected entropy of $\mathbf{f}$ if observations at $\mathbf{x}$ were acquired, simply reads

$$
H_{\text{aug}}\big(\mathbf{f} \mid D, \mathbf{x}\big) = \frac{Q}{2} \ln(2\pi e) + \frac{1}{2} \ln \det(C_{\text{aug}}(\mathbf{x})), \tag{10}
$$

where

$$
\begin{aligned}
C_{\text{aug}}(\mathbf{x}) &= K(\mathbf{x}, \mathbf{x}) - K(\mathbf{x}, \mathbf{x}_{\text{aug}}) \cdot M_{\text{aug}}^{-1} \cdot K(\mathbf{x}_{\text{aug}}, \mathbf{x}) \\
M_{\text{aug}} &= K(\mathbf{x}_{\text{aug}}, \mathbf{x}_{\text{aug}}) + \sigma^2(\mathbf{x}_{\text{aug}})
\end{aligned} \tag{11}
$$

and $\mathbf{x}_{\text{aug}} = \mathbf{x}_{D_{\text{aug}}}$. The BEEBO acquisition function is then given by

$$
a_{\text{BEEBO}}(\mathbf{x}) = \mathbb{E}[\tilde{E}(\mathbf{x})] \cdot Q + T \cdot I(\mathbf{x}) \tag{12}
$$

where the expectation is either the mean, $\frac{1}{Q} \sum_{q=1}^{Q} \mu_q$, or the closed form approximation of the softmax-weighted sum described in Appendix A, and

$$
I(\mathbf{x}) = \frac{1}{2} \ln \det(C(\mathbf{x})) - \frac{1}{2} \ln \det(C_{\text{aug}}(\mathbf{x})). \tag{13}
$$

All operations needed to compute the acquisition value $a_{\text{BEEBO}}(\mathbf{x})$ are analytical. Using automatic differentiation, the batch of points $\mathbf{x}$ can therefore be optimized with gradient-based methods, as laid out for meanBEEBO in Algorithm 1, with learning rate $\gamma$. In the pseudocode, $\mathcal{GP}$ denotes a trained GP model object that holds the training data and the kernel function. Using the kernel's learned amplitude $A$, we can relate BEEBO's $T$ parameter to the $\kappa$ of UCB. This allows us to configure BEEBO using a scaled temperature $T'$ that ensures both methods have equal gradients at isosurfaces, enabling the user to follow existing guidance and intuition from UCB to control the trade-off. A derivation is provided in Appendix B.1.

---

**Algorithm 1:** meanBEEBO optimization

**Input:** model $\mathcal{GP}$, initial batch points $\mathbf{x}$, temperature $T$

**repeat**

    Calculate $\mu(\mathbf{x}), C(\mathbf{x})$ from Equation 8 using $\mathcal{GP}$

    $E \leftarrow -\sum_{q=1}^{Q} \mu_q$

    $\mathcal{GP}_{\text{aug}} \leftarrow \text{fantasize}(\mathcal{GP}, \mathbf{x})$

    Calculate $C_{\text{aug}}(\mathbf{x})$ from Equation 11 using $\mathcal{GP}_{\text{aug}}$

    $I \leftarrow \frac{1}{2} \ln \det (C(\mathbf{x})) - \frac{1}{2} \ln \det (C_{\text{aug}}(\mathbf{x}))$

    $a \leftarrow -E + T * I$

    $\mathbf{x} \leftarrow \mathbf{x} + \gamma \nabla a$

**until** converged

**Output:** optimized batch points $\mathbf{x}$

---

# 4 Experiments

**Test problems** We benchmark acquisition function performance on a range of maximization test problems with varying dimensions (Table 1) available in BoTorch [52]. Test problems that are evaluated on multiple dimensions support specifying the respective arbitrary $d$. As a high-dimensional problem with low inherent dimensionality, we embed the six-dimensional Hartmann function in $d = 100$ [50, 53, 54]. We additionally test on two robot control problems (robot arm pushing and rover trajectory planning) in Appendix D.3 [55, 56].

Table 1: Overview of the test problems used in the experiments.

| Function | Dimension |
|---|---|
| Ackley | 2, 10, 20, 50, 100 |
| Shekel | 4 |
| Hartmann | 6 |
| Cosine | 8 |
| Rastrigin | 2, 10, 20, 50, 100 |
| Rosenbrock | 2, 10, 20, 50, 100 |
| Styblinski-Tang | 2, 10, 20, 50, 100 |
| Powell | 10, 20, 50, 100 |
| Embedded Hartmann 6 | 100 |

On each test problem, we perform 10 rounds of BO using $q$-UCB or BEEBO with a given explore-exploit parameter for direct comparison. We use the scaled temperature $T'$ (B.1) to ensure that both methods operate at the same trade-off. In round 0, we seed the surrogate GP with $Q$ random points that were drawn so that each point has a minimum distance of 0.5 to the test problem's true optimum. We perform ten replicate runs for each problem and method, with replicate seeds controlled so that all methods start from the same $Q$ random points in a replicate. As we evaluate performance in a fixed-round, fixed-$Q$ optimization scenario, we set the explore hyperparameter to 0 in the last round (for maxBEEBO, we also set the softmax $\beta$ to 0). We use $Q = 100$ for all experiments, which is commonly understood to be a large batch size [50]. Additional results on small batch sizes (5, 10) are provided in Appendix D.2. All experiments use BoTorch's default utilities for acquisition function optimization and GPyTorch [57] GP training (C.1).

**Heteroskedastic noise** We investigate performance when optimizing under heteroskedastic noise on the 2D Branin function with three global optima. To construct a heteroskedastic problem, we specify noise so that the noise level is maximal at optima 2 and 3, decaying exponentially with distance to any of the two noised optima (C.4). No noise maximum is added at optimum 1. Therefore, while all three optima share the same $f_{\text{true}}(x)$ (Figure A1), only optimum 1 is favorable in terms of heteroskedastic risk. We perform BO for ten rounds with $\beta = 0.1$ and $Q = 10$ using a heteroskedastic GP that learns surrogate models for both $f_{\text{true}}(x)$ and $\sigma^2(x)$. We report results over five replicate runs.

**Metrics** We report the mean best observed objective value after 10 rounds over the five replicates. As test problems have highly varying scales, we normalize the results on each test problem using min-max normalization. Typically, the minimum of a maximization problem is not known explicitly. We therefore set the minimum for normalization to the highest value observed among the random seed points. The maximum is given by the $f_{\text{true}}(x^*)$ of the problem. The metric thus directly quantifies how much progress has been made to the true optimum from the random starting configuration on a 0-1 scale.

As we are not only interested in identifying a single $x$ with good $f_{\text{true}}(x)$, we additionally quantify the overall quality of the final (exploitative) batch. We compute the batch instantaneous regret $R = \sum_{q<Q} f_{\text{true}}(x^*) - f_{\text{true}}(x_q)$ of the last, exploitative batch. To bring results on all test problems to a similar scale, we divide it by the batch instantaneous regret of a batch of $Q$ random points on each problem. We refer to this metric as the relative batch instantaneous regret, $R_{\text{rel}} = R_{t=10}/R_{\text{random}}$.

For BO under heteroskedastic noise, we wish to quantify the preference of a given method for different optima. As optima share the same $f_{\text{true}}(x^*)$, metrics operating on $f_{\text{true}}(x)$ are inherently unsuitable, and preference needs to be evaluated on $x$ directly. For each acquired point $x_i$, we compute the distances $\|x_i - x_j^*\|_2$ to the $J$ individual optima. We report the mean distance to each optimum over all points in a batch.

Table 2: Highest observed value after 10 rounds of BO with $Q = 100$. The best value at each $\kappa$ is indicated in blue. BEEBO is configured with $T' = 1/2\sqrt{\kappa}$. Full BO curves are provided in D.6, confidence intervals and statistical tests in Tables A2, A3 and A4

| Problem | d | $\sqrt{\kappa} = 0.1$ | | | $\sqrt{\kappa} = 1.0$ | | | $\sqrt{\kappa} = 10.0$ | | |
|---|---|---|---|---|---|---|---|---|---|---|
| | | meanBEEBO | maxBEEBO | $q$-UCB | meanBEEBO | maxBEEBO | $q$-UCB | meanBEEBO | maxBEEBO | $q$-UCB |
| Ackley | 2 | 0.993 | 0.982 | 0.973 | 0.985 | 0.980 | 0.967 | 0.975 | 0.988 | 0.988 |
| Levy | 2 | 1.000 | 1.000 | 1.000 | 0.999 | 1.000 | 1.000 | 0.999 | 0.998 | 0.998 |
| Rastrigin | 2 | 0.981 | 0.993 | 0.951 | 0.989 | 0.983 | 0.983 | 0.983 | 0.993 | 0.933 |
| Rosenbrock | 2 | 0.976 | 0.982 | 0.949 | 0.956 | 0.979 | 0.943 | 0.955 | 0.938 | 0.962 |
| Styblinski-Tang | 2 | 0.961 | 1.000 | 1.000 | 1.000 | 1.000 | 1.000 | 1.000 | 1.000 | 0.999 |
| Shekel | 4 | 0.540 | 0.300 | 0.244 | 0.915 | 0.378 | 0.330 | 0.698 | 0.411 | 0.264 |
| Hartmann | 6 | 1.000 | 0.894 | 0.918 | 1.000 | 0.976 | 0.950 | 0.986 | 0.974 | 0.889 |
| Cosine | 8 | 1.000 | 0.999 | 0.934 | 0.999 | 0.972 | 0.924 | 0.619 | 0.895 | 0.621 |
| Ackley | 10 | 0.915 | 0.819 | 0.800 | 0.908 | 0.736 | 0.772 | 0.822 | 0.546 | 0.513 |
| Levy | 10 | 0.989 | 0.966 | 0.904 | 0.966 | 0.953 | 0.904 | 0.966 | 0.914 | 0.560 |
| Powell | 10 | 0.987 | 0.951 | 0.920 | 0.970 | 0.949 | 0.916 | 0.861 | 0.909 | 0.283 |
| Rastrigin | 10 | 0.463 | 0.558 | 0.420 | 0.536 | 0.573 | 0.522 | 0.595 | 0.590 | 0.311 |
| Rosenbrock | 10 | 0.994 | 0.991 | 0.966 | 0.991 | 0.986 | 0.971 | 0.904 | 0.975 | 0.645 |
| Styblinski-Tang | 10 | 0.837 | 0.822 | 0.309 | 0.835 | 0.638 | 0.492 | 0.289 | 0.229 | 0.049 |
| Ackley | 20 | 0.827 | 0.818 | 0.741 | 0.851 | 0.781 | 0.753 | 0.777 | 0.404 | 0.474 |
| Levy | 20 | 0.949 | 0.945 | 0.926 | 0.943 | 0.904 | 0.900 | 0.889 | 0.907 | 0.819 |
| Powell | 20 | 0.955 | 0.939 | 0.948 | 0.965 | 0.913 | 0.913 | 0.872 | 0.915 | 0.845 |
| Rastrigin | 20 | 0.399 | 0.484 | 0.423 | 0.473 | 0.472 | 0.480 | 0.508 | 0.522 | 0.401 |
| Rosenbrock | 20 | 0.993 | 0.992 | 0.973 | 0.995 | 0.983 | 0.982 | 0.907 | 0.933 | 0.924 |
| Styblinski-Tang | 20 | 0.737 | 0.667 | 0.203 | 0.689 | 0.394 | 0.561 | 0.330 | 0.274 | 0.034 |
| Ackley | 50 | 0.235 | 0.623 | 0.638 | 0.342 | 0.594 | 0.759 | 0.823 | 0.465 | 0.730 |
| Levy | 50 | 0.940 | 0.971 | 0.948 | 0.965 | 0.958 | 0.951 | 0.943 | 0.879 | 0.941 |
| Powell | 50 | 0.954 | 0.975 | 0.970 | 0.982 | 0.969 | 0.961 | 0.950 | 0.938 | 0.980 |
| Rastrigin | 50 | 0.322 | 0.472 | 0.431 | 0.476 | 0.470 | 0.397 | 0.432 | 0.439 | 0.481 |
| Rosenbrock | 50 | 0.971 | 0.976 | 0.968 | 0.984 | 0.981 | 0.986 | 0.983 | 0.962 | 0.981 |
| Styblinski-Tang | 50 | 0.584 | 0.509 | 0.312 | 0.675 | 0.342 | 0.694 | 0.393 | 0.325 | 0.356 |
| Ackley | 100 | 0.277 | 0.417 | 0.708 | 0.190 | 0.540 | 0.645 | 0.863 | 0.682 | 0.844 |
| Emb. Hartmann 6 | 100 | 0.951 | 0.957 | 0.896 | 0.987 | 0.936 | 0.913 | 0.928 | 0.907 | 0.916 |
| Levy | 100 | 0.837 | 0.966 | 0.950 | 0.961 | 0.950 | 0.934 | 0.944 | 0.940 | 0.964 |
| Powell | 100 | 0.810 | 0.985 | 0.982 | 0.952 | 0.980 | 0.981 | 0.983 | 0.979 | 0.984 |
| Rastrigin | 100 | 0.497 | 0.441 | 0.446 | 0.401 | 0.455 | 0.442 | 0.459 | 0.455 | 0.443 |
| Rosenbrock | 100 | 0.822 | 0.953 | 0.972 | 0.971 | 0.976 | 0.969 | 0.980 | 0.970 | 0.978 |
| Styblinski-Tang | 100 | 0.537 | 0.423 | 0.308 | 0.474 | 0.353 | 0.532 | 0.401 | 0.296 | 0.278 |
| **Mean** | | 0.795 | 0.811 | 0.758 | 0.828 | 0.790 | 0.801 | 0.788 | 0.744 | 0.678 |
| **Median** | | 0.940 | 0.951 | 0.920 | 0.961 | 0.949 | 0.913 | 0.889 | 0.907 | 0.819 |

# 5 Results

## 5.1 BO on test problems

We benchmark BEEBO against $q$-UCB at three rates of $\kappa$. Overall, we find that the simpler mean-BEEBO variant outperforms maxBEEBO in terms of mean performance on all but the lowest rate of $\kappa$ (Table 2). As we consider the configuration with the lowest rate to be exploit-dominated, this can be understood as a consequence of maxBEEBO effectively releasing non-contributing points for further exploration, with the low explore rate seeming sufficient to induce the necessary diversity.

While results on individual test problems vary, meanBEEBO shows improved performance over $q$-UCB especially in the medium dimension range up to 50. For $d$=100, we find mixed performance, with meanBEEBO gradually becoming more competitive with increasing $\kappa$. This is due to the fact that with increasing dimensionality, more exploration is beneficial for learning a good surrogate model before an actual BO process becomes effective. As we find that $q$-UCB inherently performs more random-like sampling at large $Q$, irrespective of $\kappa$, it benefits in such situations.

On average over all 33 performed experiments, meanBEEBO improves upon $q$-UCB for large batches at any of the three rates (Table A3). We additionally benchmarked BEEBO against other popular BO strategies without explore-exploit hyperparameters. Interestingly, we found that the Kriging Believer (KB) iterative heuristic [33] can perform very competitively for large batches when using LogEI [58] as the acquisition function (Table A1), especially on the two robot control problems (Appendix D.3), but can be slower to optimize than BEEBO (Table A9).

When evaluating $R_{\text{rel}}$ in the ultimate round of BO, we find that BEEBO allows us to effectively acquire a batch with high $f_{\text{true}}(x)$, highlighting the controllability of the acquisition function (Table 3). The $R_{\text{rel}}$ of $q$-UCB is only slightly better than the $R$ of a random batch in many cases, even though the explore component was explicitly set to 0. We note that this is not due to the surrogate function being unsuitable - the results in Table 2 indicate that in most cases the location of $f_{\text{true}}(x^*)$ is approximately

Table 3: Relative batch instantaneous regret $R_{rel}$ in round 10 ($\kappa = 0$) with $Q = 100$. The best value at each $\kappa$ is indicated in blue. BEEBO is configured with $T' = 1/2\sqrt{\kappa}$. Lower means better.

| Problem | d | $\sqrt{\kappa} = 0.1$ | | | $\sqrt{\kappa} = 1.0$ | | | $\sqrt{\kappa} = 10.0$ | | |
|---|---|---|---|---|---|---|---|---|---|---|
| | | meanBEEBO | maxBEEBO | $q$-UCB | meanBEEBO | maxBEEBO | $q$-UCB | meanBEEBO | maxBEEBO | $q$-UCB |
| Ackley | 2 | 0.292 | 0.259 | 1.006 | 0.268 | 0.245 | 0.999 | 0.257 | 0.165 | 1.002 |
| Levy | 2 | 0.134 | 0.114 | 1.236 | 0.092 | 0.102 | 1.046 | 0.102 | 0.111 | 1.114 |
| Rastrigin | 2 | 0.455 | 0.578 | 1.010 | 0.425 | 0.454 | 0.999 | 0.407 | 0.500 | 1.020 |
| Rosenbrock | 2 | 0.001 | 0.004 | 0.992 | 0.001 | 0.004 | 1.094 | 0.002 | 0.002 | 1.014 |
| Styblinski-Tang | 2 | 0.168 | 0.172 | 1.024 | 0.169 | 0.170 | 1.027 | 0.170 | 0.170 | 1.051 |
| Shekel | 4 | 0.810 | 0.776 | 0.993 | 0.688 | 0.730 | 0.995 | 0.688 | 0.695 | 0.988 |
| Hartmann | 6 | 0.060 | 0.229 | 0.968 | 0.078 | 0.086 | 0.971 | 0.100 | 0.098 | 0.862 |
| Cosine | 8 | 0.045 | 0.001 | 0.953 | 0.001 | 0.016 | 0.975 | 0.222 | 0.061 | 0.922 |
| Ackley | 10 | 0.478 | 0.338 | 0.931 | 0.314 | 0.345 | 0.943 | 0.253 | 0.452 | 0.950 |
| Levy | 10 | 0.041 | 0.030 | 1.188 | 0.023 | 0.048 | 1.011 | 0.261 | 0.103 | 1.111 |
| Powell | 10 | 0.016 | 0.027 | 1.037 | 0.009 | 0.067 | 1.101 | 0.067 | 0.151 | 1.215 |
| Rastrigin | 10 | 0.629 | 0.563 | 0.920 | 0.523 | 0.541 | 0.907 | 0.567 | 0.402 | 0.905 |
| Rosenbrock | 10 | 0.002 | 0.013 | 0.906 | 0.004 | 0.015 | 0.770 | 0.074 | 0.052 | 0.918 |
| Styblinski-Tang | 10 | 0.196 | 0.220 | 1.174 | 0.223 | 0.337 | 1.126 | 0.559 | 0.496 | 1.219 |
| Ackley | 20 | 0.629 | 0.219 | 0.945 | 0.282 | 0.292 | 0.950 | 0.226 | 0.586 | 0.917 |
| Levy | 20 | 0.128 | 0.241 | 0.839 | 0.063 | 0.113 | 0.914 | 0.140 | 0.182 | 1.056 |
| Powell | 20 | 0.093 | 0.081 | 0.809 | 0.010 | 0.074 | 0.689 | 0.028 | 0.110 | 0.870 |
| Rastrigin | 20 | 0.686 | 0.600 | 0.864 | 0.610 | 0.635 | 0.838 | 0.541 | 0.555 | 0.852 |
| Rosenbrock | 20 | 0.047 | 0.105 | 0.591 | 0.004 | 0.048 | 0.578 | 0.036 | 0.051 | 0.903 |
| Styblinski-Tang | 20 | 0.426 | 0.398 | 1.113 | 0.378 | 0.504 | 1.107 | 0.691 | 0.578 | 1.177 |
| Ackley | 50 | 0.895 | 0.464 | 0.949 | 0.738 | 0.606 | 0.947 | 0.177 | 0.530 | 0.874 |
| Levy | 50 | 0.055 | 0.029 | 0.611 | 0.033 | 0.085 | 0.681 | 0.051 | 0.268 | 0.892 |
| Powell | 50 | 0.018 | 0.021 | 0.542 | 0.014 | 0.078 | 0.499 | 0.018 | 0.064 | 0.785 |
| Rastrigin | 50 | 0.793 | 0.573 | 0.813 | 0.653 | 0.592 | 0.810 | 0.795 | 0.585 | 0.768 |
| Rosenbrock | 50 | 0.016 | 0.021 | 0.539 | 0.049 | 0.031 | 0.520 | 0.010 | 0.048 | 0.594 |
| Styblinski-Tang | 50 | 0.463 | 0.478 | 1.012 | 0.676 | 0.574 | 1.196 | 0.681 | 0.727 | 0.981 |
| Ackley | 100 | 0.718 | 0.636 | 0.948 | 0.900 | 0.466 | 0.935 | 0.137 | 0.321 | 0.863 |
| Emb. Hartmann 6 | 100 | 0.068 | 0.144 | 0.573 | 0.035 | 0.086 | 0.863 | 0.175 | 0.172 | 0.692 |
| Levy | 100 | 0.119 | 0.031 | 0.615 | 0.044 | 0.164 | 0.716 | 0.042 | 0.056 | 0.586 |
| Powell | 100 | 0.094 | 0.011 | 0.465 | 0.027 | 0.041 | 0.493 | 0.013 | 0.018 | 0.524 |
| Rastrigin | 100 | 0.506 | 0.501 | 0.759 | 0.604 | 0.557 | 0.832 | 0.540 | 0.544 | 0.780 |
| Rosenbrock | 100 | 0.114 | 0.044 | 0.518 | 0.027 | 0.048 | 0.589 | 0.014 | 0.031 | 0.507 |
| Styblinski-Tang | 100 | 0.389 | 0.522 | 0.924 | 0.503 | 0.582 | 1.203 | 0.562 | 0.742 | 0.930 |
| **Mean** | | 0.291 | 0.256 | 0.872 | 0.257 | 0.265 | 0.889 | 0.261 | 0.292 | 0.904 |
| **Median** | | 0.134 | 0.219 | 0.931 | 0.092 | 0.164 | 0.943 | 0.175 | 0.172 | 0.917 |

known by round 10. Rather, we assume that this a consequence of the challenges of MC-based optimization of the acquisition function at large $Q$.

## 5.2 BO under heteroskedastic noise

We compare performance of meanBEEBO and $q$-UCB on the 3-optimum Branin function. Under heteroskedastic noise, we find that BEEBO preferentially optimizes towards the low-noise optimum 1 at the expense of the noisy optima 2 and 3 (Figure 2) and is therefore risk-averse. In the homoskedastic case, BEEBO does not exhibit this preference and optimizes for multiple optima. As expected, $q$-UCB, which only uses the model posterior variance $\sqrt{C(x)}$ instead of quantifying the actual information gain, does not display any preference for low-noise optima, showing similar behaviour under heteroskedastic and homoskedastic noise and remaining risk-neutral.

## 6 Discussion

We introduce BEEBO, an acquisition function for BO with GPs that can be optimized analytically and that scales natively to batched acquisition. By exploiting the independence of the information gain $I(\mathbf{x})$ on measurements $\mathbf{y}$ when using GP surrogates, BEEBO models the interdependence of unknown points $\mathbf{x}$ in a batch and can optimize their positions jointly using gradient descent.

BEEBO enables full control of its explore-exploit trade-off using a hyperparameter $T$ that directly balances two terms, akin to UCB. Unlike in the reparametrization-based $q$-methods, BEEBO's $T$ has predictable behaviour also at increasing batch sizes.

The numerical complexity of BEEBO is dominated by the need to compute the inverse of $M_{\text{aug}}$ in Equation 11, which in a plain implementation scales as $\mathcal{O}((N + Q)^3)$. However, this can be reduced to $\mathcal{O}(N^2 Q)$; specifically, the Cholesky decomposition of $M_{\text{aug}}$ can be expressed as $Q$ rank-1 updates of the pre-computed Cholesky decomposition of $M_D$, where each update will have the complexity of $\mathcal{O}(N^2)$. The calculation of the energy, $E$, and the information gain, $I$, scales as $\mathcal{O}(N \cdot Q)$ and $\mathcal{O}(Q^3)$, respectively, and are thus sub-dominant to the update needed for $M_{\text{aug}}^{-1}$. For large $N$ this

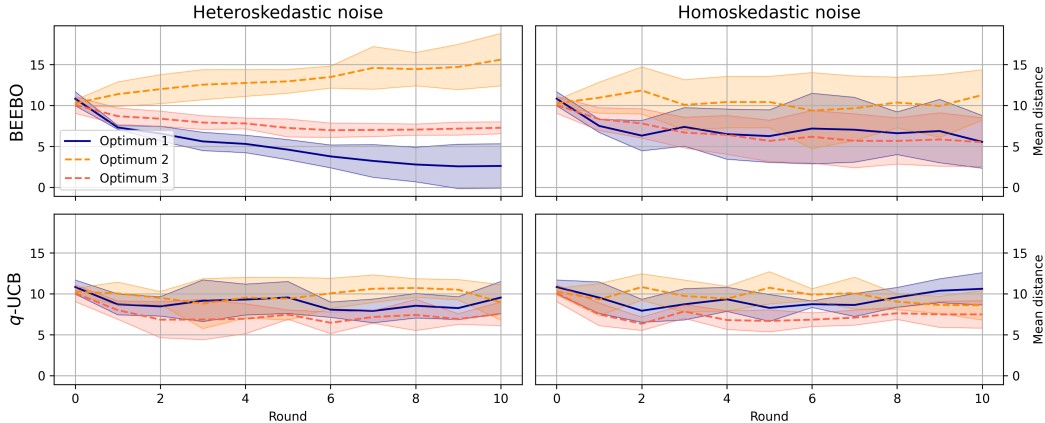

Figure 2: Mean distances of acquired points to the different optima of the Branin function. Under heteroskedastic noise, BEEBO is risk-averse and preferentially optimizes towards the low-noise optimum 1. Under homoskedastic noise, there is no preference. $q$-UCB does not adapt its behaviour to noise, remaining risk-neutral. The means and standard deviations over five replicates are shown.

approach may nevertheless become prohibitively slow. To overcome this limitation, methods for scalable GPs and fast predictive covariances such as LOVE [59] can be considered. The LOVE method allows a further reduction of the complexity of the Cholelsky update of $M_{\text{aug}}$ to $\mathcal{O}(N \cdot r \cdot Q)$ [60], where $r$ is the rank of the LOVE approximation for $M_D$, typically $r \ll N$.

As opposed to $q$-UCB, BEEBO can take heteroskedastic noise into account when computing the information gain, and preferentially acquires more informative low-noise points. We note that when the noise function $\sigma^2(x)$ is unknown, and needs to be explored at the same time with $f_{\text{true}}(x)$, it is critical that the initial random points sufficiently capture the noise landscape well enough for the information gain component to be useful, as the uncertainty of the surrogate on $\sigma^2(x)$ is not used. This would require a fully Bayesian approach that integrates over the distribution of $\sigma^2(x)$. The problem does not arise if the heteroskedastic noise of an experiment is known beforehand by e.g. instrument calibration. While not the focus of this work, we note that using the information gain could also be beneficial in sequential single-sample BO on heteroskedastic problems.

# 7 Outlook

In our experiments, we have focused on maintaining consistent explore-exploit ratios throughout the optimization rounds to ensure an equitable experimental comparison with $q$-UCB and demonstrate the effect of the hyperparameter choice. However, a more dynamic approach involving variable ratios could be more effective in real-world applications with a predetermined number of rounds [61]. Adopting a fully Bayesian perspective, one could consider the temperature hyperparameter $T$ as a random variable. This opens up an intriguing avenue for BEEBO, where $T$ could be drawn from a prior distribution that e.g. varies across optimization rounds, depending on the specific application. By tailoring this distribution, one could encourage a high level of exploration in the initial rounds, gradually transitioning towards a more exploitation-focused approach towards the end. In the presented experiments, we have implemented this as a strict constraint, maintaining a fixed $T$ until the final round, at which point we shift to full exploitation, i.e., $T = 0$.

While not explored in this work, we note that the BEEBO expression could naturally be extended to multi-objective optimization problems by capitalizing on GPs that handle vector-valued functions, such as multi-task GPs [12, 62]. Through e.g. the usage of the *intrinsic model of coregionalization*, we obtain a covariance function $k$, and thereby a covariance matrix $C(\mathbf{x})$, over all input-task pairs. As the multi-task covariance matrix is jointly Gaussian, the expression of the information gain remains unchanged and can be computed like in the single-task case. The energy $E(\mathbf{x})$ becomes vector-valued, providing an energy term for each of the tasks. This would allow for the introduction of task-specific

weights in the acquisition function. As the extension only affects the surrogate model, the scaling remains cubic in the number of input-task observations.

Beyond GPs, BEEBO could be generalized to work with any probabilistic model. However, GPs are unique in that $H_{\text{aug}}$ is available in closed form and can be used to compute $I(\mathbf{x})$ analytically, without solving the integral over $\mathbf{y}$ in Equation 4. Other models may require approximations and sampling-based approaches for computing the information gain.

## Availability

A BoTorch implementation of BEEBO is available at

`https://github.com/novonordisk-research/BEE-BO.`

## Acknowledgments and Disclosure of Funding

We thank Christoffer Riis, Jan C. Refsgaard and Kilian W. Conde-Frieboes for helpful discussions. FT was funded in part by the Novo Nordisk Foundation through the Center for Basic Machine Learning Research in Life Science (NNF20OC0062606).

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

# Appendix

## Table of Contents

## A  Approximating the expectation of the softmax weighted sum

### A.1  Motivation

We are free to choose any energy function $\tilde{E}$ in BEEBO, the only requirement being that we are able to compute an expectation of $\tilde{E}$ in order to obtain the scalar summary $E = \mathbb{E}[-\tilde{E}(\mathbf{f})]$. Of particular interest is the softmax weighted sum,

$$\tilde{E}(\mathbf{f}) = \sum_{i=1}^{Q} \text{softmax}(\beta\mathbf{f})_i f_i, \tag{A1}$$

where $\beta$ is the softmax inverse temperature. The softmax weight vector $\boldsymbol{\omega}$ computed as

$$\omega_i = \frac{\exp(\beta f_i)}{\sum_{j=1}^{Q} \exp(\beta f_j) + \exp(\beta_y y_{max})} \tag{A2}$$

where $\exp(\beta_y y_{max})$ is an optional reference threshold value, as in expected improvement, which we set to 0 if not used ($\beta_y$ is either simply $\beta$ or a dynamically scaled value that ensures $\tilde{E}(\mathbf{f})$ does

not become 0, see Equation A28). The parameter $\beta$ allows us to interpolate between two extreme regimes,

$$\tilde{E}(\mathbf{f}) = \frac{1}{Q} \sum_{i=1}^{Q} f_i \qquad \text{for} \beta \to 0 \tag{A3}$$

$$\tilde{E}(\mathbf{f}) = \max(\mathbf{f}) \qquad \text{for} \beta \to \infty \tag{A4}$$

In the first regime, all points of a batch equally contribute to the energy, whereas in the second regime only the single point "responsible" for the maximum is controlling the energy. Note that for numerical reasons, operating towards the $\beta \to \infty$ limit is impractical, as it will lead to zero gradients for all but one point, preventing optimization. We can set $\beta = A^{-1/2}$, with $A$ being the prior uncertainty scale of the GP kernel. Since $A$ represents the expected energy fluctuations for points far from data, this weighting scheme will reflect a natural compromise between Equation A3 and Equation A4.

As opposed to the mean, in the general case, the expectation of the maximum of a $Q$-dimensional multivariate normal is not available in closed form. To our best knowledge, this is also the case for the softmax weighted sum. In the following, we derive a closed-form approximation of the expectation of the softmax of $\beta\mathbf{f}$ that can be used for gradient-based optimization.

## A.2 Derivation

Consider the softmax denominator

$$d(\mathbf{f}) = \sum_{j=1}^{Q} \exp(\beta f_j) + \exp(\beta_y y_{\text{max}}). \tag{A5}$$

We will Taylor expand $\ln(d)$ to the second order, using

$$\frac{\partial \ln(d)}{\partial f_i} = \beta \frac{1}{d} \exp(\beta f_i) = \beta \omega_i \tag{A6}$$

$$\frac{\partial^2 \ln(d)}{\partial f_i \partial f_j} = \beta^2 \left( -\frac{1}{d^2} \exp(\beta(f_i + f_j) + \frac{1}{d^2} \delta_{ij} \exp(\beta f_i)) \right) \tag{A7}$$

$$= \beta^2 (\omega_i \delta_{ij} - \omega_i \omega_j), \tag{A8}$$

where $\delta_{ij}$ is the Kronecker delta. So

$$\ln(d) \approx \ln(d_{\mathbf{a}}) + \beta \mathbf{w}^T \cdot \Delta\mathbf{f} + \frac{\beta^2}{2} \Delta\mathbf{f}^T \cdot W \cdot \Delta\mathbf{f}, \tag{A9}$$

where $\mathbf{a}$ is the Taylor expansion point, $d_{\mathbf{a}}$ is $d$ evaluated at $\mathbf{a}$, $\Delta\mathbf{f} = \mathbf{f} - \mathbf{a}$, $\mathbf{w}$ is the $Q$-dimensional vector $\boldsymbol{\omega}$ evaluated at $\mathbf{a}$, and $W$ is the $Q \times Q$ matrix $W = \text{diag}(\mathbf{w}) - \mathbf{w}\mathbf{w}^T$. Inserted into Equation A2 we have

$$\omega(\mathbf{f})_i \approx w_i * \exp(\beta \Delta f_i - \beta \mathbf{w}^T \cdot \Delta\mathbf{f} - \frac{\beta^2}{2} \Delta\mathbf{f}^T \cdot W \cdot \Delta\mathbf{f}) \tag{A10}$$

With this approximation we can calculate the expectation value

$$\mathbb{E}[\omega(\mathbf{f})_i * f_i] = \frac{w_i \sqrt{\det(C(\mathbf{x})^{-1})}}{(2\pi)^{Q/2}} \int \exp(\lambda_i(\mathbf{f})) * f_i df \tag{A11}$$

$$\lambda_i(\mathbf{f}) = \beta \Delta f_i - \beta \mathbf{w}^T \cdot \Delta\mathbf{f}^T - \frac{\beta^2}{2} \Delta\mathbf{f}^T \cdot W \cdot \Delta\mathbf{f} - \frac{1}{2}(\mathbf{f} - \boldsymbol{\mu})^T \cdot C(\mathbf{x})^{-1} \cdot (\mathbf{f} - \boldsymbol{\mu}) \tag{A12}$$

$$= c^{(i)} - \frac{1}{2}(\mathbf{f} - \boldsymbol{\nu}^{(i)})^T \cdot C(\mathbf{x})_{\text{softmax}}^{-1} \cdot (\mathbf{f} - \boldsymbol{\nu}^{(i)}), \tag{A13}$$

Where $c^{(i)}$, $\boldsymbol{\nu}^{(i)}$ and $C(\mathbf{x})_{\text{softmax}}$ are defined as follows:

$$C(\mathbf{x})_{\text{softmax}} = \left(C(\mathbf{x})^{-1} + \beta^2 W\right)^{-1} \tag{A14}$$

$$\mathbf{b}^{(i)} = \mathbf{e}^{(i)} - \mathbf{w} \tag{A15}$$

$$\boldsymbol{\nu}^{(i)} = C(\mathbf{x})_{\text{softmax}} \cdot \left(\beta \mathbf{b}^{(i)} + C(\mathbf{x})^{-1} \cdot \boldsymbol{\mu} + \beta^2 W \cdot \mathbf{a}\right) \tag{A16}$$

$$c^{(i)} = \frac{1}{2}(\boldsymbol{\nu}^{(i)})^T \cdot C(\mathbf{x})_{\text{softmax}}^{-1} \cdot \boldsymbol{\nu}^{(i)} - \beta(\mathbf{b}^{(i)})^T \cdot \mathbf{a} \tag{A17}$$

$$- \frac{\beta^2}{2}\mathbf{a}^T \cdot W \cdot \mathbf{a} - \frac{1}{2}\boldsymbol{\mu}^T \cdot C(\mathbf{x})^{-1} \cdot \boldsymbol{\mu} \tag{A18}$$

and $\mathbf{e}^{(i)}$ is the $i$'th basis vector with components $e_j^{(i)} = \delta_{ij}$. We can avoid the explict use of the precision matrix by rewriting the updated covariance matrix as

$$C(\mathbf{x})_{\text{softmax}} = \left(C(\mathbf{x})^{-1} \cdot (I + \beta^2 C(\mathbf{x}) \cdot W)\right)^{-1} = U(\mathbf{x}) \cdot C(\mathbf{x}),$$

where we have defined $U(\mathbf{x}) = \left(I + \beta^2 C(\mathbf{x}) \cdot W\right)^{-1}$. The updated mean vectors can conveniently be expressed as

$$\boldsymbol{\nu}^{(i)} = C(\mathbf{x})_{\text{softmax}} \cdot (\beta \mathbf{b}^{(i)} + C(\mathbf{x})^{-1} \cdot \boldsymbol{\mu} + \beta^2 W \cdot \mathbf{a}) \tag{A19}$$

$$\boldsymbol{\nu}^{(i)} = \beta C(\mathbf{x})_{\text{softmax}} \cdot \mathbf{e}^{(i)} + C(\mathbf{x})_{\text{softmax}} \cdot \left(-\beta \mathbf{w} + C(\mathbf{x})^{-1} \cdot \boldsymbol{\mu} + \beta^2 W \cdot \mathbf{a}\right) \tag{A20}$$

$$\boldsymbol{\nu}^{(i)} = \beta C(\mathbf{x})_{\text{softmax}} \cdot \mathbf{e}^{(i)} + \boldsymbol{\nu}', \tag{A21}$$

where $\boldsymbol{\nu}'$ is a constant vector for all $(i)$. Similarly

$$c^{(i)} = -\beta a_i + \frac{1}{2}\beta^2 \left(C(\mathbf{x})_{\text{softmax}}\right)_{i,i} + \beta \nu_i' + c' \tag{A22}$$

$$c' = \beta \mathbf{w}^T \cdot \mathbf{a} + \frac{1}{2}\boldsymbol{\nu}'^T \cdot C(\mathbf{x})_{\text{softmax}}^{-1} \cdot \boldsymbol{\nu}' - \frac{\beta^2}{2}\mathbf{a}^T \cdot W \cdot \mathbf{a} - \frac{1}{2}\boldsymbol{\mu}^T \cdot C(\mathbf{x})^{-1} \cdot \boldsymbol{\mu} \tag{A23}$$

with $c'$ again being a constant for all $(i)$. The expectation of the softmax weighted summary is given by

$$\mathbb{E}[\tilde{E}] = K * \sum_{i=1}^{Q} w_i * \exp(c^{(i)}) * \nu_i^{(i)} \tag{A24}$$

where $K = \frac{\sqrt{\det(C(\mathbf{x})^{-1})}}{\sqrt{\det(C(\mathbf{x})^{-1} + \beta^2 W)}} = \sqrt{\det(U(\mathbf{x}))}$. The most natural choice for the expansion point is $\mathbf{a} = \boldsymbol{\mu}$ in which case $\boldsymbol{\nu}^{(i)}$ and $c^{(i)}$ reduces to

$$\boldsymbol{\nu}^{(i)} = \beta C(\mathbf{x})_{\text{softmax}} \cdot \left(\mathbf{e}^{(i)} - \mathbf{w}\right) + \boldsymbol{\mu} \tag{A25}$$

$$c^{(i)} = \frac{\beta^2}{2}\left(\mathbf{e}^{(i)} - \mathbf{w}\right)^T \cdot C(\mathbf{x})_{\text{softmax}} \cdot \left(\mathbf{e}^{(i)} - \mathbf{w}\right) \tag{A26}$$

### A.3 Practical considerations

**Linear algebra**  For numerical reasons, we avoid computing $C(\mathbf{x})_{\text{softmax}}$ explicitly, and instead use the $U(\mathbf{x}) \cdot C(\mathbf{x})$ factorization to compute solutions with $U(\mathbf{x})^{-1} = I + \beta^2 C(\mathbf{x}) \cdot W$.

$$C(\mathbf{x})_{\text{softmax}} \cdot \left(\mathbf{e}^{(i)} - \mathbf{w}\right) = U(\mathbf{x}) \cdot \left(C(\mathbf{x}) \cdot \mathbf{e}^{(i)} - C(\mathbf{x}) \cdot \mathbf{w}\right) \tag{A27}$$

Following GPyTorch practices, we make use of the LinearOperator package to exploit the structure of $U(\mathbf{x})^{-1}$ as an AddedDiagLinearOperator when solving. For determinants, we find that Linear-Operator's logdet implementation gives nondeterministic results, and we therefore perform a dense cast before computing $K$ using default Pytorch.

While the factorization is numerically advantageous, it is still limited with regards to $\beta$. We find that at $\beta > 5$, numerical errors prevent a reliable calculation of the expectation. In practice, $A^{-1/2}$ lies in a range that allows numerically accurate solutions.

**Softmax** When $y_{\max}$ grows much larger than the softmax input vector $\mathbf{f}$ - a situation that can arise easily when initializing with random points for gradient-based optimization - the softmax weights $\boldsymbol{\omega}$ can become numerically zero for all "real" points, thus leading to $E(\mathbf{x}) = 0$, and vanishing gradients. As we always wish to preserve a minimal energy contribution from the real points, we parametrize the inverse temperature applied to $y_{\max}$, $\beta_y$, using a hyperparameter $\alpha$ that denotes the minimal fraction of probability mass pertaining to real points. This parametrization resembles the LogEI version of the expected improvement acquisition function [58] to address the problem of vanishing EI-gradients.

Let $N$ denote the softmax denominator excluding $y_{\max}$, $N = \sum_{j=1}^{Q} \exp(\beta \Delta f_i)$. We define

$$\exp(\beta_y \Delta y_{\max}) = \min\left(\frac{1-\alpha}{\alpha} N, \exp(\beta \Delta y_{\max})\right) \tag{A28}$$

We used $\alpha = 0.05$ as a default in all our experiments.

### A.4 Number of effective points

We can interpret the softmax as the number of effective points contributing to the energy of the batch. The entropy $H$ of the softmax is given by

$$H(\boldsymbol{\omega}) = -\sum_{i=1}^{Q} \omega_i \ln(\omega_i), \tag{A29}$$

and the number of effective points, $D_{\text{eff}}$, is $\exp(H(\boldsymbol{\omega}))$, so that

$$D_{\text{eff}} = \exp\left(-\sum_{i=1}^{Q} \omega_i \ln(\omega_i)\right). \tag{A30}$$

$D_{\text{eff}}$ is bounded by 1 (approaching the maximum) and $Q$ (approaching the mean). Note that if we include $y_{\max}$ in the softmax denominator, we add $\omega_{y_{\max}} \ln(\omega_{y_{\max}})$ to $H(\boldsymbol{\omega})$, and the resulting number becomes bounded by 1 and $Q + 1$.

## B Relationship to other acquisition strategies

In the following section, we will discuss how BEEBO is related to UCB, GIBBON, Determinantal Point Processes (DPP), the Local Penalization heuristic and RAHBO. We will base our analysis on meanBEEBO, as the softmax-mediated interdependency of points in maxBEEBO prevents a simple interpretation of the objective in a single-point stepwise manner and does not allow for the same direct analogies to other strategies.

### B.1 Relationship of BEEBO $T$ and UCB $\kappa$ hyperparameters

BEEBO bears some resemblance to the UCB acquisition function, which in the single particle mode, $Q = 1$, reads

$$a_{\text{UCB}}(x) = \mu(x) + \sqrt{\kappa}\sqrt{C(x)}, \tag{A31}$$

where the parameter $\kappa$ controls the balance between exploitation and exploration and $\mu(x)$ and $C(x)$ are respectively the mean and variance of the posterior distribution, $P(f \mid x, D)$, as before. We note that $a_{\text{UCB}}$ does not account for the uncertainty of the measurement at $x$, and therefore remains risk-neutral under heteroskedastic noise [11]. To understand the relationship between BEEBO and UCB, we will therefore limit ourselves to the homoskedastic case and furthermore assume that measurement variances $\sigma^2$ are much smaller than the typical prior variance of the GP surrogate, $A$, of $f$, e.g. $A \simeq N^{-1} Tr(K)$, so $\sigma^2 \ll A$ and $M^{-1} = (K + \sigma^2)^{-1} \approx K^{-1}$. In this limit, the variance of $f(x)$ after measurement (indexed at $i = n$, say) reduces to $\sigma^2$:

$$C(x) = \left((K^{-1} + \sigma^{-2}I)^{-1}\right)_{nn} = \left(K(K + \sigma^2 I)^{-1}\sigma^2\right)_{nn} \approx \sigma^2 \tag{A32}$$

and the information gain becomes

$$I(x) \approx \frac{1}{2} \ln(C(x)) - \log(\sigma).$$

Consequently, the gradient of the two acquisition functions reads

$$\nabla a_{\text{UCB}}(x) = \nabla \mu(x) + \frac{\sqrt{\kappa}}{2 \cdot \sqrt{C(x)}} \cdot \nabla C(x)$$

$$\nabla a_{\text{BEEBO}}(x) = \nabla \mu(x) + \frac{T}{2 \cdot C(x)} \cdot \nabla C(x).$$

The two gradients will be identical at points $x$ where the posterior uncertainties satisfy $\sqrt{C(x)} = \frac{T}{\sqrt{\kappa}}$. For comparison, we may desire equal gradients at iso-surfaces corresponding to a given fraction, $\nu$, of the prior uncertainty scale $\sqrt{A}$, by setting $T$ accordingly as $T = \nu \cdot \sqrt{A} \cdot \sqrt{\kappa}$. In our experiments, we use $\nu = \frac{1}{2}$ and configure BEEBO using a dimensionless $T'$ explore-exploit parameter, defined as $T' = \frac{T}{\sqrt{A}}$, and set $T' = \frac{1}{2}\sqrt{\kappa}$ for a given benchmark experiment.

## B.2   GIBBON

GIBBON [42] approximates the (intractable) *General-purpose max-value Entropy Search* acquisition function, which quantifies the mutual information $MI(f^*_{\text{true}}; \mathbf{y}|D)$ of a batch of measurements $\mathbf{y}$ and the unknown optimum $f^*_{\text{true}}$. It does so using a lower bound on the information gain and MC estimation of the expectation over $f^*_{\text{true}}$. It can be written as

$$\alpha_{\text{GIBBON}}(\mathbf{x}) = \frac{1}{2} \ln \det(R) - \frac{1}{2|\mathcal{M}|} \sum_{m \in \mathcal{M}} \sum_{i=1}^{Q} \ln \left( 1 - \rho_i^2 \frac{\phi(\gamma_i(m))}{\phi(\gamma_i(m))} \left[ \gamma_i(m) + \frac{\phi(\gamma_i(m))}{\phi(\gamma_i(m))} \right] \right)$$

$$\alpha_{\text{GIBBON}}(\mathbf{x}) = \frac{1}{2} \ln \det(R) + \sum_{i=1}^{Q} \hat{\alpha}_{\text{GIBBON}}(x_i), \tag{A33}$$

where $R$ is the correlation matrix with entries $R_{ij} = \frac{C(\mathbf{x})_{ij}}{\sqrt{C(\mathbf{x})_{ii} C(\mathbf{x})_{jj}}}$, $\mathcal{M}$ is a set of samples for the max-value $f^*_{\text{true}}$, and $\rho_i$ is the correlation of $y_i$ and $f_{\text{true}}(x_i)$. $\phi$ and $\phi$ are the standard normal cumulative distribution and probability density functions, and $\gamma_i(m) = \frac{m-\mu}{\sigma}$.

The definition of BEEBO introduced in Equation 7, with the scalar summarization function set to the expected mean, $E(\mathbf{x}) = -\frac{1}{Q} \sum_{i=1}^{Q} \mu(x_i)$, gives

$$\alpha_{\text{BEEBO}}(\mathbf{x}) = T * \frac{1}{2} \left( \ln \det (C(\mathbf{x})) - \ln \det (C_{\text{aug}}(\mathbf{x})) \right) + \sum_{i=1}^{Q} \mu(x_i). \tag{A34}$$

From the second formulation of GIBBON, it becomes obvious that although being distinct in their motivation and derivation, BEEBO and GIBBON implement acquisition functions with a similar structure. Taking an information theoretic and multi-fidelity BO standpoint, GIBBON refers to this trade-off as *diversity* against *quality*, whereas in BEEBO we follow the intuitions of UCB, and use *exploration* and *exploitation*.

- *Quality - Exploitation*: GIBBON employs an MC estimate of the lower bound approximation of the information gain provided by each point, whereas BEEBO directly summarizes the optimality of all points in closed form, either as their mean or an approximated softmax weighted sum.

- *Diversity - Exploration*: In GIBBON, the diversity derived from the differential entropy $H(\mathbf{f}|D, \mathbf{x})$ is the entropy of the posterior correlation $\frac{1}{2} \ln \det(R)$. In BEEBO, we employ the *reduction* of entropy, the information gain $I(\mathbf{x})$. Under homoskedastic noise, $I(\mathbf{x}) \propto \ln \det(C(\mathbf{x}))$. Since $R(\mathbf{x}) = \text{diag}(C(\mathbf{x}))^{-1/2} \cdot C(\mathbf{x}) \cdot \text{diag}(C(\mathbf{x}))^{-1/2}$, we have that $\ln \det(R) = \ln \det(C(\mathbf{x})) - \sum_i^{Q} \ln(C(\mathbf{x})_{ii})$. Therefore, maximizing the log determinant of $R$ penalizes points that have high variance.

Therefore, while GIBBON presents an attractive approximation of max-value Entropy Search for batched acquisition, BEEBO is an alternative that avoids approximating a quality criterion using MC. Moreover, GIBBON's diversity criterion implicitly penalizes points that have high variance, whereas BEEBO's criterion maximizes the reduction of variance. We find that BEEBO is orders of magnitudes faster to compute than GIBBON (Figure A3).

In the context of large batches ($Q >> 10$), a modification of GIBBON exists that is further similar to BEEBO. Departing from the strict max-value entropy search derivation, a scaling factor $Q^{-2}$ is introduced to counteract a growing dominance of the diversity term:

$$\alpha_{\text{scaledGIBBON}}(\mathbf{x}) = \frac{1}{2Q^2} * \ln \det(R) + \sum_{i=1}^{Q} \hat{\alpha}_{\text{MES}}(x_i). \tag{A35}$$

This scaling is motivated by the fact that $R$ contains $Q^2$ elements. However, we note that $R$ is summarized by its log determinant, which scales linearly in $Q$: As the determinant is the product of the eigenvalues, the log determinant is the sum of the log-eigenvalues. The number of eigenvalues scales linearly with matrix size $Q$, and so does the log determinant.

## B.3  Determinantal Point Processes

A Determinantal Point Process [43] specifies a probability over a set of points, or a "configuration of points" drawn from a ground set. Specifically, the probability of a set of $Q$ points $\mathbf{x}$ is given by

$$P(\mathbf{x}) \propto \det\left(L_{\mathbf{x}}\right), \tag{A36}$$

where $L_{\mathbf{x}}$ is a $Q \times Q$ symmetric matrix. Kulesza et al. [43] provide a decomposition of the general DPP kernel $L$ that makes *quality* and *diversity* components explicit, so that

$$L_{ij} = q(x_i)q(x_j)k(x_i, x_j), \tag{A37}$$

with $k$ being a $\mathbb{R}^d \times \mathbb{R}^d \to \mathbb{R}^+$ similarity kernel, and $q$ being a unary $\mathbb{R}^d \to \mathbb{R}$ scalar quality function. This framework is naturally amenable to batch BO, as we seek to select a collection of points that trade off quality (optimality) and diversity. Note that both $k$ and $q$ are distinct functions that need to be specified by the user, leading to the practical complication that they must be chosen very carefully so that their scales do not dominate each other, which limits the utility of this decomposition in practice [42].

In the following, we show how BEEBO is equivalent to a DPP, and derive the necessary $k$ and $q$. Again, we consider BEEBO

$$\alpha_{\text{BEEBO}}(\mathbf{x}) = -E(\mathbf{x}) + T * I(\mathbf{x}), \tag{A38}$$

with the scalar summarization function set to $E(\mathbf{x}) = -\sum_{i=1}^{Q} f(x_i)$. We will first focus on the information gain term $I(\mathbf{x})$, which we can rearrange as

$$I(\mathbf{x}) = \frac{1}{2} \ln \det(C(\mathbf{x})) - \frac{1}{2} \ln \det(C_{\text{aug}}(\mathbf{x})) = \frac{1}{2} \ln \det\left(C(\mathbf{x}) \cdot C_{\text{aug}}^{-1}(\mathbf{x})\right). \tag{A39}$$

Our similarity kernel $k$ is therefore given by the entries of the matrix $S = C(\mathbf{x}) \cdot C_{\text{aug}}(\mathbf{x})^{-1}$, so that $k(x_i, x_j) = S_{ij}$. Note that due to the augmented covariance term, the implied $k$ also depends on all other currently selected points in $\mathbf{x}$, and $L_{\mathbf{x}}$ is not a submatrix of an all-sample $L$. Therefore, BEEBO does not implement a DPP under heteroskedastic noise. However, if we only consider homoskedastic noise, BEEBO's $I(\mathbf{x})$ simplifies to the posterior entropy [63], and therefore $S = C(\mathbf{x})$. As $C(\mathbf{x})$ can be accessed as a submatrix of an all-sample $C$, this permits a DPP.

Given the choice of $E(\mathbf{f})$, we can rewrite BEEBO as

$$\alpha_{\text{BEEBO}}(\mathbf{x}) = \ln\det\left(S\right) * T * \frac{1}{2} + \sum_{i=1}^{Q} \mu_i$$

$$\frac{2}{T} * \alpha_{\text{BEEBO}}(\mathbf{x}) = \ln\det\left(S\right) + \sum_{i=1}^{Q} \frac{2}{T} * \mu_i$$

$$\frac{2}{T} * \alpha_{\text{BEEBO}}(\mathbf{x}) = \ln\det\left(S\right) + \ln\det\left(D\right)) \qquad \text{with } D = \text{diag}(\exp(\frac{2}{T} * \boldsymbol{\mu}))$$

$$\frac{2}{T} * \alpha_{\text{BEEBO}}(\mathbf{x}) = \ln\det\left(D^{\frac{1}{2}} \cdot S \cdot D^{\frac{1}{2}}\right) \qquad D_{ii}^{\frac{1}{2}} = \sqrt{\exp(\frac{2}{T}\mu_i)} = \exp(\frac{1}{T}\mu_i)$$

$$\alpha_{\text{BEEBO}}(\mathbf{x}) = \ln\det\left(D^{\frac{1}{2}} \cdot S \cdot D^{\frac{1}{2}}\right) * T * \frac{1}{2}$$

$$\alpha_{\text{BEEBO}}(\mathbf{x}) = \ln\det\left(L\right) * T * \frac{1}{2}$$

$$\tag{A40}$$

where $L$ is a matrix with entries $L_{ij} = S_{ij}\exp(\frac{1}{T} * \mu_i)\exp(\frac{1}{T} * \mu_j)$. BEEBO therefore uses the DPP quality function $q(x_i) = \exp(\frac{1}{T} * \mu_i)$, and, like proven previously for GIBBON, a batch $\mathbf{x}$ with maximal $\alpha_{\text{BEEBO}}$ corresponds to the MAP of a DPP.

## B.4   Local penalization

Local penalization (LP) is a greedy batch selection strategy that given any arbitrary single-point acquisition function, ensures diversity by applying a *penalization function* $\psi(x, x')$ that downweights the acquisition value of candidate locations $x$ based on their proximity to already selected points. The criterion for selecting $x_i$ is given by

$$x_i = \arg\max \alpha(x) \prod_{j=1}^{i-1} \psi(x, x_j). \tag{A41}$$

Note that in this formulation, the product includes all previously selected points, not just the current batch. The penalization function $\psi$ may in principle be chosen freely. Gonzalez et al. [35] propose exploiting the fact that $f_{\text{true}}$ is Lipschitz continuous in order to bound the position of the unknown optimum and penalize accordingly. The Lipschitz constant $L$ is inferred from the GP surrogate and used to parametrize $\psi$. In LP, acquisition function optimization proceeds iteratively. After an $x_i$ is chosen, the corresponding penalizing multiplier is added to the objective before optimizing for the next $x_{i+1}$.

While BEEBO enables optimization to proceed in parallel, it is of course possible to also optimize BEEBO greedily (under homoskedastic noise, $I$ is submodular). In this case, it implements an LP strategy where $\alpha(x) = \mu(x)$. Rather than a product of individual $\mathbb{R}^d \times \mathbb{R}^d \to \mathbb{R}$ function evaluations, the penalizer implied by BEEBO is the information gain $I(\mathbf{x}) : \mathbb{R}^{i \times d} \to \mathbb{R}$ that we evaluate by concatenating a candidate point to the already acquired $x$ at each iteration. Like in GIBBON, this constitutes an LP strategy that does not require estimation of any properties of $f_{\text{true}}$ beyond learning the GP surrogate.

## B.5   RAHBO

Risk-averse Heteroskedastic Bayesian Optimization (RAHBO) [11] is a UCB-derived single-point acquisition function that avoids heteroskedastic risk, preferentially selecting points with low noise. While it is not applicable to batched acquisition directly, we here compare it to single-sample BEEBO to highlight different ways of addressing noise. Given a heteroskedastic surrogate model that learns an additional GP for the noise, the *variance proxy*, RAHBO reads

$$\alpha_{\mathrm{RAHBO}}(x) = \mathrm{UCB}_f(x) - \alpha * \mathrm{LCB}_{\mathrm{var}}(x)$$
$$\alpha_{\mathrm{RAHBO}}(x) = \mu_f(x) + \beta_f * \sigma_f(x) - \alpha(\mu_{\mathrm{var}}(x) - \beta_{\mathrm{var}} * \sigma_{\mathrm{var}}(x)) \quad , \tag{A42}$$

where $\mu_f$ and $\sigma_f$ are the posterior mean and variance of the surrogate model and $\beta_f$ is the standard UCB trade-off hyperparameter, yielding the standard upper confidence bound $\mathrm{UCB}_f$. $\alpha$ is the chosen risk tolerance, and LCB is the lower confidence bound of the variance GP with posterior mean $\mu_{\mathrm{var}}$ and variance $\sigma_{\mathrm{var}}$ traded off using $\beta_{\mathrm{var}}$.

At $Q = 1$, BEEBO can be expressed as

$$\alpha_{\mathrm{BEEBO}} = \mu_f(x) + T * \frac{1}{2}\ln(\sigma_f(x)) - T * \frac{1}{2}\ln(\sigma_f^{\mathrm{aug}}(x))$$
$$\alpha_{\mathrm{BEEBO}} = \mu_f(x) + T * \frac{1}{2}\ln\left(\frac{\sigma_f(x)}{\sigma_f^{\mathrm{aug}}(x)}\right), \tag{A43}$$

where the variance proxy at $x$ is considered via the augmented posterior variance $\sigma_f^{\mathrm{aug}}$.

While RAHBO penalizes risk on an absolute scale, subject to $\alpha$, BEEBO optimizes for high uncertainty reduction, quantified as the log ratio of the variance before and after making measurements.

Moreover, RAHBO differentiates between known and unknown variance proxies, and uses the $\mathrm{LCB}_{\mathrm{var}}$ term to discount the predicted variance according to its uncertainty. In its closed-form analytical expression, BEEBO does not permit for the uncertainty of the variance proxy to be taken into account, being more similar to the known variance RAHBO

$$\alpha_{\mathrm{RAHBO}}(x) = \mu_f(x) + \beta_f * \sigma_f(x) - \alpha\mu_{\mathrm{var}}(x) \tag{A44}$$

where $\mu_{\mathrm{var}}$ is a noise-free proxy. Either a sampling-based approach, or approximations to $I(x)$ would need to be introduced to handle variance proxy uncertainty in BEEBO.

## C  Implementation details

### C.1  Acquisition function optimization

BEEBO was implemented for full compatibility with the BoTorch framework (version 0.9.4) [52] as an `AnalyticAcquistionFunction`. Standard BoTorch utilities for initializing and training GPs, initializing $q$-batches and performing gradient descent optimization of the acquisition function are used. We trained GPyTorch (version 1.11) [57] GP models with KeOps [64] Matérn 5/2 kernels (following BoTorch defaults with a separate length scale for each input dimension, and Gamma priors on the length and output scales). Log determinants for the information gain were computed using singular value decomposition for numerical stability.

GPyTorch provides a `get_fantasy_model` method that allows for the efficient augmentation of the training data of a GP with a set of points, as done in BEEBO. However, we observed that GPyTorch's implementation suffers from GPU memory leaks when used with automatic differentiation enabled. We therefore instantiate augmented models explicitly, not making use of the (more efficient) augmentation strategy.

All experiments were performed with double precision. `SobolQMCNormalSampler` was used for acquisition functions making use of the reparametrization trick. Experiments were run on individual Nvidia RTX 6000 and V100 GPUs. Five replicates for the benchmarking experiments required a total of approx. 5,000 RTX 6000 GPU hours, with the majority of the run time dedicated to the GIBBON baseline, rather than BEEBO itself (Figure A3, Table A9).

### C.2  Benchmark BO methods

All methods were benchmarked in BoTorch. For $q$-EI, we used LogEI [58]. For TS, 10.000 base Sobol samples were drawn and sampled with `MaxPosteriorSampling` using the Cholesky decomposition

of the covariance matrix. GIBBON was optimized using sequential optimization following the BoTorch tutorial. We additionally implemented a custom version of GIBBON that applies the $Q^{-2}$ scaling factor to the diversity term, as proposed in GIBBON's supplementary material. We used 100,000 random discretized candidates for max-value sampling. In a few iterations, optimizing GIBBON seemed challenging, with BoTorch reporting that no nonzero initialization candidate could be identified. KB was optimized using a custom greedy optimization loop with fantasized observations, using (single-sample) LogEI as the underlying acquisition function. TuRBO-1 was optimized following its BoTorch tutorial. None of the methods use a hyperparameter for controlling their explore-exploit trade-off. The results are therefore based on 10 iterations at defaults.

## C.3 Test problems

**Test functions**  All test functions were used in their BoTorch implementations. As done in previous work, the embedded Hartmann function was created by appending all-0 dummy dimensions to the original six dimensions [53, 54, 50].

**Control problems**  We consider two control problems from previous work: A 14-dimensional parameter tuning task for controlling robot arms pushing two objects to a target location [55], and a 60-dimensional trajectory planning task for a rover navigating through a maze of obstacles [56]. Instead of converting the problem objectives into rewards as in the original work, we operate on the actual minimization objectives directly (distance to target, navigation loss), and follow BoTorch's approach of simply inverting the objective in order to yield maximization problems. Both problems were adapted from their available implementations in Wang et al. [56] to follow the BoTorch test problem API.

## C.4 Heteroskedastic noise

The (inverted) Branin function has three global optima $f(x^*) = -0.397887$ at $x_1^* = (9.42478, 2.475), x_2^* = (-\pi, 12.275)$ and $x_3^* = (\pi, 2.275)$. We define heteroskedastic noise so that the variance is maximal at $x_2^*$ and $x_3^*$. The noise decays exponentially with the distance from any of the two noised optima at a rate $\lambda$.

$$\sigma^2(x) = \sigma_{max}^2 * \exp(-\lambda * \min(\|x - x_2^*\|_2, \|x - x_3^*\|_2) \tag{A45}$$

For our experiments, we set $\sigma_{max}^2 = 100$ and $\lambda = 0.05$. As the surrogate function, we use a `HeteroskedasticSingleTaskGP` provided in BoTorch. This model learns two GPs simultaneously, one for the function $f(x)$ and one for the (also unknown) variance function $\sigma^2(x)$. When querying the oracle with a batch of points, noised observations of $f(x)$ are provided together with the true $\sigma^2$ at each point. The homoskedastic control experiment uses a `SingleTaskGP` with inferred noise level. The homoskedastic noise is set to $\sigma^2 = 77.5$, which is the average noise level of the heteroskedastic function over the whole domain.

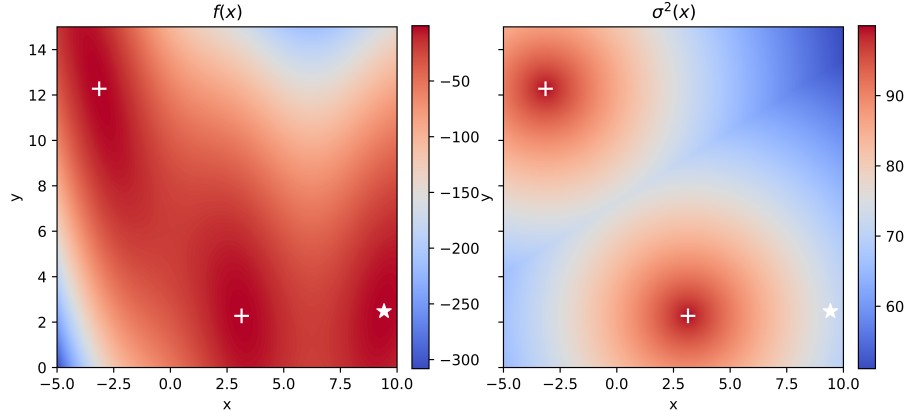

Figure A1: The Branin function with added heteroskedastic noise following Equation A45. $\sigma^2_{max} = 100$, $\lambda = 0.05$.

# D    Extended results

## D.1    Results including additional baselines

Table A1: BO on noise-free synthetic test problems. The normalized highest observed value after 10 rounds of BO with $q$=100 is shown. Colors are normalized row-wise. The BEE-BO and $q$-UCB columns are equivalent to Table 2. Higher means better. Results are means over five replicate runs.

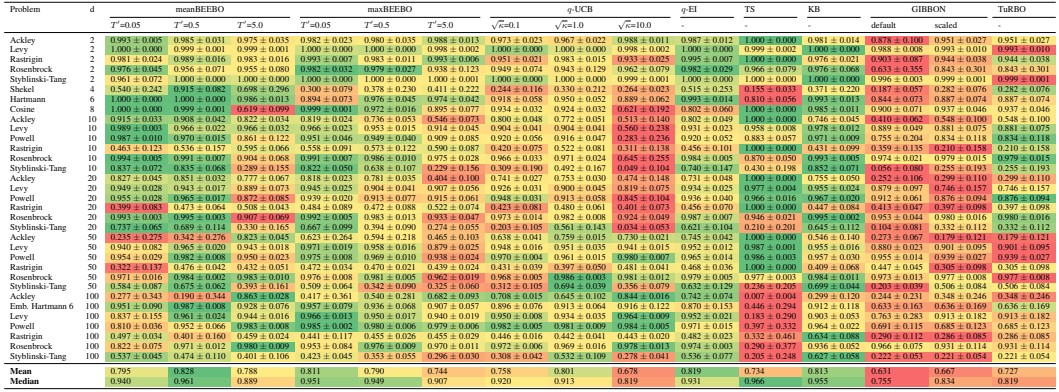

| Problem | d | meanBEEBO | | | maxBEEBO | | | q-UCB | | | q-EI | TS | KB | GIBBON | | TuRBO |
|---|---|---|---|---|---|---|---|---|---|---|---|---|---|---|---|---|
| | | $T'$=0.05 | $T'$=0.5 | $T'$=5.0 | $T'$=0.05 | $T'$=0.5 | $T'$=5.0 | $\sqrt{\kappa}$=0.1 | $\sqrt{\kappa}$=1.0 | $\sqrt{\kappa}$=10.0 | - | - | - | default | scaled | - |
| Ackley | 2 | 0.993±0.005 | 0.985±0.031 | 0.975±0.035 | 0.982±0.023 | 0.980±0.035 | 0.988±0.013 | 0.973±0.023 | 0.967±0.022 | 0.988±0.011 | 0.987±0.012 | 1.000±0.000 | 0.981±0.014 | 0.878±0.100 | 0.951±0.027 | 0.951±0.027 |
| Levy | 2 | 1.000±0.000 | 0.999±0.001 | 0.999±0.001 | 1.000±0.000 | 1.000±0.000 | 0.998±0.002 | 1.000±0.000 | 1.000±0.000 | 0.998±0.002 | 1.000±0.000 | 0.999±0.002 | 1.000±0.000 | 0.988±0.008 | 0.993±0.010 | 0.993±0.010 |
| Rastrigin | 2 | 0.981±0.024 | 0.989±0.016 | 0.983±0.016 | 0.993±0.007 | 0.983±0.011 | 0.993±0.006 | 0.951±0.021 | 0.983±0.015 | 0.933±0.025 | 0.995±0.007 | 1.000±0.000 | 0.976±0.021 | 0.903±0.087 | 0.944±0.038 | 0.944±0.038 |
| Rosenbrock | 2 | 0.976±0.045 | 0.956±0.071 | 0.955±0.080 | 0.982±0.032 | 0.979±0.027 | 0.938±0.123 | 0.949±0.074 | 0.943±0.129 | 0.962±0.079 | 0.982±0.029 | 0.966±0.079 | 0.976±0.068 | 0.633±0.355 | 0.843±0.301 | 0.843±0.301 |
| Styblinski-Tang | 2 | 0.961±0.072 | 1.000±0.000 | 1.000±0.000 | 1.000±0.000 | 1.000±0.000 | 1.000±0.001 | 1.000±0.000 | 1.000±0.000 | 0.999±0.001 | 1.000±0.000 | 1.000±0.000 | 1.000±0.000 | 0.996±0.003 | 0.999±0.001 | 0.999±0.001 |
| Shekel | 4 | 0.540±0.242 | 0.915±0.082 | 0.698±0.296 | 0.300±0.079 | 0.378±0.230 | 0.411±0.222 | 0.244±0.116 | 0.330±0.212 | 0.264±0.023 | 0.515±0.253 | 0.155±0.033 | 0.371±0.220 | 0.187±0.057 | 0.282±0.076 | 0.282±0.076 |
| Hartmann | 6 | 1.000±0.000 | 1.000±0.000 | 0.986±0.013 | 0.894±0.073 | 0.976±0.045 | 0.974±0.042 | 0.918±0.058 | 0.950±0.052 | 0.889±0.062 | 0.993±0.014 | 0.810±0.056 | 0.993±0.013 | 0.844±0.073 | 0.887±0.074 | 0.887±0.074 |
| Cosine | 8 | 1.000±0.000 | 0.999±0.001 | 0.619±0.099 | 0.999±0.001 | 0.972±0.016 | 0.895±0.077 | 0.934±0.032 | 0.924±0.032 | 0.621±0.192 | 0.802±0.060 | 1.000±0.000 | 0.985±0.011 | 0.900±0.071 | 0.937±0.046 | 0.937±0.046 |
| Ackley | 10 | 0.915±0.033 | 0.908±0.042 | 0.822±0.034 | 0.819±0.024 | 0.736±0.053 | 0.546±0.073 | 0.800±0.048 | 0.772±0.051 | 0.513±0.140 | 0.802±0.049 | 1.000±0.000 | 0.746±0.045 | 0.410±0.062 | 0.548±0.100 | 0.548±0.100 |
| Levy | 10 | 0.989±0.001 | 0.966±0.022 | 0.966±0.032 | 0.966±0.023 | 0.953±0.015 | 0.914±0.045 | 0.904±0.041 | 0.904±0.041 | 0.560±0.238 | 0.931±0.023 | 0.958±0.008 | 0.978±0.012 | 0.889±0.049 | 0.881±0.075 | 0.881±0.075 |
| Powell | 10 | 0.987±0.010 | 0.970±0.015 | 0.861±0.122 | 0.951±0.046 | 0.949±0.040 | 0.909±0.085 | 0.920±0.056 | 0.916±0.047 | 0.283±0.236 | 0.920±0.052 | 0.883±0.057 | 0.971±0.009 | 0.755±0.204 | 0.834±0.118 | 0.834±0.118 |
| Rastrigin | 10 | 0.463±0.123 | 0.536±0.157 | 0.595±0.066 | 0.558±0.091 | 0.573±0.122 | 0.590±0.087 | 0.420±0.075 | 0.522±0.081 | 0.311±0.138 | 0.456±0.101 | 1.000±0.000 | 0.431±0.099 | 0.359±0.135 | 0.210±0.158 | 0.210±0.158 |
| Rosenbrock | 10 | 0.994±0.005 | 0.991±0.007 | 0.904±0.068 | 0.991±0.007 | 0.986±0.010 | 0.975±0.028 | 0.966±0.033 | 0.971±0.024 | 0.645±0.255 | 0.984±0.005 | 0.870±0.009 | 0.993±0.005 | 0.974±0.021 | 0.979±0.015 | 0.979±0.015 |
| Styblinski-Tang | 10 | 0.837±0.072 | 0.835±0.068 | 0.289±0.155 | 0.822±0.050 | 0.638±0.107 | 0.229±0.156 | 0.309±0.190 | 0.492±0.167 | 0.049±0.104 | 0.740±0.147 | 0.430±0.198 | 0.852±0.071 | 0.056±0.080 | 0.255±0.193 | 0.255±0.193 |
| Ackley | 20 | 0.827±0.045 | 0.851±0.032 | 0.777±0.067 | 0.818±0.023 | 0.781±0.035 | 0.404±0.100 | 0.741±0.027 | 0.753±0.030 | 0.474±0.148 | 0.731±0.048 | 1.000±0.000 | 0.755±0.050 | 0.252±0.106 | 0.299±0.110 | 0.299±0.110 |
| Levy | 20 | 0.949±0.028 | 0.943±0.017 | 0.889±0.073 | 0.945±0.025 | 0.904±0.041 | 0.907±0.056 | 0.926±0.031 | 0.900±0.045 | 0.819±0.075 | 0.934±0.025 | 0.977±0.004 | 0.955±0.024 | 0.879±0.097 | 0.746±0.157 | 0.746±0.157 |
| Powell | 20 | 0.955±0.028 | 0.965±0.017 | 0.872±0.085 | 0.939±0.020 | 0.913±0.077 | 0.915±0.061 | 0.948±0.031 | 0.913±0.058 | 0.845±0.104 | 0.936±0.040 | 0.966±0.016 | 0.967±0.020 | 0.912±0.061 | 0.876±0.094 | 0.876±0.094 |
| Rastrigin | 20 | 0.399±0.083 | 0.473±0.064 | 0.508±0.043 | 0.484±0.089 | 0.472±0.088 | 0.522±0.074 | 0.423±0.081 | 0.480±0.061 | 0.401±0.073 | 0.456±0.070 | 1.000±0.000 | 0.447±0.084 | 0.413±0.047 | 0.397±0.098 | 0.397±0.098 |
| Rosenbrock | 20 | 0.993±0.003 | 0.995±0.003 | 0.907±0.069 | 0.992±0.005 | 0.983±0.013 | 0.933±0.047 | 0.973±0.014 | 0.982±0.008 | 0.924±0.049 | 0.987±0.007 | 0.946±0.021 | 0.995±0.002 | 0.953±0.044 | 0.980±0.016 | 0.980±0.016 |
| Styblinski-Tang | 20 | 0.737±0.065 | 0.689±0.114 | 0.330±0.165 | 0.667±0.099 | 0.394±0.090 | 0.274±0.055 | 0.203±0.105 | 0.561±0.143 | 0.034±0.053 | 0.621±0.104 | 0.210±0.200 | 0.645±0.112 | 0.104±0.081 | 0.332±0.112 | 0.332±0.112 |
| Ackley | 50 | 0.235±0.275 | 0.342±0.276 | 0.823±0.045 | 0.623±0.264 | 0.594±0.218 | 0.465±0.103 | 0.638±0.041 | 0.759±0.015 | 0.730±0.021 | 0.745±0.042 | 1.000±0.000 | 0.546±0.140 | 0.273±0.067 | 0.179±0.121 | 0.179±0.121 |
| Levy | 50 | 0.940±0.082 | 0.965±0.020 | 0.943±0.018 | 0.971±0.019 | 0.958±0.016 | 0.879±0.025 | 0.948±0.016 | 0.951±0.035 | 0.941±0.015 | 0.952±0.012 | 0.987±0.001 | 0.955±0.016 | 0.880±0.023 | 0.901±0.095 | 0.901±0.095 |
| Powell | 50 | 0.954±0.029 | 0.982±0.008 | 0.950±0.023 | 0.975±0.008 | 0.969±0.010 | 0.938±0.024 | 0.970±0.004 | 0.961±0.015 | 0.980±0.007 | 0.965±0.014 | 0.986±0.003 | 0.957±0.030 | 0.955±0.014 | 0.939±0.027 | 0.939±0.027 |
| Rastrigin | 50 | 0.322±0.137 | 0.476±0.042 | 0.432±0.051 | 0.472±0.034 | 0.470±0.021 | 0.439±0.024 | 0.431±0.039 | 0.397±0.050 | 0.481±0.041 | 0.468±0.036 | 1.000±0.000 | 0.409±0.068 | 0.447±0.045 | 0.305±0.098 | 0.305±0.098 |
| Rosenbrock | 50 | 0.971±0.016 | 0.984±0.002 | 0.983±0.010 | 0.976±0.008 | 0.981±0.005 | 0.962±0.019 | 0.968±0.005 | 0.986±0.003 | 0.981±0.012 | 0.979±0.005 | 0.977±0.003 | 0.984±0.011 | 0.973±0.013 | 0.977±0.008 | 0.977±0.008 |
| Styblinski-Tang | 50 | 0.584±0.087 | 0.675±0.062 | 0.393±0.161 | 0.509±0.064 | 0.342±0.090 | 0.325±0.060 | 0.312±0.105 | 0.694±0.039 | 0.356±0.079 | 0.632±0.129 | 0.236±0.205 | 0.699±0.044 | 0.203±0.039 | 0.506±0.084 | 0.506±0.084 |
| Ackley | 100 | 0.277±0.343 | 0.190±0.344 | 0.863±0.028 | 0.417±0.361 | 0.540±0.281 | 0.682±0.093 | 0.708±0.015 | 0.645±0.102 | 0.844±0.016 | 0.742±0.074 | 0.007±0.004 | 0.299±0.120 | 0.244±0.231 | 0.348±0.246 | 0.348±0.246 |
| Emb. Hartmann 6 | 100 | 0.951±0.090 | 0.987±0.008 | 0.928±0.076 | 0.957±0.079 | 0.936±0.068 | 0.907±0.057 | 0.896±0.076 | 0.913±0.064 | 0.916±0.122 | 0.870±0.153 | 0.446±0.294 | 0.912±0.118 | 0.633±0.163 | 0.636±0.169 | 0.636±0.169 |
| Levy | 100 | 0.837±0.155 | 0.961±0.024 | 0.944±0.016 | 0.966±0.013 | 0.950±0.017 | 0.940±0.019 | 0.950±0.008 | 0.934±0.035 | 0.964±0.009 | 0.952±0.021 | 0.183±0.290 | 0.903±0.053 | 0.763±0.283 | 0.913±0.182 | 0.913±0.182 |
| Powell | 100 | 0.810±0.036 | 0.952±0.066 | 0.983±0.008 | 0.985±0.002 | 0.980±0.006 | 0.979±0.006 | 0.982±0.005 | 0.981±0.009 | 0.984±0.005 | 0.971±0.015 | 0.397±0.332 | 0.964±0.022 | 0.691±0.115 | 0.685±0.123 | 0.685±0.123 |
| Rastrigin | 100 | 0.497±0.034 | 0.401±0.160 | 0.459±0.024 | 0.441±0.117 | 0.455±0.026 | 0.455±0.029 | 0.446±0.016 | 0.442±0.041 | 0.443±0.020 | 0.482±0.023 | 0.332±0.461 | 0.634±0.088 | 0.290±0.112 | 0.286±0.085 | 0.286±0.085 |
| Rosenbrock | 100 | 0.822±0.075 | 0.971±0.012 | 0.980±0.009 | 0.953±0.084 | 0.976±0.009 | 0.970±0.011 | 0.972±0.006 | 0.969±0.016 | 0.978±0.013 | 0.974±0.003 | 0.290±0.377 | 0.936±0.052 | 0.966±0.075 | 0.931±0.114 | 0.931±0.114 |
| Styblinski-Tang | 100 | 0.537±0.045 | 0.474±0.110 | 0.401±0.106 | 0.423±0.045 | 0.353±0.055 | 0.296±0.030 | 0.308±0.042 | 0.532±0.109 | 0.278±0.041 | 0.536±0.077 | 0.205±0.248 | 0.627±0.058 | 0.222±0.053 | 0.221±0.054 | 0.221±0.054 |
| **Mean** | | 0.795 | 0.828 | 0.788 | 0.811 | 0.790 | 0.744 | 0.758 | 0.801 | 0.678 | 0.819 | 0.734 | 0.813 | 0.631 | 0.667 | 0.727 |
| **Median** | | 0.940 | 0.961 | 0.889 | 0.951 | 0.949 | 0.907 | 0.920 | 0.913 | 0.819 | 0.931 | 0.966 | 0.955 | 0.755 | 0.834 | 0.819 |

Table A2: BO on noise-free synthetic test problems. The relative batch instantaneous regret of the last, exploitative batch is shown. Colors are normalized row-wise. The BEEBO and $q$-UCB columns are equivalent to Table 3. Lower means better. Results are means over five replicate runs.

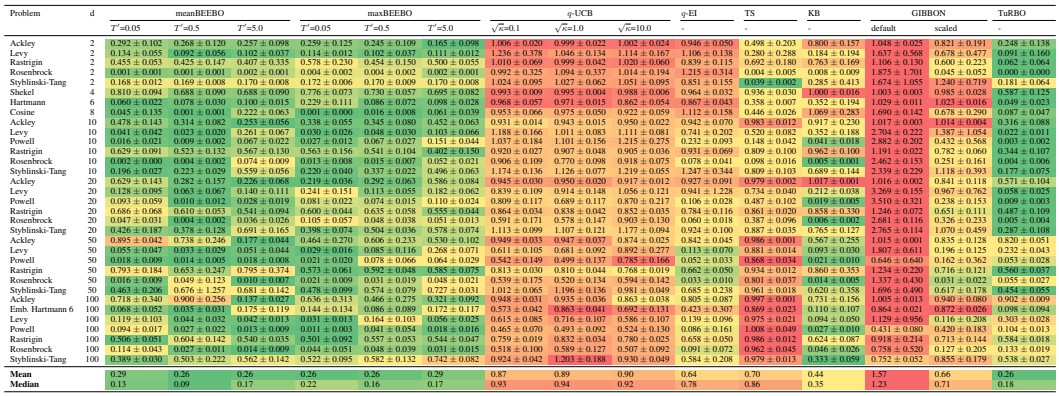

Table A3: Paired t-test p-values for the results of meanBEEBO in Table 2. The combined p-value was computed using Fisher's method. P-values smaller than 0.05 are indicated in bold.

Table A4: Paired t-test p-values for the results of maxBEEBO in Table 2. The combined p-value was computed using Fisher's method. P-values smaller than 0.05 are indicated in bold.

## D.2 Results for batch sizes 5 and 10

Table A5: BO on noise-free synthetic test problems. The normalized highest observed value after 10 rounds of BO with q=5 is shown. Colors are normalized row-wise. Higher means better. Results are means over ten replicate runs.

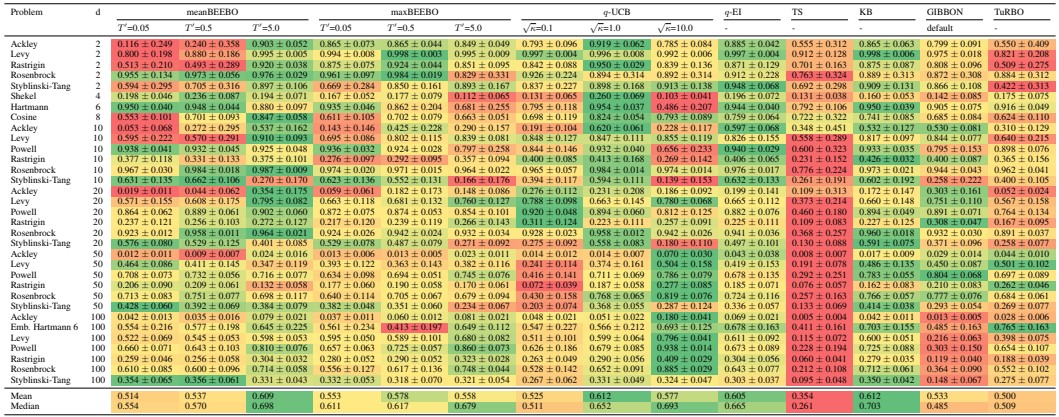

Table A6: BO on noise-free synthetic test problems. The normalized highest observed value after 10 rounds of BO with q=10 is shown. Colors are normalized row-wise. Higher means better. Results are means over ten replicate runs.

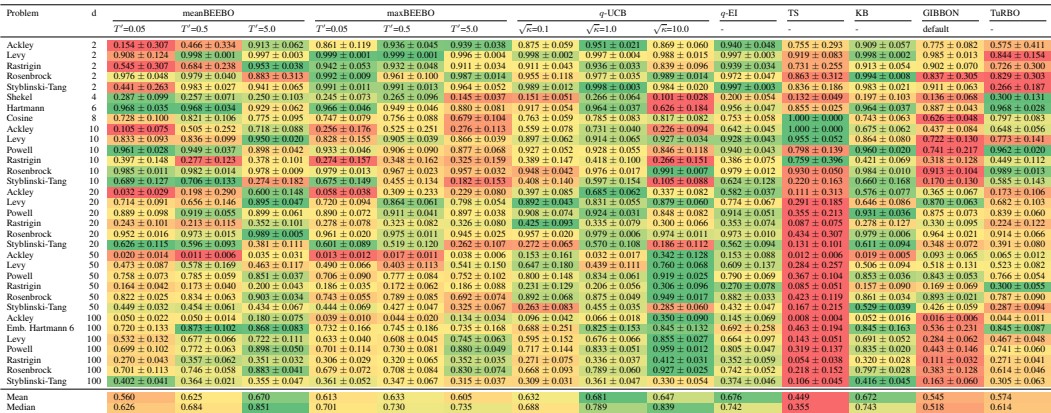

Table A7: BO on noise-free synthetic test problems. The relative batch instantaneous regret of the last, exploitative batch with $q$=5 is shown. Colors are normalized row-wise. Lower means better. Results are means over ten replicate runs.

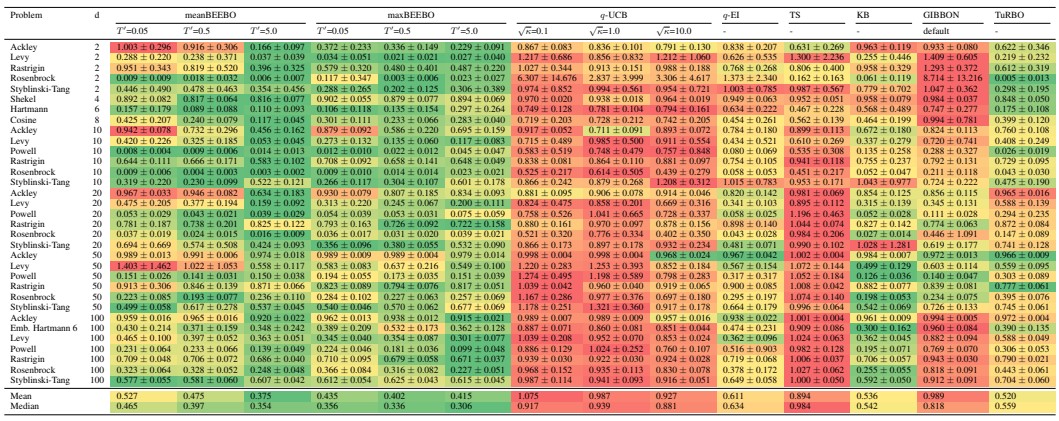

| Problem | d | meanBEEBO $T'$=0.05 | $T'$=0.5 | $T'$=5.0 | maxBEEBO $T'$=0.05 | $T'$=0.5 | $T'$=5.0 | $q$-UCB $\sqrt{\kappa}$=0.1 | $\sqrt{\kappa}$=1.0 | $\sqrt{\kappa}$=10.0 | $q$-EI | TS | KB | GIBBON | TuRBO |
|---|---|---|---|---|---|---|---|---|---|---|---|---|---|---|---|
| Ackley | 2 | 1.003±0.296 | 0.916±0.306 | 0.166±0.097 | 0.372±0.233 | 0.336±0.149 | 0.229±0.091 | 0.867±0.083 | 0.836±0.101 | 0.791±0.130 | 0.838±0.207 | 0.631±0.269 | 0.963±0.119 | 0.933±0.080 | 0.622±0.346 |
| Levy | 2 | 0.288±0.227 | 0.238±0.371 | 0.037±0.039 | 0.034±0.051 | 0.021±0.021 | 0.027±0.040 | 1.217±0.686 | 0.856±0.832 | 1.212±1.060 | 0.626±0.535 | 1.300±2.236 | 0.255±0.446 | 1.409±0.605 | 0.219±0.232 |
| Rastrigin | 2 | 0.951±0.343 | 0.819±0.520 | 0.396±0.325 | 0.579±0.320 | 0.480±0.401 | 0.487±0.220 | 1.027±0.344 | 0.913±0.151 | 0.988±0.188 | 0.768±0.268 | 0.806±0.400 | 0.958±0.329 | 1.293±0.372 | 0.612±0.319 |
| Rosenbrock | 2 | 0.009±0.009 | 0.018±0.032 | 0.006±0.007 | 0.117±0.347 | 0.003±0.006 | 0.023±0.027 | 6.307±14.676 | 2.837±3.999 | 3.306±4.617 | 1.373±2.340 | 0.162±0.163 | 0.061±0.119 | 8.714±13.216 | 0.005±0.013 |
| Styblinski-Tang | 2 | 0.446±0.490 | 0.478±0.463 | 0.354±0.456 | 0.288±0.265 | 0.202±0.125 | 0.306±0.389 | 0.974±0.852 | 0.994±0.561 | 0.954±0.721 | 1.003±0.785 | 0.987±0.567 | 0.779±0.702 | 1.047±0.362 | 0.298±0.195 |
| Shekel | 4 | 0.892±0.082 | 0.817±0.064 | 0.816±0.077 | 0.902±0.055 | 0.879±0.077 | 0.894±0.069 | 0.970±0.020 | 0.938±0.018 | 0.964±0.019 | 0.949±0.063 | 0.952±0.051 | 0.958±0.079 | 0.984±0.037 | 0.848±0.050 |
| Hartmann | 6 | 0.157±0.179 | 0.089±0.088 | 0.110±0.093 | 0.106±0.118 | 0.135±0.154 | 0.297±0.264 | 0.749±0.128 | 0.781±0.104 | 0.794±0.161 | 0.634±0.222 | 0.467±0.228 | 0.568±0.489 | 0.747±0.277 | 0.175±0.108 |
| Cosine | 8 | 0.425±0.207 | 0.240±0.079 | 0.117±0.045 | 0.301±0.111 | 0.233±0.066 | 0.283±0.040 | 0.719±0.203 | 0.728±0.212 | 0.742±0.205 | 0.454±0.261 | 0.562±0.139 | 0.464±0.199 | 0.994±0.781 | 0.399±0.120 |
| Ackley | 10 | 0.942±0.078 | 0.732±0.296 | 0.456±0.162 | 0.879±0.092 | 0.586±0.220 | 0.695±0.159 | 0.917±0.052 | 0.711±0.091 | 0.893±0.072 | 0.784±0.180 | 0.899±0.113 | 0.672±0.180 | 0.824±0.113 | 0.760±0.108 |
| Levy | 10 | 0.420±0.226 | 0.325±0.185 | 0.053±0.045 | 0.273±0.132 | 0.135±0.060 | 0.117±0.083 | 0.715±0.489 | 0.985±0.500 | 0.911±0.554 | 0.434±0.521 | 0.610±0.269 | 0.337±0.279 | 0.720±0.741 | 0.408±0.249 |
| Powell | 10 | 0.008±0.004 | 0.009±0.006 | 0.014±0.013 | 0.012±0.010 | 0.022±0.012 | 0.045±0.047 | 0.583±0.519 | 0.748±0.479 | 0.757±0.848 | 0.080±0.069 | 0.535±0.308 | 0.135±0.258 | 0.288±0.327 | 0.026±0.019 |
| Rastrigin | 10 | 0.644±0.111 | 0.666±0.171 | 0.583±0.102 | 0.708±0.092 | 0.658±0.141 | 0.648±0.049 | 0.838±0.081 | 0.864±0.110 | 0.881±0.097 | 0.754±0.105 | 0.941±0.118 | 0.755±0.237 | 0.792±0.131 | 0.729±0.095 |
| Rosenbrock | 10 | 0.009±0.006 | 0.004±0.003 | 0.003±0.002 | 0.009±0.010 | 0.014±0.014 | 0.023±0.021 | 0.525±0.217 | 0.614±0.505 | 0.439±0.279 | 0.058±0.053 | 0.451±0.217 | 0.052±0.047 | 0.211±0.118 | 0.043±0.030 |
| Styblinski-Tang | 10 | 0.319±0.220 | 0.230±0.099 | 0.522±0.121 | 0.266±0.117 | 0.304±0.107 | 0.601±0.178 | 0.866±0.242 | 0.879±0.268 | 1.208±0.312 | 1.015±0.783 | 0.953±0.171 | 1.043±0.977 | 0.724±0.222 | 0.475±0.190 |
| Ackley | 20 | 0.967±0.033 | 0.946±0.082 | 0.634±0.183 | 0.930±0.079 | 0.807±0.185 | 0.834±0.093 | 0.881±0.095 | 0.906±0.078 | 0.914±0.046 | 0.820±0.142 | 0.981±0.069 | 0.854±0.125 | 0.856±0.115 | 0.965±0.016 |
| Levy | 20 | 0.475±0.205 | 0.377±0.194 | 0.159±0.092 | 0.313±0.220 | 0.245±0.067 | 0.200±0.111 | 0.824±0.475 | 0.858±0.201 | 0.669±0.316 | 0.341±0.103 | 0.895±0.112 | 0.315±0.139 | 0.345±0.131 | 0.584±0.139 |
| Powell | 20 | 0.053±0.029 | 0.043±0.021 | 0.039±0.029 | 0.054±0.039 | 0.053±0.031 | 0.075±0.059 | 0.758±0.526 | 1.041±0.665 | 0.728±0.337 | 0.058±0.025 | 1.196±0.463 | 0.052±0.028 | 0.111±0.028 | 0.294±0.235 |
| Rastrigin | 20 | 0.781±0.187 | 0.738±0.201 | 0.825±0.122 | 0.793±0.163 | 0.726±0.092 | 0.722±0.158 | 0.880±0.161 | 0.970±0.097 | 0.878±0.156 | 0.898±0.140 | 1.044±0.074 | 0.827±0.142 | 0.774±0.063 | 0.872±0.084 |
| Rosenbrock | 20 | 0.037±0.019 | 0.024±0.015 | 0.016±0.009 | 0.036±0.017 | 0.031±0.020 | 0.039±0.021 | 0.521±0.320 | 0.776±0.334 | 0.402±0.350 | 0.043±0.028 | 0.984±0.206 | 0.027±0.014 | 0.446±1.091 | 0.147±0.089 |
| Styblinski-Tang | 20 | 0.694±0.669 | 0.574±0.508 | 0.424±0.093 | 0.356±0.096 | 0.380±0.055 | 0.532±0.090 | 0.866±0.173 | 0.897±0.178 | 0.932±0.234 | 0.481±0.071 | 0.990±0.102 | 1.028±1.281 | 0.619±0.177 | 0.741±0.128 |
| Ackley | 50 | 0.989±0.013 | 0.991±0.006 | 0.974±0.018 | 0.989±0.009 | 0.989±0.004 | 0.979±0.014 | 0.998±0.004 | 0.998±0.004 | 0.968±0.024 | 0.967±0.042 | 1.002±0.004 | 0.984±0.007 | 0.972±0.013 | 0.966±0.009 |
| Levy | 50 | 1.403±1.462 | 1.022±1.053 | 0.558±0.117 | 0.583±0.083 | 0.637±0.216 | 0.549±0.100 | 1.220±0.283 | 1.253±0.393 | 0.852±0.184 | 0.567±0.154 | 1.072±0.144 | 0.499±0.129 | 0.603±0.114 | 0.559±0.095 |
| Powell | 50 | 0.151±0.026 | 0.141±0.031 | 0.150±0.038 | 0.194±0.055 | 0.173±0.035 | 0.151±0.039 | 1.274±0.495 | 1.198±0.589 | 0.794±0.283 | 0.317±0.317 | 1.052±0.184 | 0.126±0.036 | 0.140±0.047 | 0.303±0.089 |
| Rastrigin | 50 | 0.913±0.306 | 0.846±0.139 | 0.871±0.066 | 0.823±0.089 | 0.794±0.076 | 0.817±0.051 | 1.039±0.042 | 0.960±0.040 | 0.919±0.065 | 0.900±0.085 | 1.008±0.042 | 0.882±0.077 | 0.339±0.081 | 0.777±0.061 |
| Rosenbrock | 50 | 0.223±0.085 | 0.193±0.077 | 0.236±0.110 | 0.284±0.102 | 0.227±0.063 | 0.257±0.069 | 1.167±0.286 | 0.977±0.376 | 0.697±0.180 | 0.295±0.197 | 1.074±0.140 | 0.198±0.053 | 0.234±0.075 | 0.395±0.076 |
| Styblinski-Tang | 50 | 0.499±0.058 | 0.617±0.278 | 0.537±0.045 | 0.540±0.046 | 0.570±0.062 | 0.677±0.069 | 1.178±0.251 | 1.321±0.360 | 0.917±0.178 | 0.664±0.179 | 0.996±0.064 | 0.542±0.069 | 0.726±0.133 | 0.745±0.061 |
| Ackley | 100 | 0.959±0.016 | 0.965±0.016 | 0.920±0.022 | 0.962±0.013 | 0.938±0.012 | 0.915±0.021 | 0.989±0.007 | 0.989±0.009 | 0.957±0.016 | 0.938±0.022 | 1.001±0.004 | 0.961±0.009 | 0.994±0.005 | 0.972±0.004 |
| Emb. Hartmann 6 | 100 | 0.430±0.214 | 0.371±0.159 | 0.583±0.242 | 0.389±0.229 | 0.532±0.173 | 0.362±0.128 | 0.887±0.071 | 0.864±0.081 | 0.851±0.044 | 0.474±0.231 | 0.909±0.086 | 0.300±0.162 | 0.960±0.084 | 0.390±0.135 |
| Levy | 100 | 0.465±0.100 | 0.397±0.052 | 0.363±0.051 | 0.345±0.040 | 0.354±0.087 | 0.301±0.077 | 1.039±0.208 | 0.952±0.070 | 0.853±0.024 | 0.362±0.096 | 1.024±0.063 | 0.362±0.045 | 0.882±0.094 | 0.588±0.049 |
| Powell | 100 | 0.231±0.064 | 0.233±0.066 | 0.139±0.049 | 0.224±0.046 | 0.181±0.036 | 0.099±0.048 | 0.886±0.129 | 1.024±0.252 | 0.760±0.107 | 0.516±0.903 | 0.982±0.128 | 0.195±0.071 | 0.769±0.070 | 0.306±0.053 |
| Rastrigin | 100 | 0.709±0.048 | 0.706±0.072 | 0.686±0.040 | 0.710±0.095 | 0.679±0.058 | 0.611±0.037 | 0.939±0.030 | 0.922±0.030 | 0.924±0.028 | 0.719±0.068 | 1.006±0.037 | 0.706±0.057 | 0.943±0.030 | 0.790±0.021 |
| Rosenbrock | 100 | 0.323±0.064 | 0.328±0.052 | 0.248±0.048 | 0.366±0.084 | 0.316±0.082 | 0.227±0.051 | 0.968±0.152 | 0.935±0.113 | 0.830±0.078 | 0.378±0.172 | 1.027±0.062 | 0.255±0.055 | 0.318±0.091 | 0.443±0.061 |
| Styblinski-Tang | 100 | 0.577±0.055 | 0.581±0.060 | 0.607±0.042 | 0.612±0.054 | 0.625±0.043 | 0.615±0.045 | 0.987±0.114 | 0.941±0.093 | 0.916±0.051 | 0.649±0.058 | 1.000±0.050 | 0.592±0.050 | 0.912±0.091 | 0.704±0.050 |
| Mean | | 0.527 | 0.475 | 0.375 | 0.435 | 0.402 | 0.415 | 1.075 | 0.987 | 0.927 | 0.611 | 0.894 | 0.536 | 0.989 | 0.520 |
| Median | | 0.465 | 0.397 | 0.354 | 0.356 | 0.336 | 0.306 | 0.917 | 0.939 | 0.881 | 0.634 | 0.984 | 0.542 | 0.818 | 0.559 |

Table A8: BO on noise-free synthetic test problems. The relative batch instantaneous regret of the last, exploitative batch with $q$=10 is shown. Colors are normalized row-wise. Lower means better. Results are means over ten replicate runs.

| Problem | d | meanBEEBO $T'$=0.05 | $T'$=0.5 | $T'$=5.0 | maxBEEBO $T'$=0.05 | $T'$=0.5 | $T'$=5.0 | $q$-UCB $\sqrt{\kappa}$=0.1 | $\sqrt{\kappa}$=1.0 | $\sqrt{\kappa}$=10.0 | $q$-EI | TS | KB | GIBBON | TuRBO |
|---|---|---|---|---|---|---|---|---|---|---|---|---|---|---|---|
| Ackley | 2 | 0.935±0.250 | 0.582±0.306 | 0.213±0.092 | 0.333±0.167 | 0.221±0.113 | 0.181±0.093 | 0.907±0.036 | 0.887±0.061 | 0.885±0.033 | 0.749±0.093 | 0.622±0.333 | 0.846±0.240 | 0.958±0.029 | 0.530±0.402 |
| Levy | 2 | 0.157±0.099 | 0.060±0.075 | 0.063±0.052 | 0.033±0.027 | 0.087±0.039 | 0.077±0.067 | 1.166±0.735 | 1.137±1.309 | 1.049±0.691 | 0.638±0.350 | 0.407±0.453 | 0.073±0.113 | 1.996±1.164 | 0.138±0.148 |
| Rastrigin | 2 | 0.631±0.269 | 0.676±0.294 | 0.404±0.209 | 0.566±0.189 | 0.482±0.399 | 0.662±0.296 | 0.994±0.255 | 0.927±0.213 | 1.016±0.234 | 0.608±0.109 | 0.831±0.427 | 0.708±0.121 | 1.217±0.214 | 0.292±0.299 |
| Rosenbrock | 2 | 0.002±0.002 | 0.002±0.006 | 0.005±0.004 | 0.003±0.003 | 0.002±0.002 | 0.007±0.012 | 1.054±0.835 | 1.335±1.611 | 2.055±4.821 | 0.417±0.644 | 0.116±0.228 | 0.009±0.011 | 2.095±3.809 | 0.001±0.001 |
| Styblinski-Tang | 2 | 0.233±0.080 | 0.124±0.071 | 0.371±0.396 | 0.190±0.181 | 0.290±0.353 | 0.234±0.259 | 1.129±0.298 | 1.104±0.304 | 0.861±0.295 | 0.930±0.436 | 0.359±0.312 | 0.294±0.400 | 1.273±0.469 | 0.261±0.105 |
| Shekel | 4 | 0.825±0.067 | 0.827±0.039 | 0.783±0.095 | 0.862±0.050 | 0.827±0.078 | 0.871±0.039 | 0.934±0.014 | 0.965±0.015 | 0.978±0.012 | 0.936±0.031 | 0.968±0.034 | 1.007±0.013 | 0.990±0.012 | 0.720±0.138 |
| Hartmann | 6 | 0.189±0.110 | 0.118±0.070 | 0.106±0.096 | 0.059±0.080 | 0.061±0.064 | 0.126±0.098 | 0.861±0.097 | 0.853±0.079 | 0.775±0.092 | 0.834±0.169 | 0.347±0.157 | 0.748±0.293 | 0.876±0.085 | 0.039±0.027 |
| Cosine | 8 | 0.217±0.084 | 0.159±0.100 | 0.167±0.073 | 0.213±0.066 | 0.188±0.064 | 0.270±0.092 | 0.831±0.118 | 0.917±0.166 | 0.797±0.215 | 1.055±0.570 | 0.570±0.194 | 0.793±0.664 | 1.003±0.159 | 0.200±0.062 |
| Ackley | 10 | 0.892±0.082 | 0.548±0.223 | 0.277±0.090 | 0.734±0.162 | 0.531±0.226 | 0.707±0.123 | 0.833±0.061 | 0.842±0.063 | 0.804±0.264 | 0.562±0.210 | 0.859±0.026 | 0.965±0.095 | 0.972±0.018 | 0.445±0.082 |
| Levy | 10 | 0.196±0.167 | 0.143±0.123 | 0.024±0.010 | 0.123±0.098 | 0.086±0.055 | 0.118±0.064 | 0.936±0.297 | 0.845±0.284 | 0.866±0.284 | 0.504±0.386 | 0.591±0.168 | 0.305±0.230 | 0.747±0.286 | 0.139±0.060 |
| Powell | 10 | 0.012±0.006 | 0.017±0.015 | 0.018±0.017 | 0.027±0.026 | 0.032±0.017 | 0.056±0.033 | 0.885±0.507 | 0.948±0.604 | 0.985±0.461 | 0.213±0.204 | 0.682±0.268 | 0.322±0.709 | 0.442±0.327 | 0.009±0.007 |
| Rastrigin | 10 | 0.773±0.187 | 0.775±0.116 | 0.615±0.122 | 0.714±0.177 | 0.593±0.117 | 0.598±0.169 | 0.869±0.097 | 0.909±0.084 | 0.856±0.128 | 0.719±0.042 | 0.828±0.090 | 0.738±0.149 | 0.791±0.067 | 0.423±0.077 |
| Rosenbrock | 10 | 0.008±0.008 | 0.006±0.005 | 0.004±0.002 | 0.011±0.008 | 0.013±0.007 | 0.034±0.017 | 0.550±0.287 | 0.684±0.259 | 0.744±0.178 | 0.155±0.115 | 0.303±0.136 | 0.087±0.072 | 0.633±0.537 | 0.005±0.006 |
| Styblinski-Tang | 10 | 0.276±0.155 | 0.389±0.329 | 0.523±0.135 | 0.257±0.086 | 0.414±0.177 | 0.545±0.070 | 1.006±0.346 | 1.079±0.255 | 1.074±0.198 | 1.060±0.498 | 0.941±0.118 | 0.907±0.125 | 0.926±0.186 | 0.256±0.084 |
| Ackley | 20 | 0.957±0.040 | 0.858±0.204 | 0.392±0.146 | 0.923±0.048 | 0.674±0.233 | 0.757±0.071 | 0.850±0.045 | 0.822±0.008 | 0.899±0.044 | 0.778±0.141 | 0.984±0.034 | 0.492±0.078 | 0.978±0.012 | 0.871±0.062 |
| Levy | 20 | 0.232±0.066 | 0.685±0.872 | 0.065±0.033 | 0.262±0.209 | 0.099±0.039 | 0.135±0.034 | 0.683±0.291 | 0.970±0.274 | 0.794±0.390 | 0.348±0.134 | 0.950±0.119 | 0.425±0.183 | 0.431±0.394 | 0.327±0.102 |
| Powell | 20 | 0.078±0.144 | 0.025±0.017 | 0.029±0.022 | 0.038±0.027 | 0.030±0.012 | 0.036±0.011 | 0.568±0.241 | 0.748±0.472 | 0.603±0.397 | 0.082±0.039 | 0.937±0.209 | 0.035±0.014 | 0.126±0.054 | 0.098±0.057 |
| Rastrigin | 20 | 0.739±0.102 | 0.730±0.076 | 0.707±0.074 | 0.732±0.101 | 0.660±0.102 | 0.660±0.099 | 0.843±0.119 | 0.865±0.055 | 0.824±0.059 | 0.712±0.035 | 0.999±0.053 | 0.747±0.093 | 0.724±0.061 | 0.783±0.081 |
| Rosenbrock | 20 | 0.019±0.007 | 0.013±0.008 | 0.004±0.002 | 0.018±0.009 | 0.011±0.005 | 0.031±0.030 | 0.461±0.335 | 0.784±0.288 | 0.410±0.200 | 0.036±0.008 | 0.788±0.119 | 0.018±0.007 | 0.095±0.039 | 0.074±0.045 |
| Styblinski-Tang | 20 | 0.268±0.074 | 0.288±0.069 | 0.445±0.058 | 0.311±0.082 | 0.410±0.117 | 0.679±0.152 | 1.104±0.212 | 1.104±0.296 | 0.628±0.224 | 0.985±0.078 | 0.852±1.063 | 0.859±0.657 | 0.523±0.131 | |
| Ackley | 50 | 0.981±0.018 | 0.990±0.005 | 0.988±0.015 | 0.987±0.009 | 0.961±0.011 | 0.960±0.046 | 0.994±0.007 | 0.994±0.001 | 0.862±0.058 | 0.893±0.085 | 1.001±0.005 | 0.982±0.005 | 0.934±0.064 | 0.949±0.010 |
| Levy | 50 | 0.464±0.115 | 0.965±1.096 | 0.448±0.107 | 0.482±0.119 | 0.489±0.120 | 0.417±0.127 | 1.023±0.305 | 1.373±0.357 | 0.611±0.174 | 0.516±0.309 | 1.032±0.194 | 1.034±1.514 | 0.554±0.194 | 0.487±0.055 |
| Powell | 50 | 0.127±0.030 | 0.109±0.018 | 0.076±0.032 | 0.147±0.031 | 0.113±0.037 | 0.177±0.076 | 1.177±0.598 | 1.302±0.455 | 0.522±0.332 | 0.269±0.418 | 0.972±0.192 | 0.082±0.031 | 0.398±0.913 | 0.218±0.062 |
| Rastrigin | 50 | 0.889±0.124 | 0.858±0.050 | 0.764±0.111 | 0.804±0.065 | 0.789±0.045 | 0.776±0.085 | 0.943±0.093 | 0.940±0.037 | 0.829±0.063 | 0.826±0.133 | 0.997±0.096 | 1.000±0.323 | 0.832±0.041 | 0.696±0.035 |
| Rosenbrock | 50 | 0.131±0.030 | 0.123±0.034 | 0.123±0.057 | 0.191±0.043 | 0.159±0.073 | 0.249±0.103 | 0.784±0.390 | 1.066±0.234 | 0.417±0.180 | 0.107±0.023 | 1.039±0.137 | 0.117±0.042 | 0.135±0.056 | 0.285±0.085 |
| Styblinski-Tang | 50 | 0.450±0.039 | 0.500±0.148 | 0.534±0.238 | 0.467±0.089 | 0.487±0.068 | 0.593±0.053 | 1.050±0.210 | 1.313±0.213 | 0.797±0.117 | 0.520±0.073 | 0.990±0.055 | 0.633±0.631 | 0.804±0.821 | 0.697±0.068 |
| Ackley | 100 | 0.950±0.023 | 0.948±0.016 | 0.815±0.075 | 0.961±0.010 | 0.954±0.019 | 0.861±0.033 | 0.968±0.022 | 0.992±0.010 | 0.976±0.022 | 0.905±0.057 | 0.999±0.002 | 0.949±0.015 | 0.992±0.002 | 0.959±0.007 |
| Emb. Hartmann 6 | 100 | 0.265±0.179 | 0.118±0.092 | 0.107±0.085 | 0.220±0.151 | 0.199±0.155 | 0.208±0.125 | 0.784±0.109 | 0.815±0.127 | 0.756±0.133 | 0.543±0.163 | 0.870±0.048 | 0.157±0.175 | 0.941±0.065 | 0.301±0.120 |
| Levy | 100 | 0.402±0.108 | 0.291±0.069 | 0.242±0.083 | 0.328±0.078 | 0.351±0.077 | 0.247±0.053 | 1.017±0.196 | 1.154±0.164 | 0.976±0.179 | 0.442±0.386 | 0.999±0.040 | 0.318±0.120 | 0.821±0.054 | 0.515±0.037 |
| Powell | 100 | 0.189±0.063 | 0.141±0.031 | 0.066±0.024 | 0.190±0.067 | 0.166±0.047 | 0.083±0.027 | 0.865±0.237 | 1.232±0.274 | 0.861±0.315 | 0.112±0.020 | 0.916±0.092 | 0.112±0.020 | 0.727±0.063 | 0.226±0.039 |
| Rastrigin | 100 | 0.726±0.068 | 0.702±0.077 | 0.628±0.042 | 0.709±0.084 | 0.666±0.085 | 0.636±0.046 | 0.946±0.057 | 0.946±0.051 | 0.806±0.090 | 0.788±0.119 | 1.002±0.027 | 0.730±0.058 | 0.927±0.024 | 0.715±0.037 |
| Rosenbrock | 100 | 0.227±0.078 | 0.200±0.059 | 0.093±0.040 | 0.255±0.080 | 0.229±0.080 | 0.153±0.080 | 0.880±0.167 | 1.060±0.106 | 0.823±0.223 | 0.225±0.047 | 0.988±0.070 | 0.164±0.016 | 0.766±0.057 | 0.366±0.047 |
| Styblinski-Tang | 100 | 0.527±0.052 | 0.568±0.036 | 0.598±0.054 | 0.566±0.044 | 0.581±0.036 | 0.625±0.029 | 1.051±0.145 | 1.267±0.286 | 0.949±0.017 | 0.667±0.271 | 0.987±0.039 | 0.923±0.020 | 0.898±0.033 | 0.677±0.041 |
| Mean | | 0.422 | 0.410 | 0.322 | 0.386 | 0.360 | 0.387 | 0.909 | 1.005 | 0.867 | 0.581 | 0.813 | 0.525 | 0.844 | 0.401 |
| Median | | 0.268 | 0.291 | 0.242 | 0.262 | 0.290 | 0.249 | 0.936 | 0.965 | 0.861 | 0.628 | 0.937 | 0.522 | 0.859 | 0.327 |

## D.3    Control problems

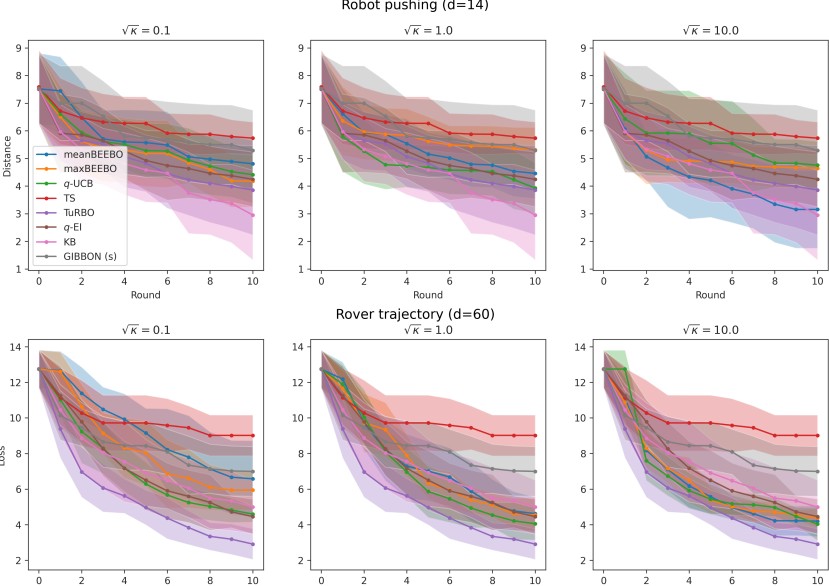

Figure A2: Experiments on the 14D robot arm pushing and 60D rover trajectory planning control problems. 10 replicates each. GIBBON (s) refers to the scaled larged-batch variant of GIBBON.

## D.4    Run time

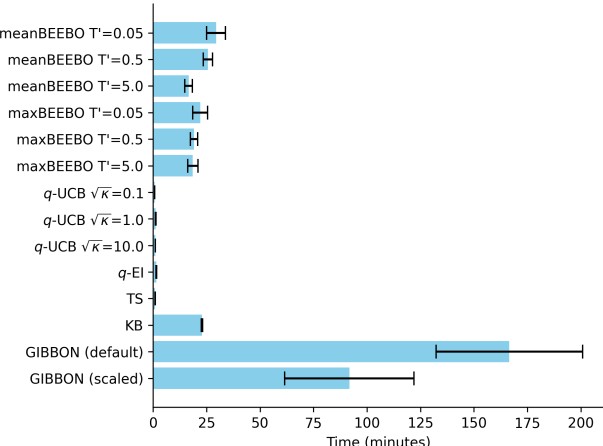

Figure A3: Example run times for the 10-round BO experiment on the 6D Hartmann problem with $Q$=100. Error bars are over 5 replicate runs. Run times vary depending on the test problem, with GIBBON appearing especially sensitive, becoming e.g. 10x slower on the 50D Ackley problem.

Table A9: Total run times for five replicates of the experiments presented in Table A1. We sum over all test problems.

| Method | Configuration | Total time [h] |
|---|---|---|
| meanBEEBO | $T' = 0.05$ | 66.12 |
| meanBEEBO | $T' = 0.5$ | 47.08 |
| meanBEEBO | $T' = 5.0$ | 37.13 |
| maxBEEBO | $T' = 0.05$ | 54.85 |
| maxBEEBO | $T' = 0.5$ | 44.20 |
| maxBEEBO | $T' = 5.0$ | 47.63 |
| $q$-UCB | $\sqrt{\kappa} = 0.1$ | 3.33 |
| $q$-UCB | $\sqrt{\kappa} = 1.0$ | 3.70 |
| $q$-UCB | $\sqrt{\kappa} = 10.0$ | 4.41 |
| $q$-EI | - | 24.33 |
| TS | - | 6.56 |
| KB | - | 223.78 |
| GIBBON | default | 3380.48 |
| GIBBON | scaled | 1055.93 |

## D.5 Results with random initialization in round 0

Table A10: BO with random initialization on noise-free synthetic test problems. The normalized highest observed value after 10 rounds of BO with $q$=100 is shown. Colors are normalized row-wise. Higher means better. Results are means over five replicate runs.

| Problem | d | meanBEEBO $T'$=0.05 | meanBEEBO $T'$=0.5 | meanBEEBO $T'$=5.0 | maxBEEBO $T'$=0.05 | maxBEEBO $T'$=0.5 | maxBEEBO $T'$=5.0 | $q$-UCB $\sqrt{\kappa}$=0.1 | $q$-UCB $\sqrt{\kappa}$=1.0 | $q$-UCB $\sqrt{\kappa}$=10.0 | $q$-EI | TS | KB | TuRBO |
|---|---|---|---|---|---|---|---|---|---|---|---|---|---|---|
| Ackley | 2 | 0.994 ± 0.011 | 0.971 ± 0.041 | 0.988 ± 0.006 | 0.985 ± 0.012 | 0.977 ± 0.029 | 0.912 ± 0.069 | 0.793 ± 0.255 | 0.904 ± 0.059 | 0.968 ± 0.019 | 0.957 ± 0.056 | 1.000 ± 0.000 | 0.928 ± 0.076 | 0.972 ± 0.027 |
| Levy | 2 | 0.995 ± 0.008 | 0.997 ± 0.002 | 0.995 ± 0.005 | 0.997 ± 0.002 | 0.998 ± 0.002 | 0.966 ± 0.044 | 0.983 ± 0.023 | 0.985 ± 0.010 | 0.954 ± 0.047 | 0.994 ± 0.004 | 0.993 ± 0.004 | 0.997 ± 0.003 | 0.992 ± 0.017 |
| Rastrigin | 2 | 0.838 ± 0.190 | 0.674 ± 0.433 | 0.836 ± 0.138 | 0.784 ± 0.350 | 0.756 ± 0.356 | 0.847 ± 0.199 | 0.540 ± 0.359 | 0.675 ± 0.250 | 0.296 ± 0.255 | 0.873 ± 0.173 | 0.800 ± 0.447 | 0.691 ± 0.299 | 0.902 ± 0.220 |
| Rosenbrock | 2 | 0.893 ± 0.080 | 0.900 ± 0.103 | 0.605 ± 0.395 | 0.525 ± 0.496 | 0.705 ± 0.413 | 0.242 ± 0.306 | 0.678 ± 0.394 | 0.475 ± 0.452 | 0.469 ± 0.485 | 0.728 ± 0.304 | 0.888 ± 0.249 | 0.753 ± 0.423 | 0.875 ± 0.266 |
| Styblinski-Tang | 2 | 1.000 ± 0.001 | 1.000 ± 0.000 | 1.000 ± 0.000 | 0.998 ± 0.002 | 0.999 ± 0.002 | 0.999 ± 0.002 | 0.998 ± 0.001 | 0.999 ± 0.002 | 0.993 ± 0.008 | 0.999 ± 0.001 | 0.998 ± 0.002 | 1.000 ± 0.000 | 1.000 ± 0.000 |
| Shekel | 4 | 0.839 ± 0.311 | 0.829 ± 0.360 | 0.706 ± 0.326 | 0.390 ± 0.353 | 0.476 ± 0.274 | 0.388 ± 0.316 | 0.183 ± 0.068 | 0.376 ± 0.308 | 0.266 ± 0.073 | 0.530 ± 0.350 | 0.087 ± 0.060 | 0.178 ± 0.071 | 0.824 ± 0.247 |
| Hartmann | 6 | 1.000 ± 0.000 | 1.000 ± 0.000 | 0.984 ± 0.016 | 0.955 ± 0.063 | 0.989 ± 0.024 | 0.992 ± 0.010 | 0.960 ± 0.052 | 0.998 ± 0.002 | 0.919 ± 0.057 | 0.991 ± 0.020 | 0.715 ± 0.150 | 0.998 ± 0.001 | 0.965 ± 0.045 |
| Cosine | 8 | 1.000 ± 0.000 | 0.997 ± 0.002 | 0.377 ± 0.144 | 0.999 ± 0.000 | 0.968 ± 0.023 | 0.871 ± 0.057 | 0.906 ± 0.079 | 0.903 ± 0.050 | 0.390 ± 0.242 | 0.787 ± 0.118 | 1.000 ± 0.000 | 0.987 ± 0.009 | 0.912 ± 0.066 |
| Ackley | 10 | 0.935 ± 0.026 | 0.904 ± 0.039 | 0.820 ± 0.037 | 0.816 ± 0.037 | 0.742 ± 0.052 | 0.507 ± 0.137 | 0.790 ± 0.015 | 0.776 ± 0.038 | 0.561 ± 0.170 | 0.782 ± 0.036 | 1.000 ± 0.000 | 0.789 ± 0.023 | 0.784 ± 0.026 |
| Levy | 10 | 0.981 ± 0.013 | 0.957 ± 0.028 | 0.930 ± 0.034 | 0.956 ± 0.026 | 0.926 ± 0.024 | 0.919 ± 0.022 | 0.866 ± 0.036 | 0.839 ± 0.068 | 0.813 ± 0.126 | 0.920 ± 0.064 | 0.942 ± 0.026 | 0.949 ± 0.041 | 0.941 ± 0.072 |
| Powell | 10 | 0.971 ± 0.029 | 0.953 ± 0.030 | 0.672 ± 0.387 | 0.923 ± 0.109 | 0.900 ± 0.103 | 0.702 ± 0.398 | 0.886 ± 0.061 | 0.822 ± 0.161 | 0.219 ± 0.302 | 0.851 ± 0.191 | 0.833 ± 0.169 | 0.939 ± 0.058 | 0.985 ± 0.008 |
| Rastrigin | 10 | 0.465 ± 0.144 | 0.495 ± 0.180 | 0.570 ± 0.091 | 0.526 ± 0.087 | 0.516 ± 0.140 | 0.642 ± 0.083 | 0.394 ± 0.143 | 0.564 ± 0.152 | 0.193 ± 0.172 | 0.441 ± 0.065 | 1.000 ± 0.000 | 0.394 ± 0.088 | 0.672 ± 0.165 |
| Rosenbrock | 10 | 0.992 ± 0.006 | 0.990 ± 0.002 | 0.865 ± 0.068 | 0.990 ± 0.003 | 0.986 ± 0.007 | 0.966 ± 0.037 | 0.965 ± 0.043 | 0.952 ± 0.019 | 0.220 ± 0.373 | 0.975 ± 0.017 | 0.820 ± 0.029 | 0.992 ± 0.003 | 0.993 ± 0.004 |
| Styblinski-Tang | 10 | 0.815 ± 0.051 | 0.814 ± 0.062 | 0.245 ± 0.050 | 0.784 ± 0.025 | 0.643 ± 0.124 | 0.217 ± 0.157 | 0.165 ± 0.015 | 0.584 ± 0.177 | 0.028 ± 0.002 | 0.619 ± 0.093 | 0.399 ± 0.163 | 0.827 ± 0.066 | 0.654 ± 0.085 |
| Robot Pushing | 14 | 0.350 ± 0.121 | 0.377 ± 0.124 | 0.560 ± 0.172 | 0.425 ± 0.107 | 0.310 ± 0.140 | 0.395 ± 0.131 | 0.424 ± 0.154 | 0.522 ± 0.170 | 0.379 ± 0.093 | 0.417 ± 0.160 | 0.247 ± 0.118 | 0.694 ± 0.255 | 0.518 ± 0.149 |
| Ackley | 20 | 0.843 ± 0.016 | 0.857 ± 0.028 | 0.789 ± 0.048 | 0.819 ± 0.030 | 0.788 ± 0.040 | 0.390 ± 0.106 | 0.706 ± 0.063 | 0.775 ± 0.039 | 0.460 ± 0.052 | 0.740 ± 0.054 | 1.000 ± 0.000 | 0.763 ± 0.048 | 0.438 ± 0.103 |
| Levy | 20 | 0.936 ± 0.035 | 0.939 ± 0.019 | 0.896 ± 0.044 | 0.953 ± 0.027 | 0.901 ± 0.046 | 0.911 ± 0.038 | 0.928 ± 0.016 | 0.929 ± 0.029 | 0.768 ± 0.048 | 0.921 ± 0.024 | 0.979 ± 0.003 | 0.956 ± 0.013 | 0.925 ± 0.041 |
| Powell | 20 | 0.947 ± 0.036 | 0.966 ± 0.013 | 0.840 ± 0.111 | 0.936 ± 0.019 | 0.880 ± 0.100 | 0.908 ± 0.076 | 0.946 ± 0.007 | 0.928 ± 0.050 | 0.819 ± 0.122 | 0.926 ± 0.047 | 0.964 ± 0.014 | 0.969 ± 0.016 | 0.957 ± 0.036 |
| Rastrigin | 20 | 0.373 ± 0.042 | 0.462 ± 0.049 | 0.518 ± 0.054 | 0.514 ± 0.059 | 0.480 ± 0.077 | 0.491 ± 0.087 | 0.450 ± 0.115 | 0.463 ± 0.069 | 0.383 ± 0.094 | 0.451 ± 0.053 | 1.000 ± 0.000 | 0.481 ± 0.068 | 0.523 ± 0.064 |
| Rosenbrock | 20 | 0.992 ± 0.004 | 0.993 ± 0.004 | 0.920 ± 0.041 | 0.991 ± 0.005 | 0.982 ± 0.013 | 0.923 ± 0.054 | 0.967 ± 0.018 | 0.984 ± 0.005 | 0.915 ± 0.044 | 0.984 ± 0.007 | 0.939 ± 0.020 | 0.994 ± 0.003 | 0.993 ± 0.002 |
| Styblinski-Tang | 20 | 0.706 ± 0.061 | 0.669 ± 0.113 | 0.305 ± 0.198 | 0.607 ± 0.086 | 0.417 ± 0.122 | 0.279 ± 0.077 | 0.204 ± 0.130 | 0.524 ± 0.184 | 0.054 ± 0.074 | 0.639 ± 0.082 | 0.271 ± 0.262 | 0.665 ± 0.129 | 0.604 ± 0.088 |
| Ackley | 50 | 0.221 ± 0.293 | 0.146 ± 0.116 | 0.842 ± 0.010 | 0.622 ± 0.243 | 0.705 ± 0.042 | 0.457 ± 0.098 | 0.627 ± 0.053 | 0.739 ± 0.034 | 0.736 ± 0.020 | 0.727 ± 0.051 | 1.000 ± 0.000 | 0.683 ± 0.122 | 0.175 ± 0.022 |
| Levy | 50 | 0.976 ± 0.010 | 0.978 ± 0.012 | 0.943 ± 0.021 | 0.977 ± 0.012 | 0.955 ± 0.018 | 0.867 ± 0.013 | 0.952 ± 0.016 | 0.966 ± 0.025 | 0.933 ± 0.007 | 0.943 ± 0.017 | 0.987 ± 0.002 | 0.926 ± 0.041 | 0.793 ± 0.054 |
| Powell | 50 | 0.940 ± 0.036 | 0.978 ± 0.010 | 0.959 ± 0.025 | 0.976 ± 0.009 | 0.970 ± 0.014 | 0.929 ± 0.024 | 0.965 ± 0.014 | 0.958 ± 0.016 | 0.978 ± 0.007 | 0.964 ± 0.013 | 0.985 ± 0.004 | 0.957 ± 0.022 | 0.920 ± 0.039 |
| Rastrigin | 50 | 0.273 ± 0.156 | 0.505 ± 0.031 | 0.453 ± 0.040 | 0.473 ± 0.042 | 0.466 ± 0.016 | 0.445 ± 0.027 | 0.466 ± 0.073 | 0.418 ± 0.023 | 0.503 ± 0.047 | 0.468 ± 0.012 | 1.000 ± 0.000 | 0.423 ± 0.085 | 0.459 ± 0.070 |
| Rosenbrock | 50 | 0.976 ± 0.012 | 0.985 ± 0.003 | 0.988 ± 0.005 | 0.978 ± 0.011 | 0.981 ± 0.005 | 0.968 ± 0.016 | 0.974 ± 0.004 | 0.987 ± 0.003 | 0.984 ± 0.013 | 0.981 ± 0.003 | 0.979 ± 0.003 | 0.987 ± 0.003 | 0.968 ± 0.018 |
| Styblinski-Tang | 50 | 0.605 ± 0.067 | 0.693 ± 0.038 | 0.417 ± 0.163 | 0.536 ± 0.072 | 0.415 ± 0.055 | 0.332 ± 0.037 | 0.341 ± 0.082 | 0.716 ± 0.031 | 0.371 ± 0.040 | 0.690 ± 0.069 | 0.254 ± 0.220 | 0.720 ± 0.041 | 0.499 ± 0.073 |
| Rover trajectory | 60 | 0.448 ± 0.209 | 0.678 ± 0.029 | 0.708 ± 0.060 | 0.533 ± 0.066 | 0.657 ± 0.104 | 0.629 ± 0.040 | 0.626 ± 0.076 | 0.613 ± 0.070 | 0.665 ± 0.079 | 0.635 ± 0.074 | 0.265 ± 0.069 | 0.616 ± 0.043 | 0.764 ± 0.074 |
| Ackley | 100 | 0.310 ± 0.408 | 0.347 ± 0.423 | 0.864 ± 0.023 | 0.526 ± 0.335 | 0.536 ± 0.293 | 0.707 ± 0.086 | 0.696 ± 0.015 | 0.721 ± 0.080 | 0.850 ± 0.007 | 0.747 ± 0.071 | 0.007 ± 0.005 | 0.289 ± 0.081 | 0.110 ± 0.014 |
| Emb. Hartmann 6 | 100 | 0.980 ± 0.009 | 0.988 ± 0.008 | 0.916 ± 0.101 | 0.982 ± 0.016 | 0.933 ± 0.057 | 0.914 ± 0.051 | 0.941 ± 0.035 | 0.915 ± 0.038 | 0.913 ± 0.116 | 0.922 ± 0.110 | 0.554 ± 0.315 | 0.949 ± 0.065 | 0.931 ± 0.084 |
| Levy | 100 | 0.890 ± 0.150 | 0.966 ± 0.024 | 0.943 ± 0.019 | 0.962 ± 0.012 | 0.942 ± 0.017 | 0.946 ± 0.013 | 0.952 ± 0.005 | 0.937 ± 0.029 | 0.964 ± 0.014 | 0.943 ± 0.030 | 0.310 ± 0.382 | 0.908 ± 0.056 | 0.692 ± 0.013 |
| Powell | 100 | 0.786 ± 0.051 | 0.929 ± 0.099 | 0.985 ± 0.004 | 0.985 ± 0.002 | 0.981 ± 0.009 | 0.981 ± 0.003 | 0.983 ± 0.004 | 0.978 ± 0.013 | 0.983 ± 0.008 | 0.963 ± 0.018 | 0.288 ± 0.165 | 0.967 ± 0.021 | 0.860 ± 0.027 |
| Rastrigin | 100 | 0.522 ± 0.027 | 0.367 ± 0.194 | 0.467 ± 0.027 | 0.481 ± 0.057 | 0.479 ± 0.007 | 0.469 ± 0.037 | 0.442 ± 0.021 | 0.442 ± 0.047 | 0.432 ± 0.020 | 0.493 ± 0.014 | 0.238 ± 0.426 | 0.674 ± 0.126 | 0.394 ± 0.022 |
| Rosenbrock | 100 | 0.810 ± 0.029 | 0.972 ± 0.012 | 0.976 ± 0.008 | 0.928 ± 0.119 | 0.978 ± 0.006 | 0.975 ± 0.008 | 0.977 ± 0.008 | 0.968 ± 0.012 | 0.985 ± 0.008 | 0.974 ± 0.003 | 0.270 ± 0.406 | 0.943 ± 0.058 | 0.857 ± 0.010 |
| Styblinski-Tang | 100 | 0.564 ± 0.034 | 0.470 ± 0.152 | 0.396 ± 0.079 | 0.432 ± 0.043 | 0.331 ± 0.061 | 0.309 ± 0.022 | 0.321 ± 0.027 | 0.542 ± 0.130 | 0.280 ± 0.017 | 0.591 ± 0.023 | 0.198 ± 0.273 | 0.593 ± 0.063 | 0.412 ± 0.034 |
| **Mean** | | 0.776 | 0.793 | 0.751 | 0.779 | 0.762 | 0.697 | 0.714 | 0.768 | 0.618 | 0.788 | 0.692 | 0.788 | 0.750 |
| **Median** | | 0.890 | 0.929 | 0.840 | 0.923 | 0.880 | 0.847 | 0.793 | 0.822 | 0.665 | 0.851 | 0.888 | 0.908 | 0.857 |

Table A11: BO with random initialization on noise-free synthetic test problems. The relative batch instantaneous regret of the last, exploitative batch is shown. Colors are normalized row-wise. Lower means better. Results are means over five replicate runs.

| Problem | d | meanBEEBO | | | maxBEEBO | | | q-UCB | | | q-EI | TS | KB | TuRBO |
|---|---|---|---|---|---|---|---|---|---|---|---|---|---|---|
| | | T'=0.05 | T'=0.5 | T'=5.0 | T'=0.05 | T'=0.5 | T'=5.0 | $\sqrt{\kappa}$=0.1 | $\sqrt{\kappa}$=1.0 | $\sqrt{\kappa}$=10.0 | - | - | - | - |
| Ackley | 2 | 0.268 ± 0.132 | 0.189 ± 0.049 | 0.334 ± 0.082 | 0.259 ± 0.187 | 0.221 ± 0.145 | 0.299 ± 0.151 | 1.011 ± 0.027 | 0.993 ± 0.016 | 0.994 ± 0.018 | 0.806 ± 0.209 | 0.624 ± 0.361 | 0.749 ± 0.266 | 0.145 ± 0.101 |
| Levy | 2 | 0.153 ± 0.024 | 0.130 ± 0.055 | 0.066 ± 0.059 | 0.111 ± 0.010 | 0.091 ± 0.034 | 0.109 ± 0.009 | 1.260 ± 0.454 | 1.195 ± 0.283 | 1.256 ± 0.263 | 1.219 ± 0.204 | 0.280 ± 0.401 | 0.088 ± 0.100 | 0.000 ± 0.000 |
| Rastrigin | 2 | 0.427 ± 0.019 | 0.600 ± 0.381 | 0.523 ± 0.228 | 0.306 ± 0.210 | 0.543 ± 0.073 | 0.491 ± 0.053 | 1.009 ± 0.061 | 0.991 ± 0.104 | 1.047 ± 0.058 | 0.808 ± 0.060 | 0.728 ± 0.196 | 0.851 ± 0.103 | 0.031 ± 0.063 |
| Rosenbrock | 2 | 0.001 ± 0.000 | 0.001 ± 0.000 | 0.002 ± 0.001 | 0.003 ± 0.002 | 0.002 ± 0.000 | 0.003 ± 0.003 | 0.895 ± 0.134 | 0.898 ± 0.131 | 0.917 ± 0.264 | 1.101 ± 0.303 | 0.002 ± 0.001 | 0.003 ± 0.004 | 0.000 ± 0.000 |
| Styblinski-Tang | 2 | 0.173 ± 0.007 | 0.170 ± 0.009 | 0.170 ± 0.008 | 0.169 ± 0.007 | 0.171 ± 0.008 | 0.170 ± 0.008 | 1.118 ± 0.087 | 1.046 ± 0.080 | 1.047 ± 0.097 | 0.751 ± 0.154 | 0.471 ± 0.591 | 0.169 ± 0.320 | 0.000 ± 0.000 |
| Shekel | 4 | 0.790 ± 0.049 | 0.635 ± 0.094 | 0.707 ± 0.047 | 0.757 ± 0.097 | 0.644 ± 0.229 | 0.727 ± 0.096 | 0.992 ± 0.006 | 0.989 ± 0.006 | 0.992 ± 0.004 | 0.959 ± 0.041 | 0.945 ± 0.033 | 1.001 ± 0.011 | 0.387 ± 0.223 |
| Hartmann | 6 | 0.052 ± 0.017 | 0.087 ± 0.030 | 0.096 ± 0.012 | 0.189 ± 0.119 | 0.085 ± 0.029 | 0.065 ± 0.029 | 0.959 ± 0.075 | 0.957 ± 0.017 | 0.851 ± 0.087 | 0.863 ± 0.067 | 0.356 ± 0.006 | 0.288 ± 0.171 | 0.028 ± 0.031 |
| Cosine | 8 | 0.060 ± 0.119 | 0.304 ± 0.006 | 0.304 ± 0.037 | 0.000 ± 0.000 | 0.015 ± 0.010 | 0.062 ± 0.033 | 0.987 ± 0.097 | 0.971 ± 0.071 | 0.966 ± 0.068 | 1.111 ± 0.214 | 0.436 ± 0.018 | 1.217 ± 0.099 | 0.380 ± 0.053 |
| Ackley | 10 | 0.447 ± 0.082 | 0.329 ± 0.074 | 0.250 ± 0.035 | 0.324 ± 0.037 | 0.321 ± 0.047 | 0.485 ± 0.075 | 0.936 ± 0.025 | 0.930 ± 0.021 | 0.949 ± 0.020 | 0.937 ± 0.038 | 0.983 ± 0.015 | 0.903 ± 0.260 | 0.299 ± 0.004 |
| Levy | 10 | 0.079 ± 0.068 | 0.025 ± 0.018 | 0.296 ± 0.067 | 0.037 ± 0.031 | 0.048 ± 0.032 | 0.093 ± 0.078 | 1.324 ± 0.110 | 0.979 ± 0.112 | 1.088 ± 0.175 | 0.737 ± 0.322 | 0.595 ± 0.100 | 0.552 ± 0.407 | 0.024 ± 0.011 |
| Powell | 10 | 0.019 ± 0.028 | 0.009 ± 0.001 | 0.051 ± 0.004 | 0.026 ± 0.012 | 0.052 ± 0.008 | 0.156 ± 0.056 | 1.045 ± 0.237 | 0.926 ± 0.175 | 1.248 ± 0.273 | 0.262 ± 0.173 | 0.144 ± 0.029 | 0.046 ± 0.038 | 0.003 ± 0.003 |
| Rastrigin | 10 | 0.625 ± 0.076 | 0.550 ± 0.082 | 0.599 ± 0.129 | 0.533 ± 0.133 | 0.524 ± 0.119 | 0.420 ± 0.151 | 0.911 ± 0.017 | 0.930 ± 0.018 | 0.921 ± 0.016 | 0.961 ± 0.070 | 0.763 ± 0.108 | 0.926 ± 0.100 | 0.355 ± 0.163 |
| Rosenbrock | 10 | 0.003 ± 0.002 | 0.004 ± 0.002 | 0.083 ± 0.008 | 0.016 ± 0.011 | 0.010 ± 0.005 | 0.065 ± 0.014 | 0.895 ± 0.170 | 0.803 ± 0.115 | 0.962 ± 0.127 | 0.044 ± 0.014 | 0.085 ± 0.011 | 0.004 ± 0.001 | 0.001 ± 0.000 |
| Styblinski-Tang | 10 | 0.200 ± 0.023 | 0.225 ± 0.042 | 0.571 ± 0.053 | 0.228 ± 0.028 | 0.333 ± 0.018 | 0.487 ± 0.068 | 1.236 ± 0.149 | 1.216 ± 0.053 | 1.184 ± 0.027 | 1.173 ± 0.326 | 0.815 ± 0.044 | 0.676 ± 0.113 | 0.170 ± 0.049 |
| Robot Pushing | 14 | 0.800 ± 0.177 | 0.797 ± 0.081 | 0.879 ± 0.082 | 0.970 ± 0.057 | 0.972 ± 0.057 | 0.986 ± 0.058 | 0.970 ± 0.026 | 0.795 ± 0.123 | 0.984 ± 0.031 | 0.892 ± 0.103 | 0.949 ± 0.043 | 0.673 ± 0.077 | 0.506 ± 0.070 |
| Ackley | 20 | 0.668 ± 0.127 | 0.261 ± 0.133 | 0.211 ± 0.050 | 0.219 ± 0.030 | 0.309 ± 0.078 | 0.607 ± 0.090 | 0.924 ± 0.019 | 0.931 ± 0.016 | 0.910 ± 0.007 | 0.959 ± 0.085 | 0.980 ± 0.002 | 0.912 ± 0.221 | 0.636 ± 0.111 |
| Levy | 20 | 0.078 ± 0.032 | 0.078 ± 0.078 | 0.117 ± 0.092 | 0.186 ± 0.157 | 0.114 ± 0.060 | 0.207 ± 0.052 | 0.924 ± 0.108 | 0.859 ± 0.093 | 1.151 ± 0.105 | 0.473 ± 0.078 | 0.743 ± 0.042 | 0.219 ± 0.086 | 0.093 ± 0.045 |
| Powell | 20 | 0.097 ± 0.068 | 0.006 ± 0.002 | 0.035 ± 0.023 | 0.082 ± 0.020 | 0.077 ± 0.008 | 0.118 ± 0.030 | 0.757 ± 0.147 | 0.690 ± 0.195 | 0.842 ± 0.134 | 0.086 ± 0.040 | 0.446 ± 0.142 | 0.020 ± 0.007 | 0.011 ± 0.006 |
| Rastrigin | 20 | 0.713 ± 0.083 | 0.614 ± 0.063 | 0.506 ± 0.090 | 0.618 ± 0.049 | 0.644 ± 0.077 | 0.562 ± 0.006 | 0.860 ± 0.048 | 0.833 ± 0.015 | 0.850 ± 0.022 | 0.923 ± 0.228 | 0.864 ± 0.018 | 0.725 ± 0.020 | 0.476 ± 0.102 |
| Rosenbrock | 20 | 0.038 ± 0.039 | 0.004 ± 0.002 | 0.029 ± 0.019 | 0.117 ± 0.064 | 0.055 ± 0.054 | 0.050 ± 0.016 | 0.645 ± 0.113 | 0.587 ± 0.109 | 0.978 ± 0.124 | 0.065 ± 0.022 | 0.394 ± 0.125 | 0.008 ± 0.004 | 0.005 ± 0.001 |
| Styblinski-Tang | 20 | 0.405 ± 0.163 | 0.357 ± 0.111 | 0.730 ± 0.069 | 0.396 ± 0.047 | 0.529 ± 0.030 | 0.560 ± 0.038 | 1.161 ± 0.102 | 1.093 ± 0.077 | 1.167 ± 0.034 | 1.193 ± 0.399 | 0.903 ± 0.037 | 0.723 ± 0.090 | 0.257 ± 0.059 |
| Ackley | 50 | 0.897 ± 0.052 | 0.859 ± 0.128 | 0.159 ± 0.010 | 0.402 ± 0.248 | 0.465 ± 0.246 | 0.538 ± 0.098 | 0.932 ± 0.036 | 0.957 ± 0.024 | 0.849 ± 0.024 | 0.863 ± 0.024 | 0.986 ± 0.001 | 0.360 ± 0.115 | 0.868 ± 0.009 |
| Levy | 50 | 0.039 ± 0.032 | 0.023 ± 0.041 | 0.043 ± 0.012 | 0.020 ± 0.007 | 0.048 ± 0.015 | 0.239 ± 0.080 | 0.667 ± 0.072 | 0.582 ± 0.135 | 0.883 ± 0.224 | 0.099 ± 0.019 | 0.873 ± 0.009 | 0.109 ± 0.026 | 0.227 ± 0.053 |
| Powell | 50 | 0.022 ± 0.010 | 0.016 ± 0.006 | 0.015 ± 0.008 | 0.025 ± 0.026 | 0.036 ± 0.035 | 0.076 ± 0.031 | 0.470 ± 0.158 | 0.533 ± 0.048 | 0.575 ± 0.194 | 0.046 ± 0.024 | 0.880 ± 0.034 | 0.017 ± 0.006 | 0.041 ± 0.015 |
| Rastrigin | 50 | 0.753 ± 0.083 | 0.601 ± 0.031 | 0.891 ± 0.420 | 0.587 ± 0.062 | 0.585 ± 0.055 | 0.579 ± 0.093 | 0.798 ± 0.052 | 0.820 ± 0.021 | 0.792 ± 0.028 | 0.644 ± 0.031 | 0.932 ± 0.003 | 0.769 ± 0.203 | 0.558 ± 0.042 |
| Rosenbrock | 50 | 0.016 ± 0.007 | 0.010 ± 0.003 | 0.007 ± 0.003 | 0.014 ± 0.006 | 0.035 ± 0.024 | 0.051 ± 0.021 | 0.656 ± 0.165 | 0.540 ± 0.085 | 0.698 ± 0.174 | 0.030 ± 0.006 | 0.794 ± 0.039 | 0.012 ± 0.003 | 0.057 ± 0.033 |
| Styblinski-Tang | 50 | 0.460 ± 0.223 | 1.063 ± 1.782 | 0.709 ± 0.130 | 0.449 ± 0.113 | 0.574 ± 0.097 | 0.721 ± 0.032 | 1.632 ± 0.109 | 1.134 ± 0.117 | 0.995 ± 0.045 | 0.613 ± 0.127 | 0.960 ± 0.012 | 0.436 ± 0.334 | 0.483 ± 0.059 |
| Rover trajectory | 60 | 0.475 ± 0.127 | 0.403 ± 0.115 | 0.511 ± 0.187 | 0.380 ± 0.083 | 0.485 ± 0.110 | 0.679 ± 0.139 | 0.684 ± 0.150 | 0.450 ± 0.145 | 0.473 ± 0.115 | 0.561 ± 0.060 | 0.923 ± 0.029 | 0.479 ± 0.160 | 0.186 ± 0.042 |
| Ackley | 100 | 0.684 ± 0.402 | 0.815 ± 0.360 | 0.136 ± 0.022 | 0.554 ± 0.291 | 0.476 ± 0.283 | 0.295 ± 0.082 | 0.952 ± 0.027 | 0.908 ± 0.057 | 0.883 ± 0.039 | 0.672 ± 0.129 | 0.997 ± 0.001 | 0.799 ± 0.159 | 0.904 ± 0.010 |
| Emb. Hartmann 6 | 100 | 0.089 ± 0.057 | 0.039 ± 0.028 | 0.179 ± 0.142 | 0.140 ± 0.093 | 0.100 ± 0.051 | 0.194 ± 0.128 | 0.636 ± 0.095 | 0.843 ± 0.026 | 0.717 ± 0.197 | 0.562 ± 0.324 | 0.882 ± 0.019 | 0.118 ± 0.125 | 0.068 ± 0.036 |
| Levy | 100 | 0.089 ± 0.105 | 0.043 ± 0.034 | 0.044 ± 0.014 | 0.039 ± 0.012 | 0.173 ± 0.085 | 0.061 ± 0.031 | 0.629 ± 0.051 | 0.735 ± 0.134 | 0.624 ± 0.127 | 0.131 ± 0.111 | 0.980 ± 0.021 | 0.086 ± 0.037 | 0.300 ± 0.020 |
| Powell | 100 | 0.117 ± 0.020 | 0.038 ± 0.029 | 0.008 ± 0.002 | 0.010 ± 0.003 | 0.034 ± 0.040 | 0.011 ± 0.004 | 0.482 ± 0.066 | 0.549 ± 0.089 | 0.477 ± 0.067 | 0.031 ± 0.012 | 1.019 ± 0.069 | 0.021 ± 0.009 | 0.112 ± 0.017 |
| Rastrigin | 100 | 0.503 ± 0.074 | 0.584 ± 0.192 | 0.548 ± 0.029 | 0.474 ± 0.068 | 0.550 ± 0.075 | 0.554 ± 0.061 | 0.759 ± 0.021 | 0.825 ± 0.041 | 0.774 ± 0.031 | 0.628 ± 0.049 | 0.990 ± 0.007 | 0.833 ± 0.365 | 0.584 ± 0.009 |
| Rosenbrock | 100 | 0.119 ± 0.018 | 0.026 ± 0.010 | 0.015 ± 0.004 | 0.059 ± 0.071 | 0.061 ± 0.049 | 0.036 ± 0.019 | 0.415 ± 0.078 | 0.600 ± 0.100 | 0.502 ± 0.068 | 0.199 ± 0.102 | 0.982 ± 0.040 | 0.043 ± 0.037 | 0.141 ± 0.012 |
| Styblinski-Tang | 100 | 0.372 ± 0.034 | 0.448 ± 0.158 | 0.505 ± 0.074 | 0.547 ± 0.103 | 0.635 ± 0.145 | 0.774 ± 0.097 | 0.947 ± 0.064 | 1.211 ± 0.140 | 0.910 ± 0.048 | 0.429 ± 0.077 | 0.988 ± 0.009 | 0.349 ± 0.051 | 0.527 ± 0.035 |
| **Mean** | | 0.307 | 0.287 | 0.295 | 0.264 | 0.286 | 0.329 | 0.882 | 0.866 | 0.899 | 0.624 | 0.734 | 0.434 | 0.245 |
| **Median** | | 0.173 | 0.170 | 0.179 | 0.189 | 0.173 | 0.239 | 0.924 | 0.908 | 0.917 | 0.672 | 0.873 | 0.360 | 0.145 |

## D.6 BO curves for all experiments in Table 2 and Table A1

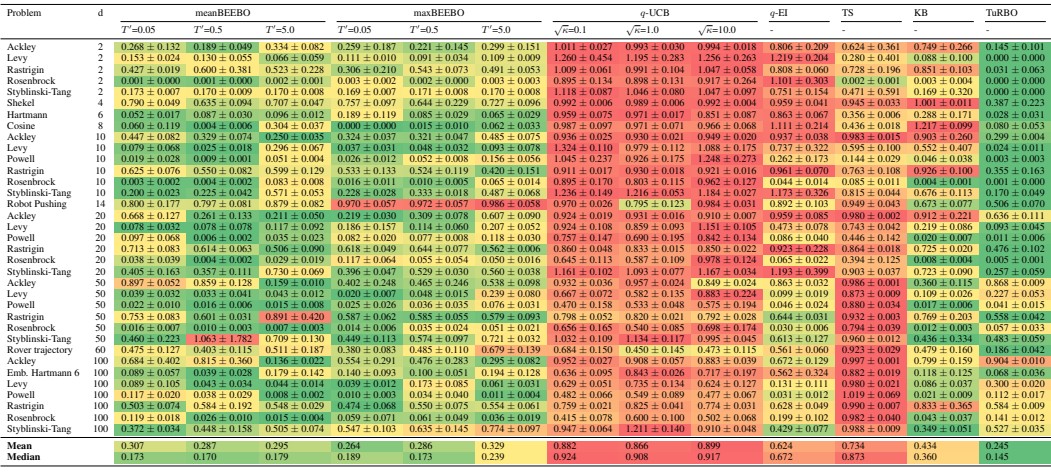

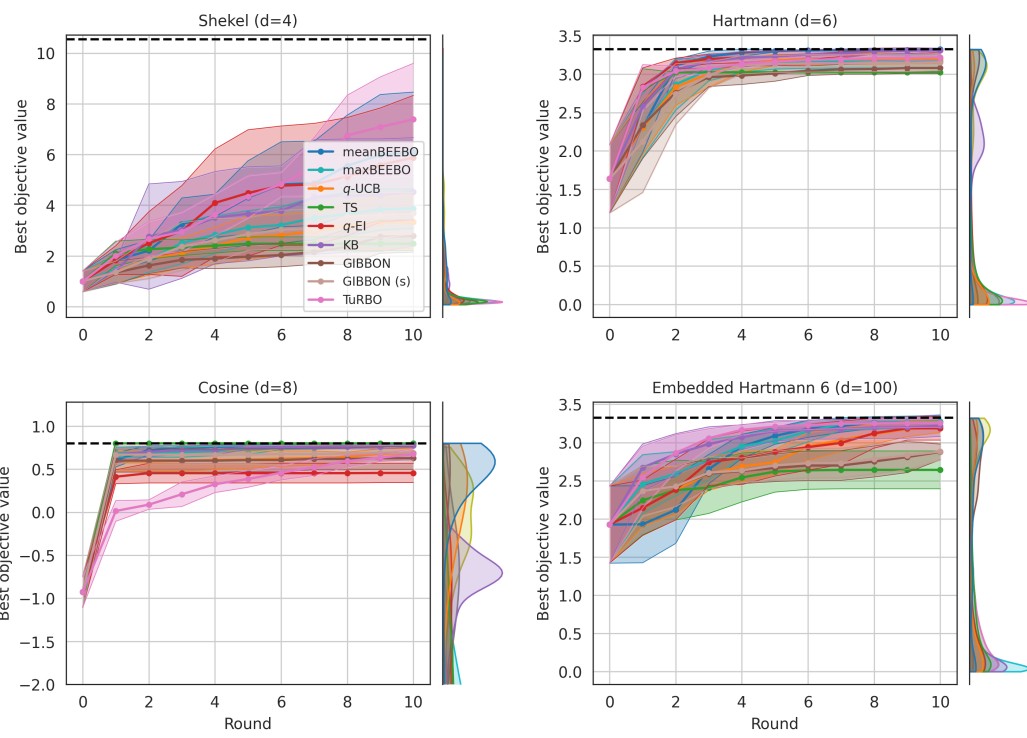

Figure A4: Experiments on the Shekel, Hartmann, Cosine and embedded Hartmann test functions with $\sqrt{\kappa} = 0.1$ for BEEBO and q-UCB.

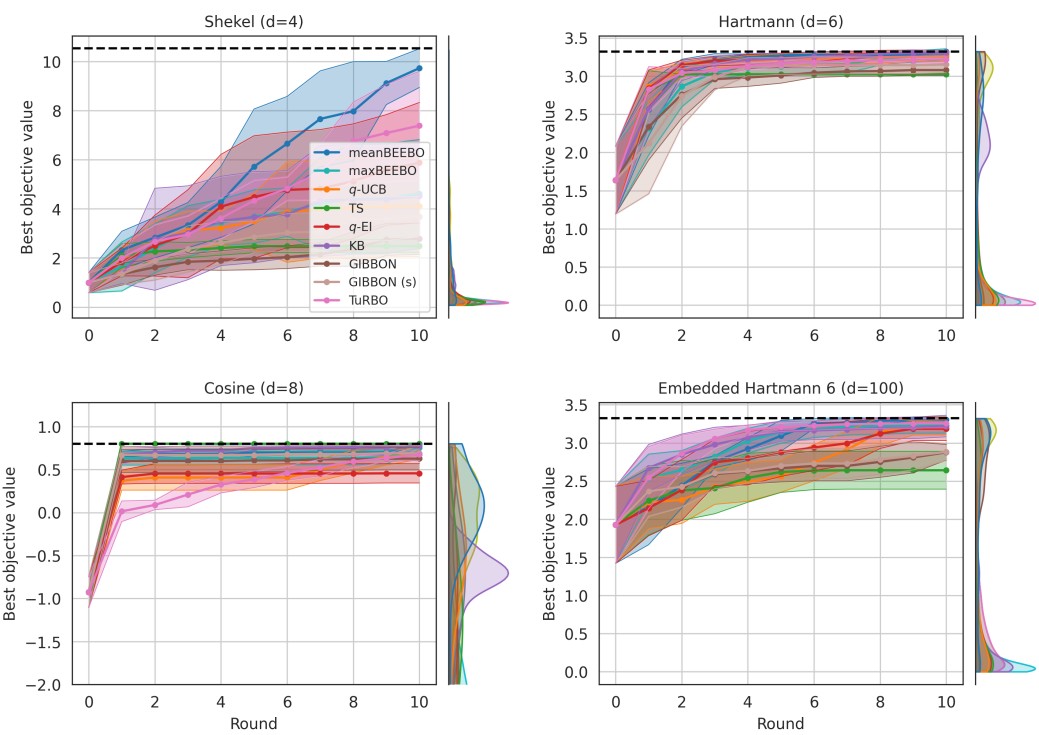

Figure A5: Experiments on the Shekel, Hartmann, Cosine and embedded Hartmann test functions with $\sqrt{\kappa} = 1.0$ for BEEBO and $q$-UCB.

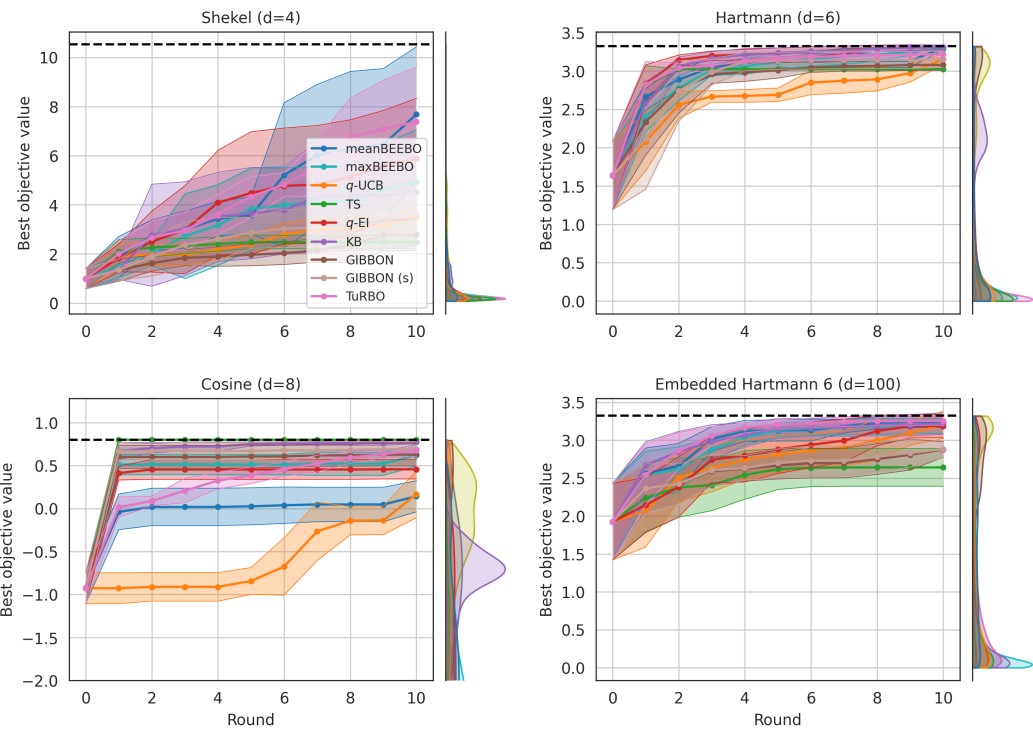

Figure A6: Experiments on the Shekel, Hartmann, Cosine and embedded Hartmann test functions with $\sqrt{\kappa} = 10.0$ for BEEBO and $q$-UCB.

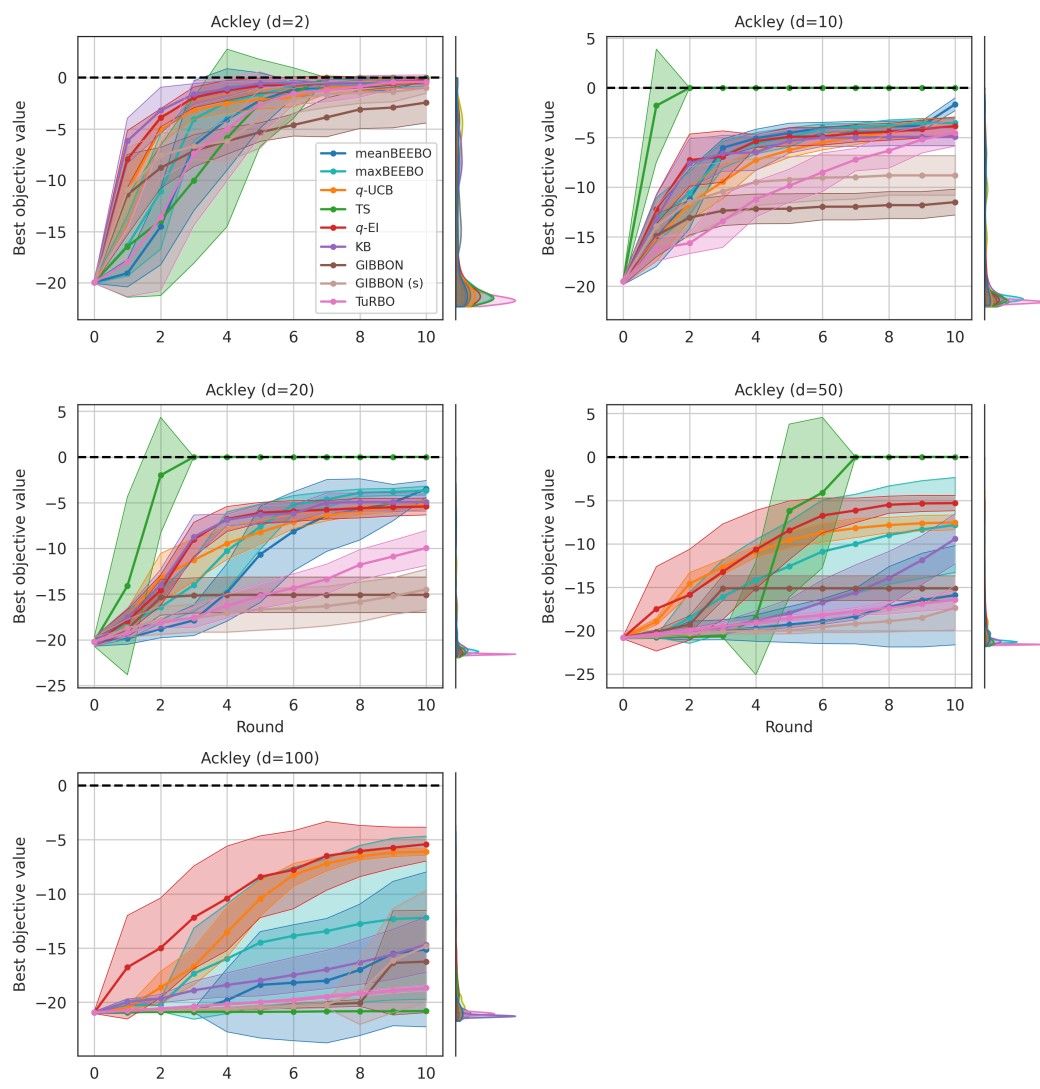

Figure A7: Experiments on the Ackley test function with $\sqrt{\kappa} = 0.1$ for BEEBO and $q-$UCB.

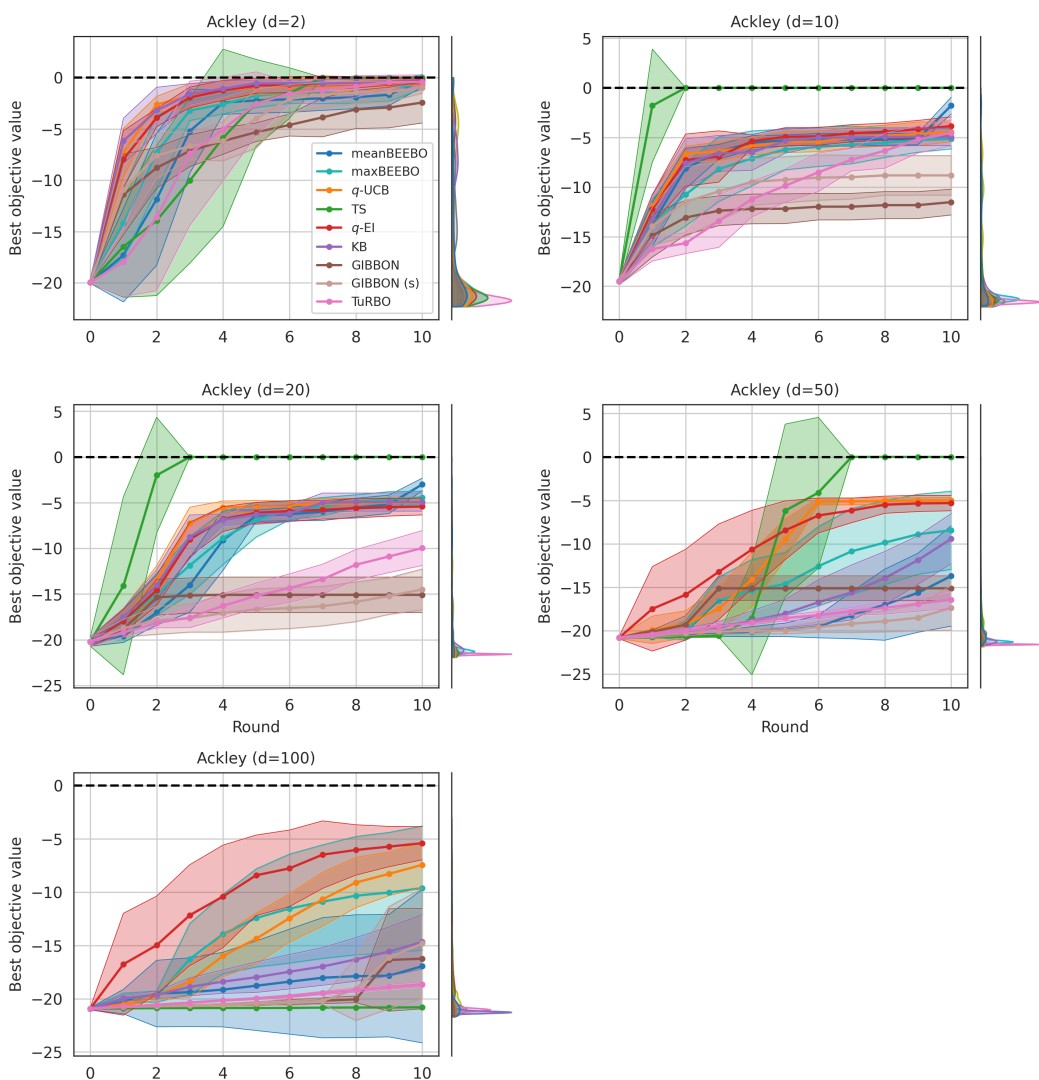

Figure A8: Experiments on the Ackley test function with $\sqrt{\kappa} = 1.0$ for BEEBO and $q-$UCB.

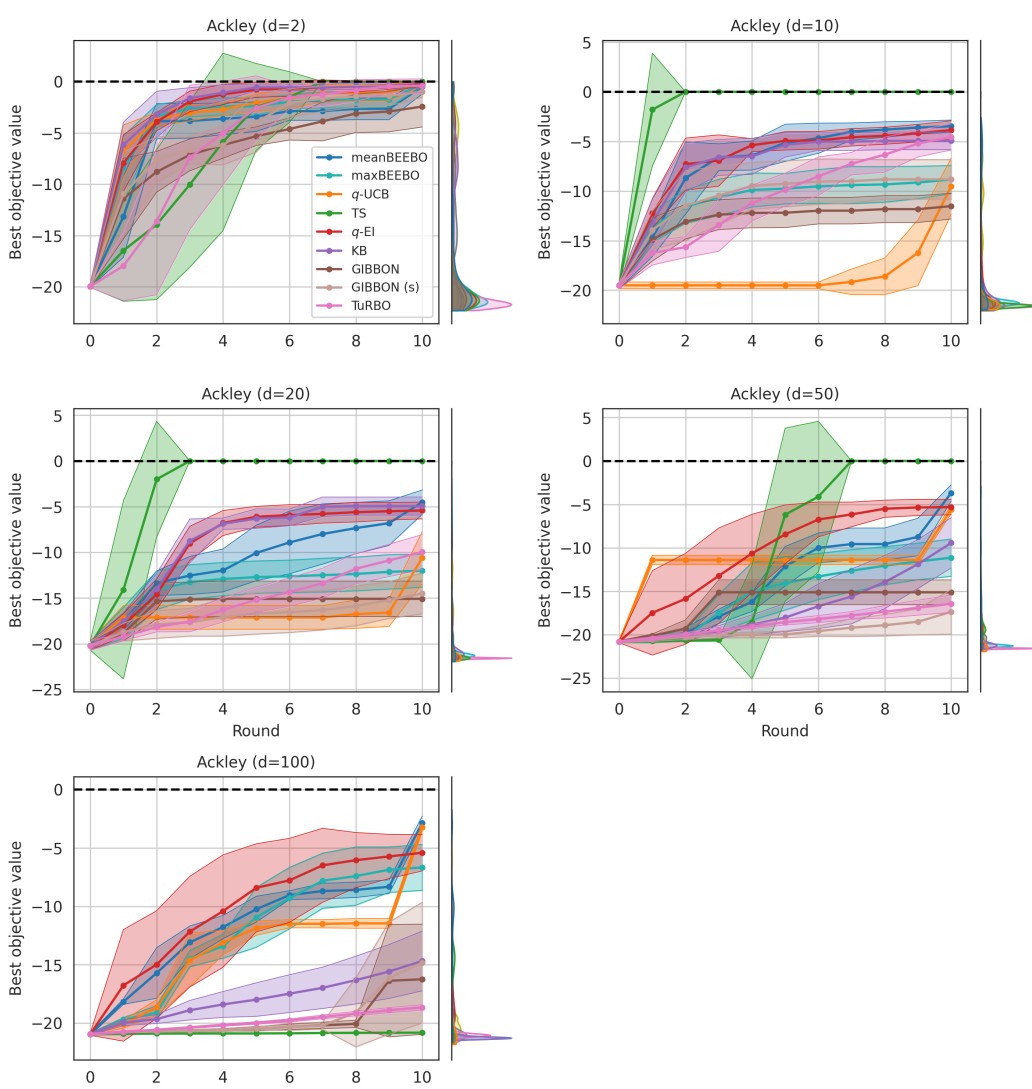

Figure A9: Experiments on the Ackley test function with $\sqrt{\kappa} = 10.0$ for BEEBO and $q-$UCB.

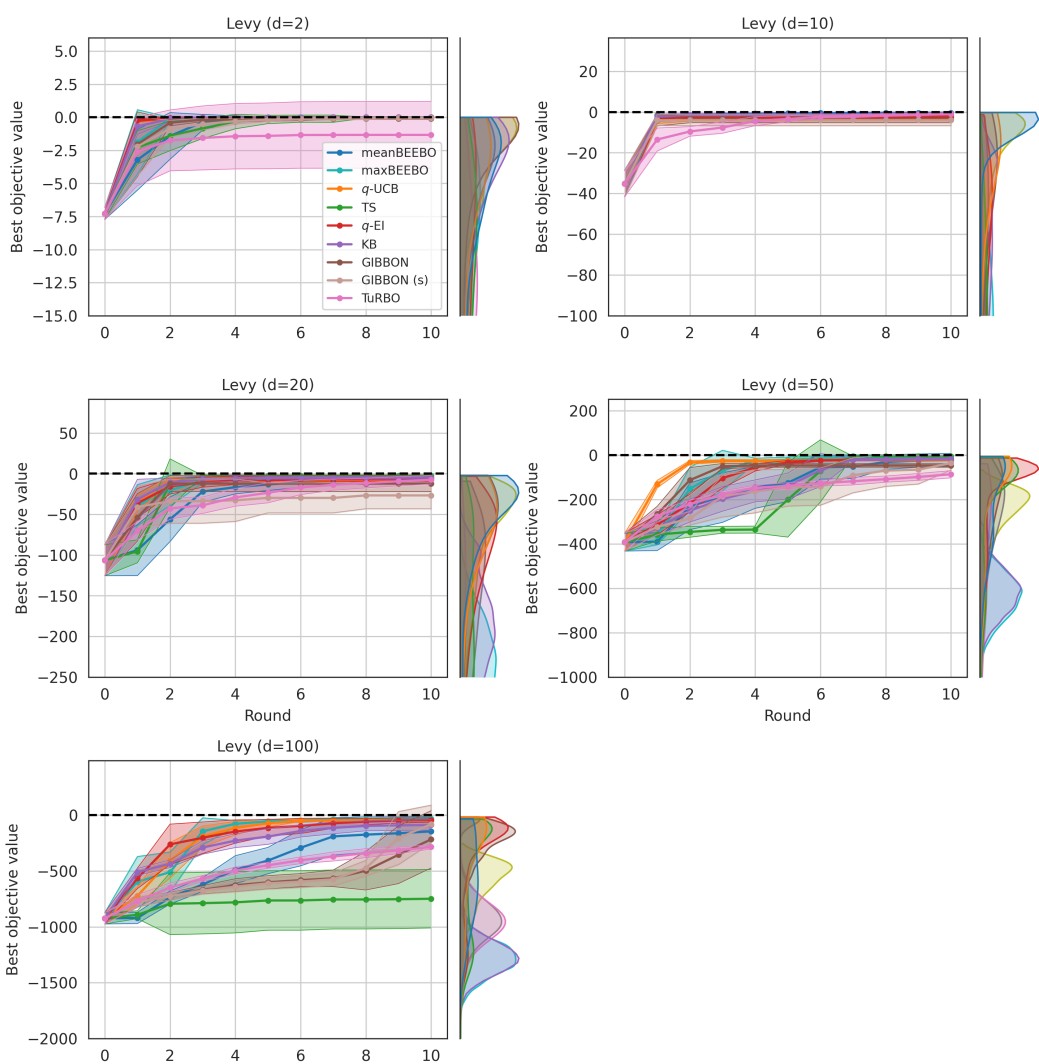

Figure A10: Experiments on the Levy test function with $\sqrt{\kappa} = 0.1$ for BEEBO and $q-$UCB.

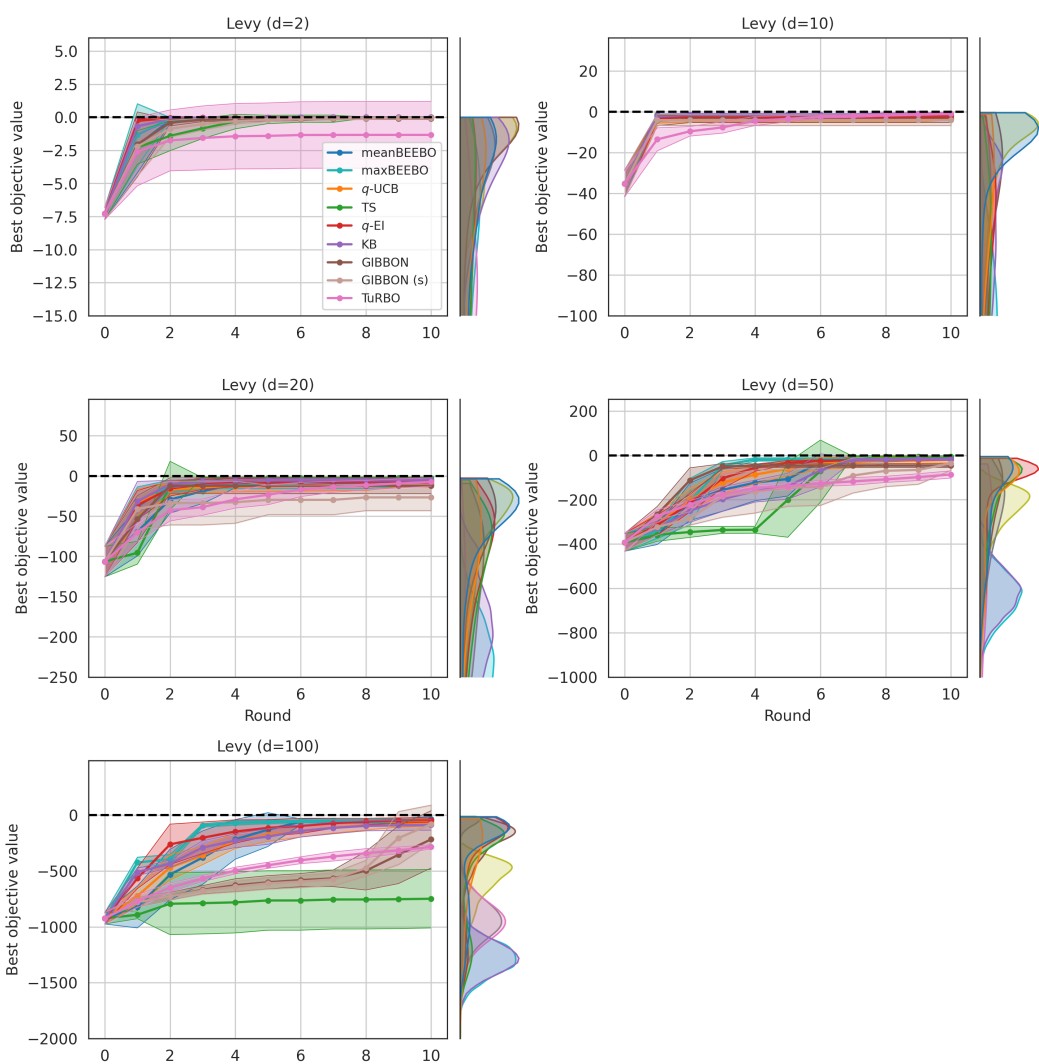

Figure A11: Experiments on the Levy test function with $\sqrt{\kappa} = 1.0$ for BEEBO and $q-$UCB.

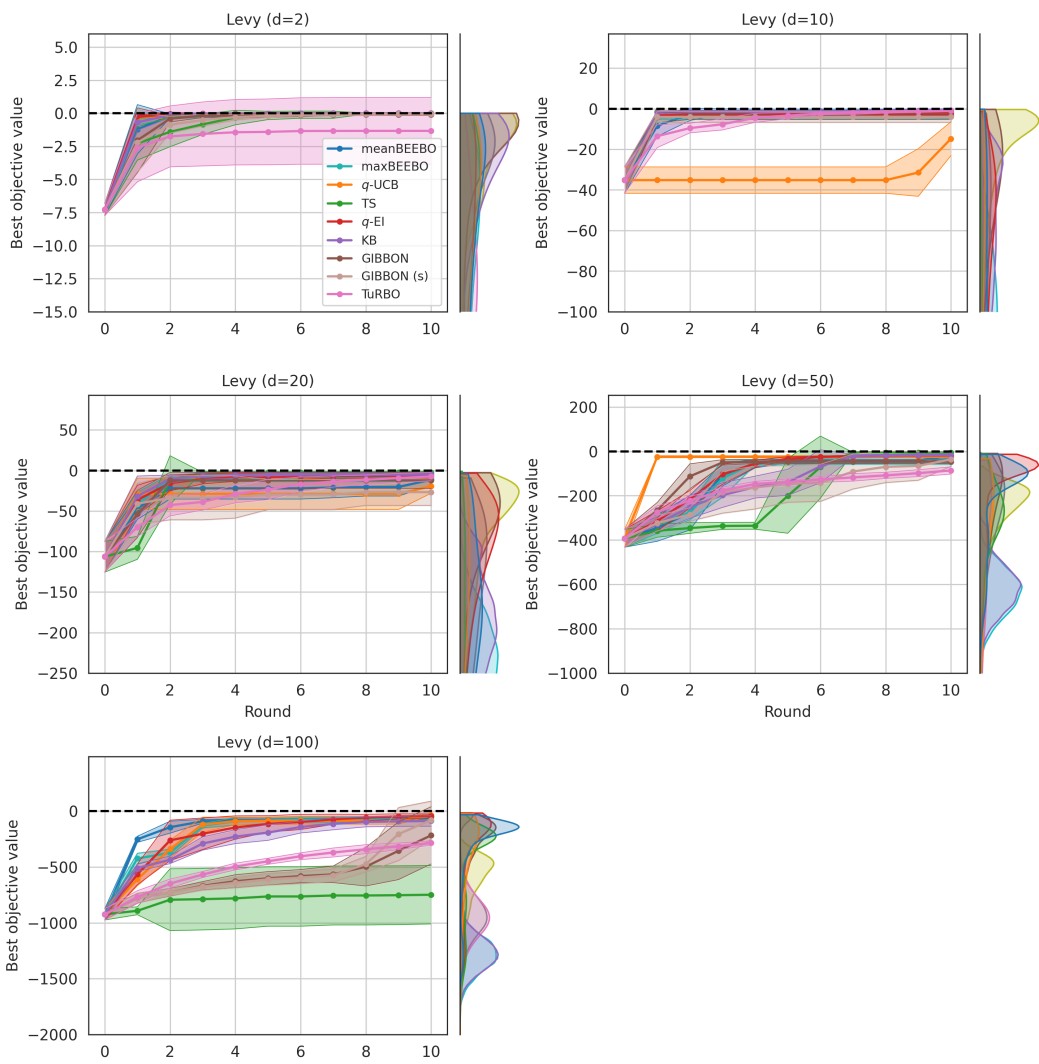

Figure A12: Experiments on the Levy test function with $\sqrt{\kappa} = 10.0$ for BEEBO and $q-$UCB.

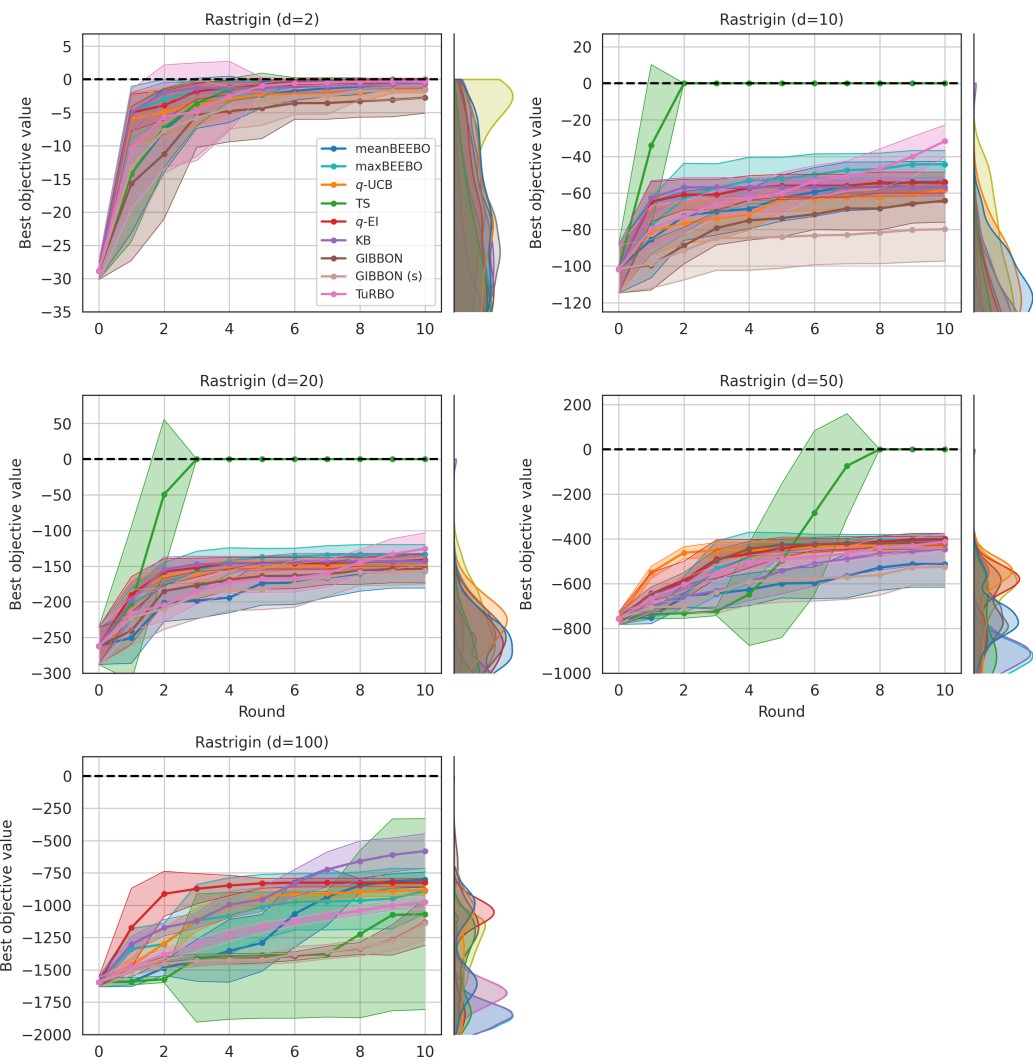

Figure A13: Experiments on the Rastrigin test function with $\sqrt{\kappa} = 0.1$ for BEEBO and $q$-UCB.

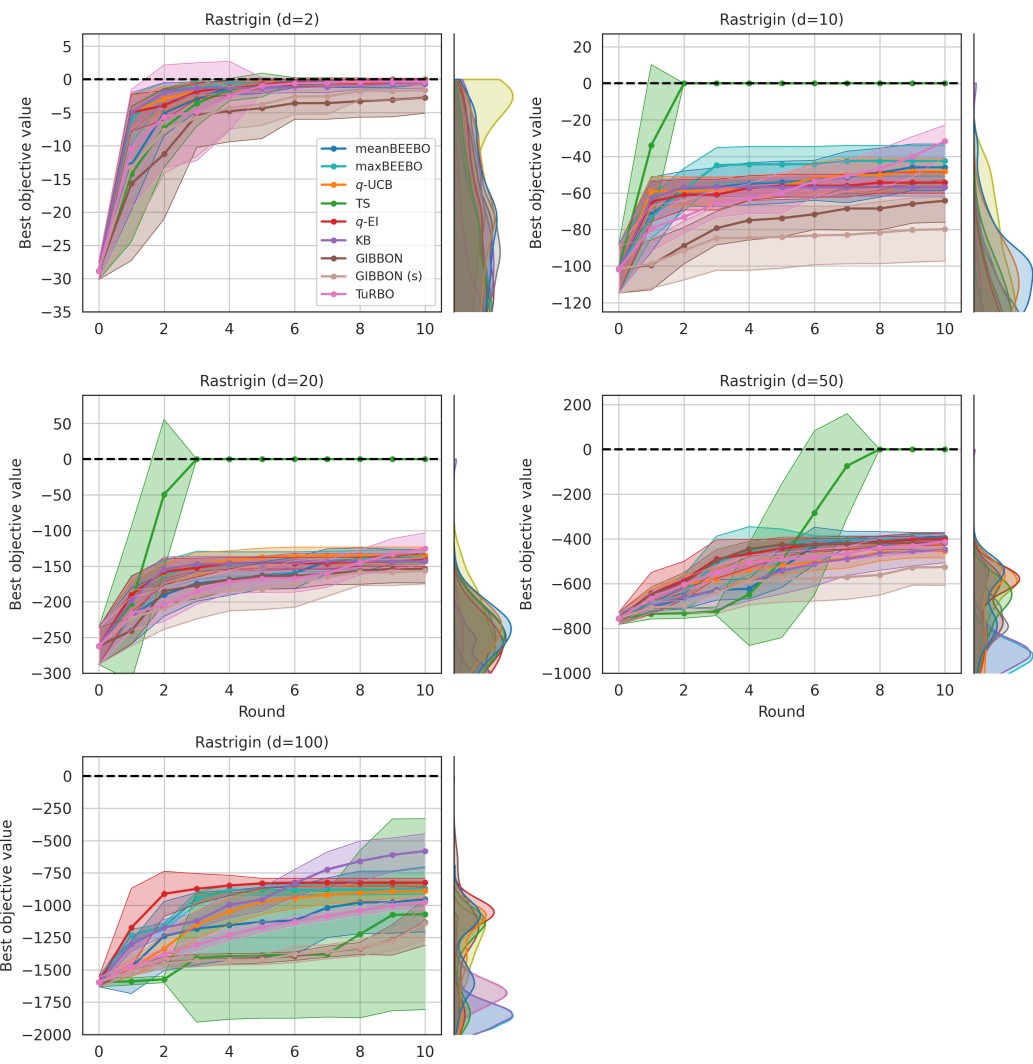

Figure A14: Experiments on the Rastrigin test function with $\sqrt{\kappa} = 1.0$ for BEEBO and $q$-UCB.

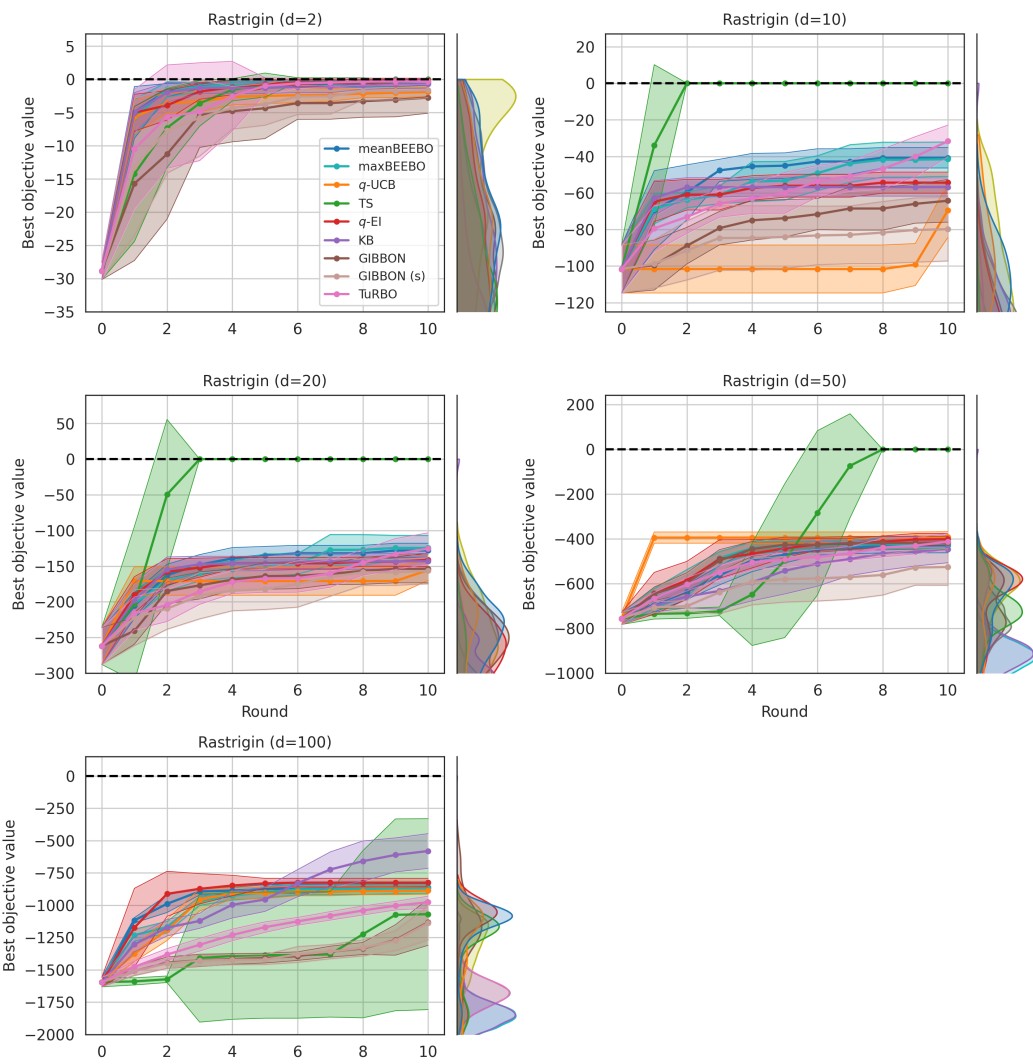

Figure A15: Experiments on the Rastrigin test function with $\sqrt{\kappa} = 10.0$ for BEEBO and $q$-UCB.

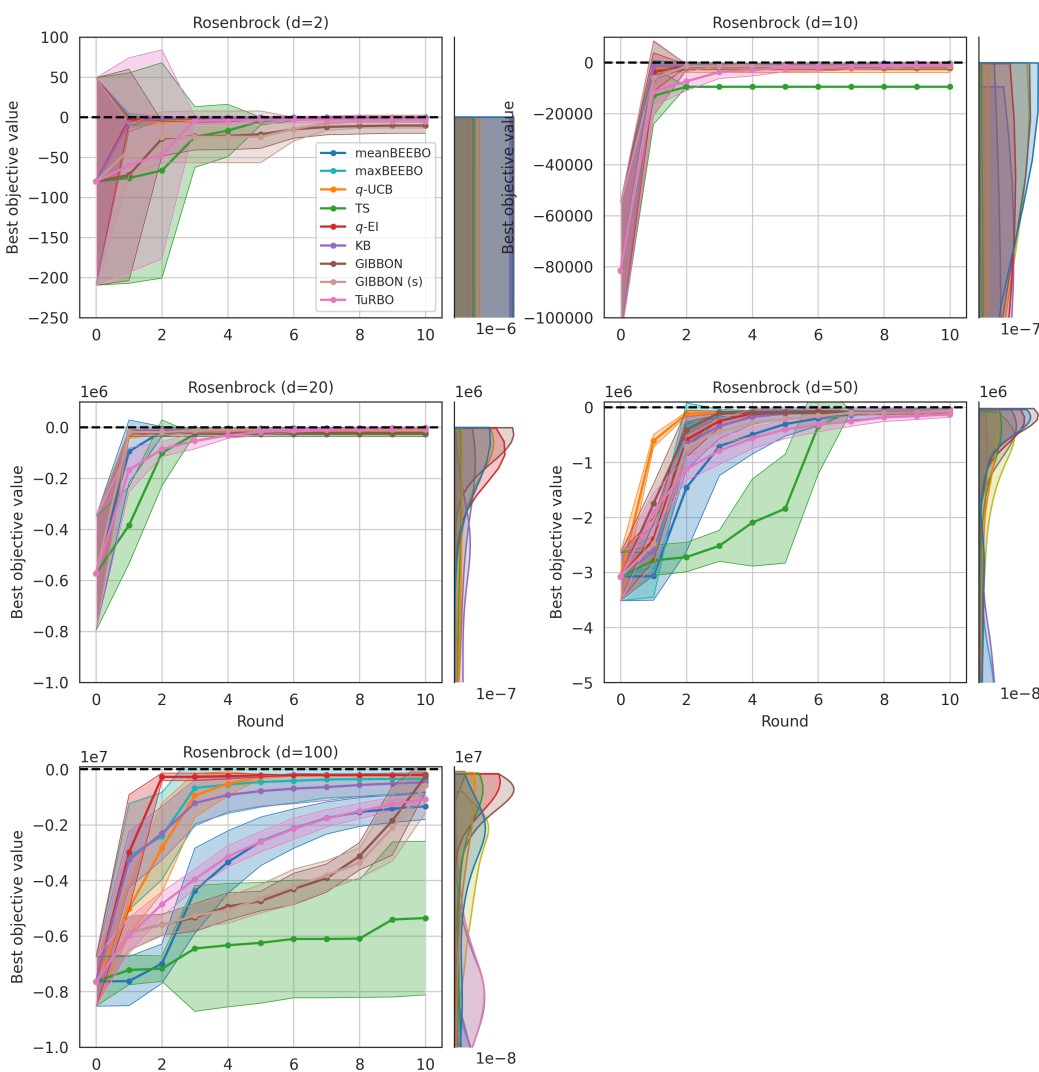

Figure A16: Experiments on the Rosenbrock test function with $\sqrt{\kappa} = 0.1$ for BEEBO and $q$-UCB.

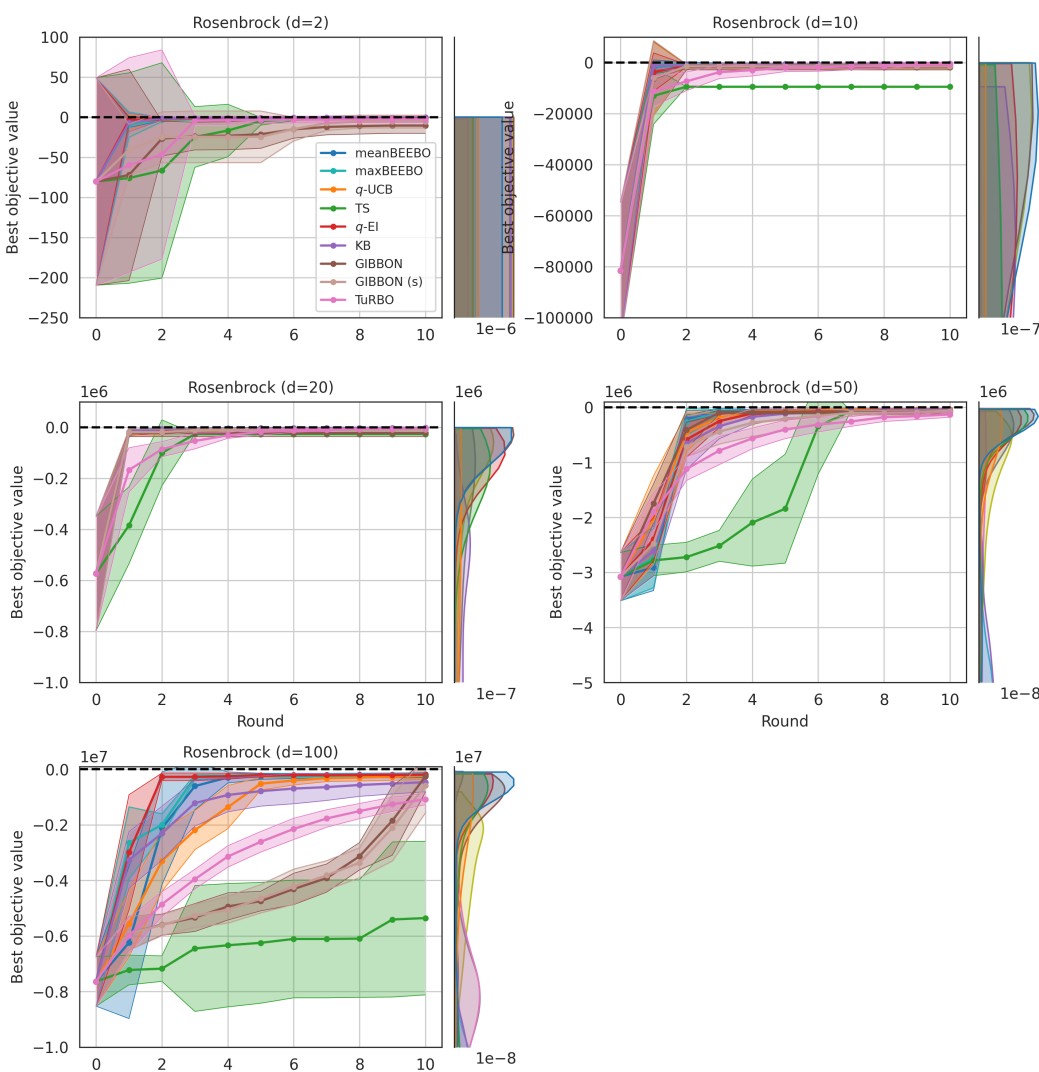

Figure A17: Experiments on the Rosenbrock test function with $\sqrt{\kappa} = 1.0$ for BEEBO and $q$-UCB.

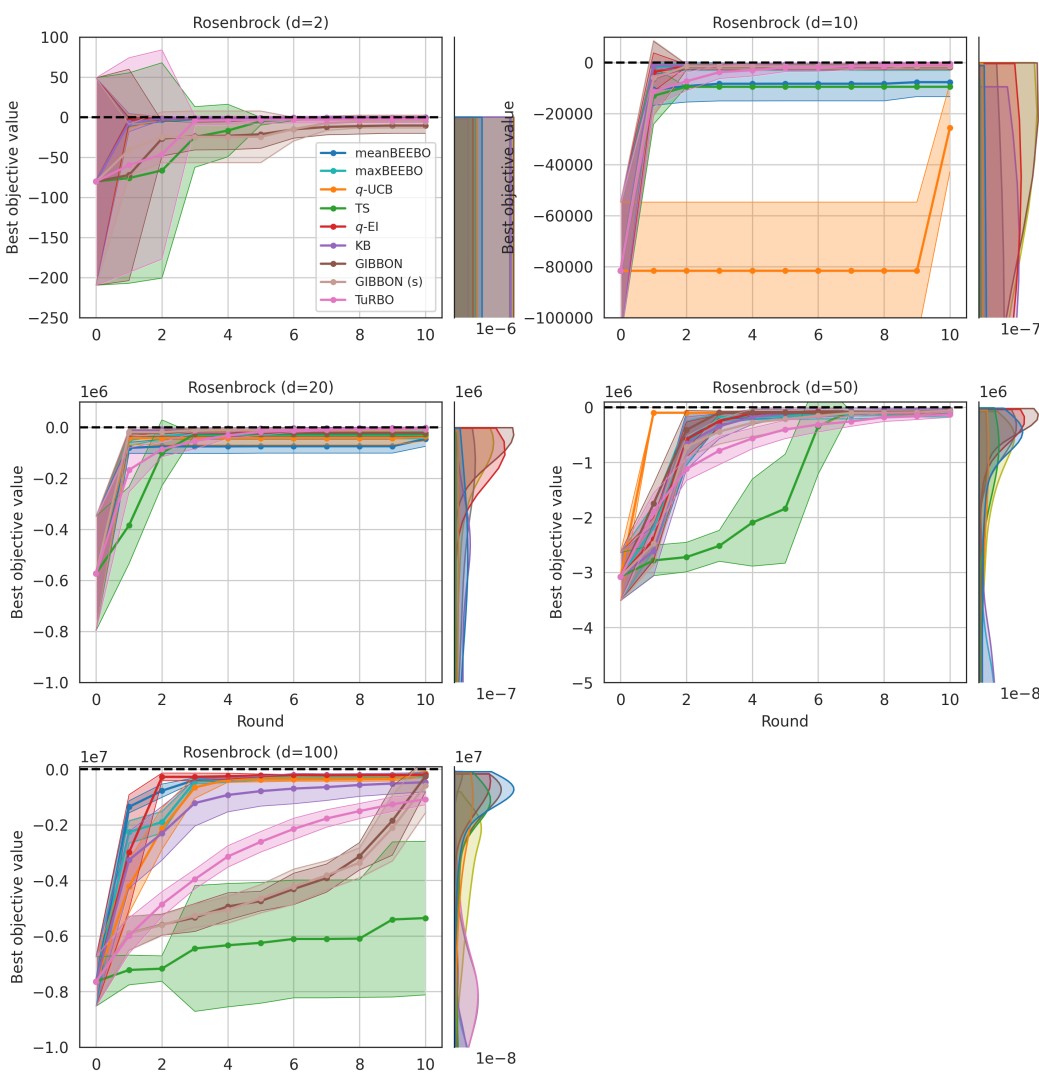

Figure A18: Experiments on the Rosenbrock test function with $\sqrt{\kappa} = 10.0$ for BEEBO and $q$-UCB.

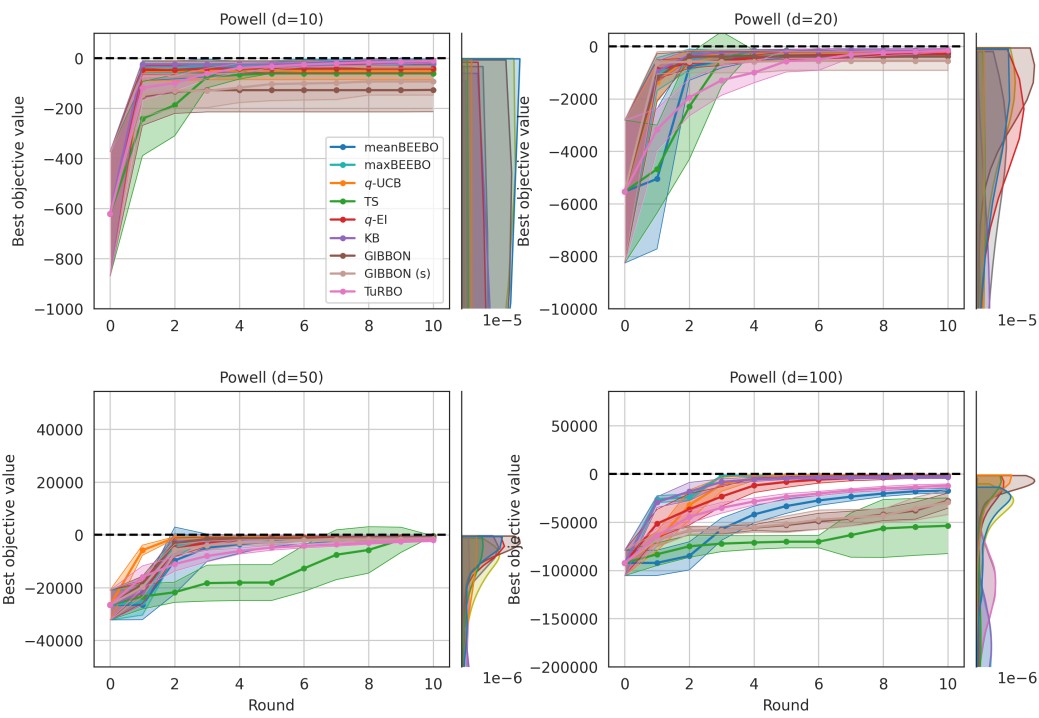

Figure A19: Experiments on the Powell test function with $\sqrt{\kappa} = 0.1$ for BEEBO and $q$-UCB.

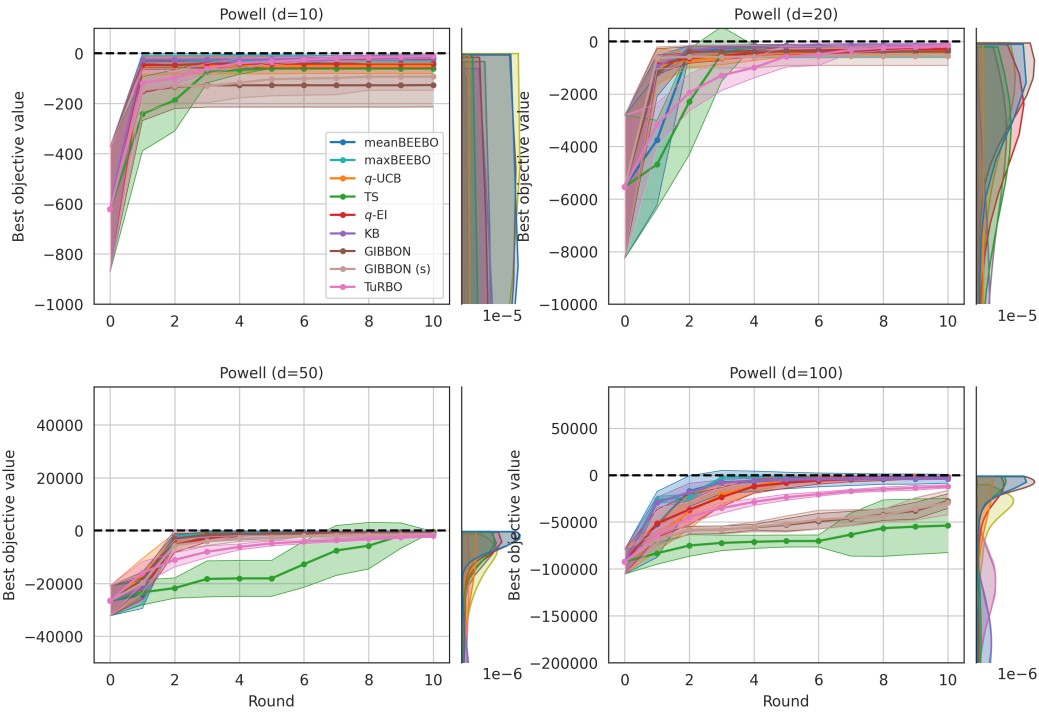

Figure A20: Experiments on the Powell test function with $\sqrt{\kappa} = 1.0$ for BEEBO and $q$-UCB.

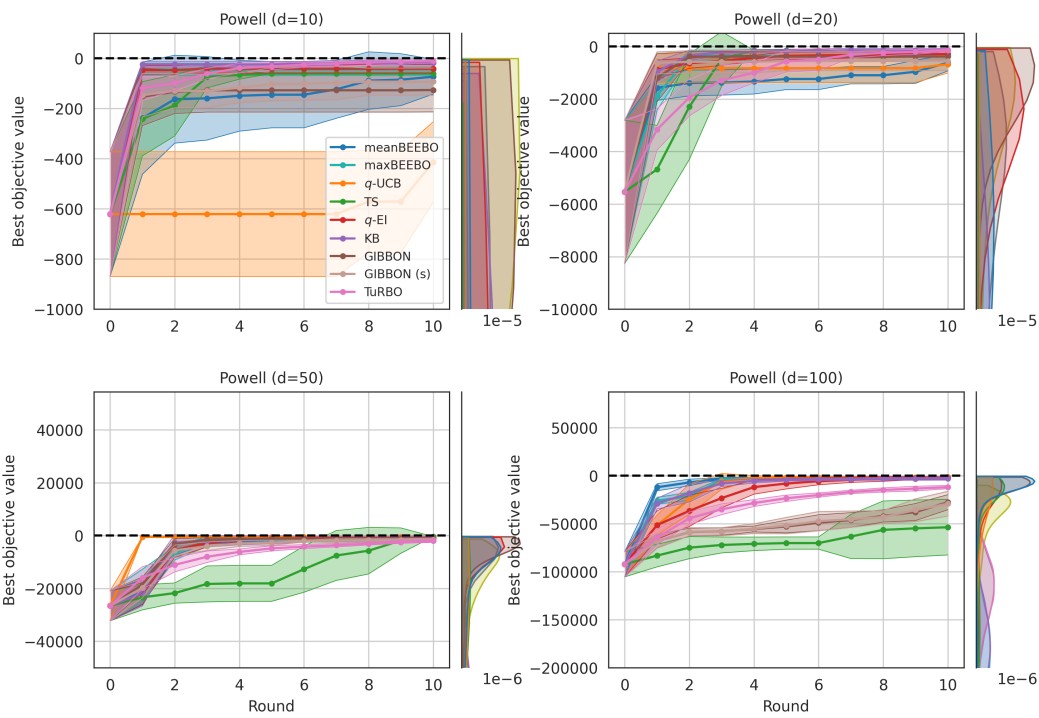

Figure A21: Experiments on the Powell test function with $\sqrt{\kappa} = 10.0$ for BEEBO and $q$-UCB.

