# OpenReview forum: "Batched Energy-Entropy acquisition for Bayesian Optimization"
_NeurIPS.cc/2024/Conference — NeurIPS 2024 poster_

### Official Review · Reviewer_CMG8 · 2024-07-07

**Soundness:** 2
**Presentation:** 3
**Contribution:** 2
**Rating:** 3
**Confidence:** 3

**Summary:**

This paper introduces a new acquisition function BEBBO for batched BO. BEBBO tries to build (negative) free-energies-like acquisition function, enabling gradient-based optimization, tight exploration-exploitation control, and risk-averse BO under heteroskedastic noise. It tries to improve existing parallel acquisition functions in the following ways:
1.	uses a hyper-parameter T to directly balance exploration and exploitation by separating these two parts clearly;
2.	keeps the behavior predictable by scaling E and I with batch size;
3.	enables the optimization of gradient descent by holding the availability of closed-form expressions for GP.
This paper demonstrates several experimental comparisons and shows its effectiveness.

**Strengths:**

1.	This paper shows an enlightening acquisition function method inspired by statistical physics.

2.	The experimental results show the effectiveness of BEBBO on problems without noise or with heteroskedastic noise.

**Weaknesses:**

1.	The idea is straightforward: simply combine two common components—entropy reduction and the weighted sum of values. In my opinion, the novelty is not very strong.

2.	The article doesn't discuss the situation when BEEBO is used with other surrogate models. It seems that if BEEBO is not used with GP, it loses the advantages of closed-form expressions and gradient descent optimization.

3.	In control problems shown in Figure A2, the performances of meanBEBBO and maxBEBBO are not outstanding. Especially, these two variants are surpassed by KB in Robot pushing problem.

4.	Although BEEBO performs well on many synthetic test problems, its versatility and effectiveness require more experimental validation in specific applications.

5.	In the experiments in main text, the authors only showed the comparison with q-UCB. It would be better to show comparisons with other batched baselines and provide a thorough analysis. The comparison with q-UCB shows the advantage on the balance between exploration and exploitation. But other advantages emphasized in the paper, such as the benefits of gradient descent optimization and the tight control of the explore-exploit trade-off, are not fully demonstrated. I suggest that the authors can re-organize the paper to move the comparison with other baselines to the main paper.

6.	The theoretical analysis is not deep. No regret bound is analyzed.

**Questions:**

- How to set the parameter T properly? Do you have some instructions?

- In line 158, why does I(x) also scale linearly with batch size?

- The Equation A5 is not shown.

- What are the advantages of multiplying batch size Q in E(X)?

- How does the behavior of BEBBO change with batch size?

**Limitations:**

Yes

---

> ### Author Rebuttal · Authors · 2024-08-06
>
> Thank you for your comments and questions. We respond to them individually below, and are looking forward to further discussion.
>
> > 1. The idea is straightforward [..]
> We are convinced that BEEBO is a novel acquisition function. In section B.2, we extensively compare BEEBO to the, to our knowledge, most similar existing work, GIBBON.  We show how BEEBO’s entropy term can have advantages over GIBBON’s, and empirically show that the closed-form summarization term works better for large batches than GIBBON’s lower bound approximation.
>
> Moreover, BEEBO’s closed-form softmax-weighted sum expectation is novel. It is of course not our intent to claim that we have discovered entropy reduction. However, we are not aware of any existing work that proposed an acquisition function of this form for batch BO.
> >2 . [...] other surrogate models.
>
> Indeed, BEEBO was designed to work primarily with GPs, as closed-form expressions and multivariate normal posteriors enable efficient computation. We have edited the abstract to already mention the GP focus there. In the outlook section, we now provide a discussion of how BEEBO could be generalized:
>
> >Beyond GPs, BEEBO could be generalized to work with any probabilistic model. However, GPs are unique in that $H_{\textup{aug}}$ is available in closed form and can be used to compute $I(\mathbf{x})$ analytically, without solving the integral over $\mathbf{y}$ in Eq. 4. Other models may require approximations and sampling-based approaches for computing the information gain.
> >3. In control problems [...] the performances [...] are not outstanding [...]
>
> We agree - on these problems, KB turns out to be very strong, not just in comparison to BEEBO, but all baselines including TuRBO, which was originally found to be state of the art for these problems. We think that this is a consequence of using LogEI. Hence, we also mentioned in the results section of the main text that KB using LogEI performs very competitively for large batches. We have now updated this statement to explicitly point to the control problems also.
> >4 . Although BEEBO performs well on many synthetic test problems [..] more experimental validation in specific applications.
>
> We are happy to include additional experiments. Are there any specific ones that we should consider? Overall, we are convinced that our empirical section is comprehensive and well aligned with common practices in BO research, such as e.g. Wilson et al, NeurIPS 2017 [24], Ament et al, NeurIPS 2023 [57], Gonzalez et al. AISTATS 2016 [37].
>
> > 5. [...] comparisons with other batched baselines [...]
> Thank you for the recommendation. We had tested multiple layouts for the results and ultimately decided that given the number of experiments and methods, including the baselines in the main text tables makes them too hard to read and interpret. We have thus maintained the focus on q-UCB, and discuss additional baselines in the text, ensuring the supplement is referred to at numerous places.
>
> We believe that both Figure 1, as well as the 33 batch regret experiments in Table 3, fully demonstrate the controllability of the trade-off in BEEBO and highlight the key problem with q-UCB not having tight control with large batch sizes.
> > 6. The theoretical analysis is not deep. No regret bound is analyzed.
>
> We provide an extended analysis of the BEEBO algorithm in the supplementary section B, tying BEEBO to the rich theoretical framework of existing methodology, such as UCB, DPP and local penalization.
> A theoretical analysis focusing on regret bounds, along the approach outlined for B-UCB’s bounds in [34] would indeed be very interesting, and we believe this deserves its own paper.
> Given the provided theoretical linkage to existing methodology, we consider the manuscript to be quite comprehensive, focusing on empirical demonstration of BEEBO's efficiency. With 33 experiments conducted, three different batch sizes (q=5,10,100), eight different acquisition strategies, several of them operated at different scales (GIBBON default & scaled, q-UCB & BEEBO at different T), and 10 replicates each (corresponding to more than 12,800 recorded BO trajectories total), we consider our benchmarking approach thorough. We believe that including an analysis of regret bounds in the manuscript as well will only challenge the focus of our paper for a reader even more, as already indicated by reviewer mXVT.
> > How to set the parameter T properly
>
> This is indeed very important - we do not want the controllability (which we consider an advantageous feature in general) to become a nuisance. As outlined in our main response, we now feature the relation of T to UCB more prominently.
> > In line 158, why does I(x) also scale linearly with batch size?
>
> The linear scaling of $I(x)$ is a consequence of I being a log determinant (or more precisely, the difference of two log determinants). Considering the log determinant as the sum of log eigenvalues, we see that the scaling is in fact linear, as we add an eigenvalue for each increase in Q.
> > The Equation A5 is not shown.
>
> We removed this trailing newline.
> >What are the advantages of multiplying batch size Q in E(X)?
>
> To maintain a Q-independent T parameter, we wish for both $I(x)$ and $E(x)$ to scale equally. As we define $E(x)$ to be the mean or the softmax-weighted sum, this expression does not scale with Q itself, and we therefore multiply it.
> >How does the behavior of BEBBO change with batch size?
>
> As just outlined, the explore-exploit parametrization of BEEBO is invariant. That being said, it is correct that there will always be an effect from Q. Specifically, at large Q compared to the search domain, one might expect to see the exploration (=repulsion) term grow larger, as points will compete to occupy the same regions of the domain. We believe that our additional low-Q experiments in the updated manuscript support this, showing that at low Q higher explore rates are beneficial for BEEBO as well as for q-UCB.

---

> > ### Comment · Reviewer_CMG8 · 2024-08-11
> >
> > Thanks for the response. I keep my evaluation.

---

### Official Review · Reviewer_4oKm · 2024-07-07

**Soundness:** 2
**Presentation:** 2
**Contribution:** 2
**Rating:** 4
**Confidence:** 3

**Summary:**

Proposing a new acquisition function inspired by statistical physics, which allows explicit control of exploration-exploitation trade-offs in a batch BO setting.

**Strengths:**

Drawing inspiration from statistical physics is a promising direction, as it naturally aligns with Bayesian approaches.

**Weaknesses:**

**Major Points**
1. **Lack of Unique Selling Point:**
   The method does not appear to solve any unique cases that other methods cannot. While the related work section outlines many similar approaches, this work only compares itself to q-UCB. Without a theoretical study, such as convergence rate analysis, there is insufficient motivation to adopt another heuristic approach like this. To demonstrate efficacy, a comprehensive empirical study with extensive comparisons to existing works is necessary, given the unclear advantages.

2. **Review of Claimed Selling Points:**
   - **Not Based on MC Integration Like BoTorch:**
     While this is true, it is unclear if it is beneficial. MC approaches are approximations but have convergence guarantees (refer to the BoTorch paper's appendix). This work lacks such guarantees.
   - **Tight Control of Exploration-Exploitation Trade-off:**
     The proposed method is not the only solution. UCB theory can bound the feasible region by [max LCB(x), UCB(x)] with 1 - $\delta$ probability (e.g., see [1]). This region can be controlled by $\beta$ hyperparameters, corresponding to $\delta$. Constrained batch BO within this region would yield similar results with theoretical guarantees.
   - **Heteroskedastic BO:**
     There are no comparisons with existing methods. UCB variants can address this problem. Vanilla UCB theory does not differentiate between epistemic and aleatoric uncertainty. Therefore, UCB with heteroscedastic GP can serve the same purpose. For example, training a heteroscedastic GP with the observed dataset and replacing the inner GP on noise variance with normal isotropic likelihood variance (pure epistemic uncertainty) will yield similar results as risk-averse batch BO, without needing the change in modelling the acquisition functions regardless of hetero-/homo-scedastic cases like this work.

3. **Setting k in Practice:**
   Setting  $k$ is not an easy task for users. In UCB, $\beta$ presents a similar challenge, but there are theoretical guidelines and algorithm to compute this (e.g., [2]).

4. **Constrained Uncertainty Sampling:**
   This work can be understood as a variant of constrained uncertainty sampling. As [3] explains, variance reduction can be written similarly to the entropy proposed in this paper (see section 2.2). It also shows that the variance-only approach is inferior to UCB both theoretically and empirically. The batch setting may lead to model misspecification, particularly when hallucination (aka fantasization) is applied. The concerns and approach are notably similar to ([49] in your citation number), making a comparison with their method unavoidable.

5. **Data Augmentation Procedure:**
   The explanation is unclear. Is it fantasization (aka hallucination) or simply observed points? How does this differ from Eq.(3) in [3]?

- [1] Fengxue Zhang, Jialin Song, James C Bowden, Alexan- der Ladd, Yisong Yue, Thomas Desautels, and Yuxin Chen. Learning regions of interest for bayesian optimiza- tion with adaptive level-set estimation. In International Conference on Machine Learning, pages 41579–41595. PMLR, 2023.
- [2] Kihyuk Hong, Yuhang Li, and Ambuj Tewari. An optimization-based algorithm for non-stationary kernel bandits without prior knowledge. In International Conference on Artificial Intelligence and Statistics, pages 3048–3085. PMLR, 2023.
- [3] Srinivas, N., Krause, A., Kakade, S., & Seeger, M. (2010). Gaussian Process Optimization in the Bandit Setting: No Regret and Experimental Design. In Proceedings of the 27th International Conference on Machine Learning (pp. 1015-1022).

**Questions:**

Questions are written in the above weakness section.

**Limitations:**

Limitations are discussed in the discussion section.

---

> ### Author Rebuttal · Authors · 2024-08-06
>
> We thank the reviewer for their detailed comments. We have answered them individually below, and are happy to discuss further.
> >**Lack of Unique Selling Point**
>
> We would like to point out that as mentioned in line 275, supplementary section D.1 provides a comprehensive benchmark beyond q-UCB, including GIBBON, TS, KB, q-EI and now, as suggested by reviewer x4GE, also TurBO.
>
> Moreover, we provide an extensive theoretical analysis of how BEEBO compares to other strategies in supplementary section B, highlighting how BEEBO can be understood in light of existing works and improve upon them, tying our method to theory such as DPPs.
>
> BEEBO does in fact address issues that other methods face: q-UCB lacks the controllability of acquisition behavior in practice, as quantified in Table 3, and GIBBON is known to scale poorly to larger batch sizes, whereas BEEBO does so.
> >**Not Based on MC Integration**
>
> We believe that Figure 1 and Table 3 highlight a serious practical problem with large batch q-UCB in BoTorch. Moreover, the MC q-EI also failed to outperform BEEBO in our extended benchmark.
>
> As BEEBO is an analytical expression without sampling, we are unsure what kind of convergence guarantees are meant by the comment. We optimize BEEBO using gradient descent, and while it is, as most acquisition functions, multimodal, it can be optimized efficiently using multiple restarts, as is common in BO practice (and done in BoTorch).
> >**Tight Control of Trade-off**
>
> Thank you for the reference. BALLET uses local regions to shrink high-dimensional search spaces, similar to TuRBO. However, unlike TuRBO, the paper is exclusively about single-point BO. It is not directly obvious to us how batched acquisition is meant to be done using their acquisition function (Eq. 7) without further generalizing BALLET. The sampling-based acquisitions (Eq. 8,9) may permit batch mode, but no mention of this is made in the paper.
>
> We would like to note that the primary focus of our work is batch BO, rather than high-dimensional BO. We now include TuRBO as an exemplary method for trust region approaches.
>
> We think it would be inadequate to benchmark single-point works that make no mention of batched BO in their own experiments. It is beyond the scope of our paper to generalize existing single-point works to batch mode. If there is a batched BALLET that we are not aware of, we would be happy to add it to our benchmark.
> We have modified our claims to explicitly state “Tight Control of the explore-exploit trade-off *in batch mode*”.
> >**Heteroskedastic BO**
>
> We fully agree that existing (single-point) methods also address heteroskedasticity. Hence we compare to Makarova et al.’s RAHBO [11] in supplement section B. We are happy to include any other relevant UCB variants that handle batch BO. A key limitation of existing work is that to our knowledge, e.g. RAHBO has not been generalized to batches. The core focus of our work is batch, with heteroskedasticity as a secondary consideration.
>
> As you mention, UCB does not differentiate between epistemic and aleatoric uncertainty. Thus we are unfamiliar with the concept that we can get risk-averse BO from a heteroskedastic GP without modifying UCB as done in e.g. Makarova et al. Could you provide a reference?
> >**Setting k**
>
> It is true that setting parameters in BO can be challenging. However, in our work, we considered UCB-like controllability a desirable feature. In supplementary section B.1, we show how BEEBO’s $T$ and UCB’s $\beta$ (or $\kappa$, as used this work to avoid confusion with the softmax $\beta$) are related. We always use $T’$, a parameter scaled for UCB equivalence, to allow users to follow existing guidance and allow fair experimental comparison.
> As outlined in the main response, we have edited section 3.1 to feature this more prominently.
> >**Constrained Uncertainty Sampling**
>
> [3] indeed also discusses the information gain. However, we understand that it does not use it for batch BO.
> We are unsure how the “variance-only approach” comment is meant to be understood in light of our work. We would also expect only using the variance of a point to be a poor strategy for BO - however, we do no such thing.
>
> Could you provide a reference regarding misspecification, and why batch BO in general causes more misspecification vs. single-point? We understand that [49] considers misspecification of GPs irrespective of the BO strategy, with no causal relation to batch BO.
> Note that BEEBO does not use iterative fantasization - we immediately discard fantasized GPs after computing H_aug. The model itself remains unchanged for further gradient descent iterations, and no hallucinated observations are used.
>
> We attempted including SOBER [49]. Following the quickstart guide in the official repository, and plugging that into the experimental loop that we used for all other experiments, we encountered problems such as linalg errors, invalid gaussians, and OOM errors on some test problems. We have thus decided to exclude SOBER, rather than reporting results that may not be valid due to technical limitations in the available code. We are open to add a statement on the exclusion due to these problems in the manuscript.
> > **Data Augmentation Procedure**
>
> The terms in BEEBO and [3] do not differ - they both refer to the information gain of a GP (and call it so). We avoided using the term fantasization, as it is often understood as fantasizing (and using) $y$ values at $x$, as done in e.g. the KB baseline. As also pointed out in [3], when using the entropy (and not the posterior mean), $y$ is not used, so no hallucinated $y$ values affect the result.
>
> As we introduce the concept already in Eq. 4, the generic form that marginalizes over $y$, we use the term augmentation, as in the that case it would not be called fantasization.
> We have edited Alg. 1 to use the term “fantasize” instead of "augment" to make clear that we use fantasization to compute $C_{aug}$ on the implementation level.

---

> > ### Comment · Reviewer_4oKm · 2024-08-09
> >
> > Thank you for your reply, and I acknowledge your effort. Since I had not initially reviewed the appendix, my concern about weak evaluation was misplaced. However, I believe the main paper should be self-contained to convey key points effectively. While it is great to include many experimental results, they should be accompanied by clear explanations of "why". While theory is the easiest to answer, experimental investigation beyond final convergence comparison is also crucial.
> >
> > I would like to clarify a few questions, as it seems the authors may not have fully read the related papers. Given the limited time, I understand that reviewing all related literature can be challenging. Therefore, I believe another round of conference review would be beneficial, allowing more time to refine the work. Thus, I still recommend rejection but have raised my score to 4 in recognition of the extensive experimental efforts.
> >
> > **MC Acquisition Function (AF):** To decide the next query point, the AF must be maximised, which involves an "inner" optimisation loop that is non-convex. The MC AF leverages sub-modularity and has proven to converge to the true global maximum of the AF in each round. The closed-form AF requires heuristics, such as multi-start local optimer like L-BFGS-B. In this context, I do not believe a closed-form expression is necessarily better than MC methods, as they offer different benefits.
> >
> > **Tight Control:** This serves as a counter-example that "UCB theory" can achieve tight control through $\beta$. BALLET is an example of a method that restricts exploration via confidence bounds. This is not about suggesting another baseline comparison; rather, it is about preferring simpler, well-established approaches. When a provably converging method easily comes up, why introduce a new heuristic?
> >
> > **Heteroscedastic BO:** While I do not have a specific reference, this idea is fairly intuitive and the paper should exist somewhere. It serves as another counter-example.
> >
> > Good luck with the revision!

---

> > > ### Author Response · Authors · 2024-08-09
> > >
> > > Thank you for taking the time to consider our rebuttal!  We appreciate the effort to work together to improve the clarity of our work.
> > >
> > > **MC AF optimization**:
> > > We agree that MC is a powerful paradigm and that different strategies provide different benefits. Optimizing MC AFs uses sample average approximation to optimize towards the true maximizer $\alpha^* $ . As you also mentioned in your comments, Appendix D.3 in the BoTorch paper provides more theoretical background to that approach. As one would expect for MC approximations, there is a critical dependency on the number of samples: The results of theorems 1 and 2 for approximating $\alpha^{*}$ both hold in the **limit $i \to \infty$, an infinite number of samples**.
> > > In practice, one will naturally face a situation with a rather limited number of samples, as also indicated in various BoTorch tutorials (e.g. 256 samples in the $q$-NEI tutorial), so the true AF is by definition *approximated*.
> > >
> > > As mentioned in D.3, this will therefore leave us with an **integration error**, that becomes more critical in large $q$ as the AF is of dimension $\mathbb{R}^{q}$. Our closed-form expression does by design not suffer from an integration error.
> > >
> > > Note that BoTorch also uses multi-start L-BFGS-B for MC AFs to mitigate the fact that the optimization approximates the true AF (BoTorch paper section 4.1).
> > >
> > > **Tight control**:
> > > Thank you for clarifying how the BALLET comment was intended. Again, our argument is not that UCB *theory* does not provide control (we are convinced it does), but that the practical limitations of BoTorch's MC methods at large $q$ outlined above do not deliver that control in practice, as we see empirically.
> > >
> > > **Heteroscedastic BO**:
> > > We would really appreciate a reference, as it is not intuitive to us. Our understanding of heteroskedastic noise and UCB is rooted in the analysis of Makarova et al. 2021, which deals with heteroskedastic GPs, and shows how UCB in fact needs to be modified to deliver risk-averse behaviour.

---

### Official Review · Reviewer_x4Ge · 2024-07-12

**Soundness:** 3
**Presentation:** 2
**Contribution:** 2
**Rating:** 5
**Confidence:** 4

**Summary:**

The paper introduces a new approach to batch Bayesian optimization that explicitly trades off between exploration and exploitation via energy and entropy terms. The method is able to efficiently generate large batches for evaluation that outperform comparable methods for Bayesian optimization.

**Strengths:**

The method is novel and well-motivated.

I found the analysis in Appendix B to be especially strong, in which the proposed method BEEBO is compared to other methods for batch Bayesian optimization. This analysis shows the originality of the method and gives strong context for it.

The paper is clearly written and easy to read. The appendix was especially useful and had many valuable parts.

The analysis of heteroskedastic noise was great to see and shows a useful and understudied setting where the method is especially valuable.

**Weaknesses:**

I think the method is interesting and will be of value to the field. However, I do not find that the experimental evaluation of the method provides full support for the claims of the paper.

**Issue 1: Evaluation limited to large-batch setting**

The major issue is that the paper claims the method is for general batch Bayesian optimization problems, without any qualifiers that I can see. The experiments all use q=100, which is a large batch size. Smaller batch sizes are often of interest too, e.g. batch sizes of 5, 10, and 50 in the GIBBON paper. The setting q=100 used here is also used in the TURBO paper (Eriksson et al. 2019) where it is described as a "large batch size."

Given the experiment results in the paper, I don't know if this method will perform well for small- or medium-sized batches. Thus, either the experiments need to be expanded to include experiments with batch sizes such as 5 and 10, or the framing of the paper needs to be adjusted to emphasize that the method is specifically for large-batch Bayesian optimization, not general batch BO problems.

This issue also relates to the choice of baselines. The experiments only explore large-batch settings, where GIBBON (as the paper notes) is known to perform poorly. If the paper wants to claim that it performs better than GIBBON in general, then it needs to make that comparison on batch sizes of 5 and 10. If the paper wants to claim superiority only on large-batch settings, then that's fine to only use q=100, but then it needs to compare to state-of-the-art for large batch. Thompson sampling is a popular method for large-batch settings which is included as a baseline, but to my knowledge the state-of-the-art for large-batch BO is TURBO (Eriksson et al. 2019). In fact the q=100 and the robot pushing and rover trajectory problems are all exactly as in the TURBO paper, so its inclusion as a baseline is pretty obvious and, I think, necessary.

**Issue 2: Lack of statistical significance**

The results of the experiments do not appear to be statistically significant. The main results given in the main text are tables, and these tables do not have confidence intervals. The only place where uncertainty in the results is show are the figures in the appendix, and there the confidence intervals appear to overlap in most cases. This is due to the use of only 5 replicates. I appreciate that these experiments are costly to run since they are using 1000 iterations, nevertheless the lack of statistical significance in most of the results provides weak support for the claim that BEEBO is actually better, vs. what we're seeing just being noise in the experiments. The paper needs some form of statistical analysis to convince the reader that what we're seeing is not just noise in the experiments. The best way to do this would be to include confidence intervals in tables 2 and 3, and then increase the number of replicates as necessary to achieve statistical significance in the differences in means. I do not feel it appropriate to highlight the method as being "best value" when it is possible that the confidence interval for that value contains the values of the other methods.

**Questions:**

The issues raised above can be addressed by running more experiments (batch sizes 5 and 10; TURBO; more replicates to get reasonable CIs for the results tables). But it will probably require more experiments that can be run in the rebuttal period. Do we expect the method to work well for Q=5 and Q=10?

---

> ### Author Rebuttal · Authors · 2024-08-06
>
> We thank the reviewer for their encouraging comments! Having a compelling experimental evaluation is of key importance to us, and we have performed the requested experiments to further demonstrate BEEBO's performance.
>
> > **Issue 1: Evaluation limited to large-batch setting**
>
> Thank you for the helpful suggestions! We have now ensured that
> - TurBO is also discussed in the related works section.
> - TurBO is included in the experiments.
> - Experiments on batch sizes 5 and 10 were added.
>
> Please see the PDF of the global response for the results of the additional experiments.
> Indeed, it was our focus to improve performance in the “large-batch” setting, as this is where we found $q$-UCB to underperform, and GIBBON to not be applicable. We have now clarified this in multiple places in the paper to be more precise.
>
> > **Issue 2: Lack of statistical significance**
>
>
> We appreciate that we need to be more rigorous with our reporting of results. We have now doubled the number of replicates and performed statistical testing on the differences in means. We can confirm that on aggregate over all experiments, BEEBO’s performance gain over the baselines is statistically significant. We have updated all results tables accordingly, and include the results of the statistical testing in the appendix. Please see the global response for combined p-values.
>
> (For the time being, performance of GIBBON will still be based on nine replicates on five Ackley test problems, as one replicate can take multiple days. This should not affect the findings, as we already achieved statistical significance, but we will update the manuscript once the last runs finish for consistency.)
>
>
> We agree that the best way to present results would be in principle to include the confidence intervals in Tables 2 and 3. However, we found that this makes the tables very dense and extremely hard to read.. As a compromise, we added the confidence intervals to the extended tables in the supplementary material (and the rebuttal PDF), and mention this in the captions of Tables 2 and 3.
>
>
> >The issues raised above can be addressed by running more experiments [..] Do we expect the method to work well for Q=5 and Q=10?
>
> We really appreciate that the reviewer is mindful of computational requirements - thankfully, we managed to run all critical experiments in the rebuttal period, as outlined above. We find that also at Q=5 and Q=10, BEEBO is competitive with TuRBO and GIBBON. Especially at Q=10, q-UCB shows strong performance. Overall, both BEEBO and q-UCB benefit from higher exploration rates at low Q.

---

> > ### Comment · Reviewer_x4Ge · 2024-08-13
> >
> > Thank you for the additional analyses. It is good to see that BEEBO continues to perform well in the small-to-medium batch setting, and that it performs well relative to TurBO. It is interesting that the sensitivity to T' appears to be higher for q=5 and q=10, a phenomenon that probably merits investigation.
> >
> > As progress was made on my two main concerns, I will raise my score. Although other reviews have raised good points that I hope the authors will consider further.

---

### Official Review · Reviewer_mXVT · 2024-07-13

**Soundness:** 2
**Presentation:** 3
**Contribution:** 2
**Rating:** 5
**Confidence:** 4

**Summary:**

This work introduces a batched acquisition function that balances exploration and exploitation by using a weighted sum of mutual information and expected value, with the weights defining the trade-off. The discussion links the proposed algorithm to UCB and asserts that it naturally addresses heteroskedastic noise.

**Strengths:**

1. The proposed acquisition function and its optimization and approximation methods are straightforward and practical.
2. The paper provides extensive empirical results to illustrate the proposed algorithm's efficiency.

**Weaknesses:**

1. The introduced parameter controlling the trade-off lacks interpretation as in previous methods.
2. The completeness of the related work discussion is concerning. This is potential because the summarization lacks high-level extraction of the design of the algorithm, and the focus of the paper is, to some extent, scattered。

**Questions:**

One concrete example of the second aforementioned weakness is the criticism of MC approximation in high-dimensional applications. Recent advancements in applying MCMC address this issue in a principled manner and might be of interest.

***Reference:***

Yi, Zeji, Yunyue Wei, Chu Xin Cheng, Kaibo He, and Yanan Sui. "Improving sample efficiency of high dimensional Bayesian optimization with MCMC." arXiv preprint arXiv:2401.02650 (2024).

**Limitations:**

Discussed above.

---

> ### Author Rebuttal · Authors · 2024-08-06
>
> Thank you for your comments! We have incorporated the feedback in the updated manuscript, and look forward to further discussion.
>
> > The introduced parameter controlling the trade-off lacks interpretation as in previous methods.
>
> We fully agree that an entropy quantity is in principle harder to intuitively interpret than a univariate variance, as used in (single-point) UCB. However, we note that when working in batch mode, the $q$-UCB parameter also becomes harder to interpret, as illustrated in Figure 1. In supplementary section B.1, we provide a derivation showing how BEEBO’s $T$ parameter can be set to match the interpretation of the (single-point) UCB parameter.
> In the experiments as well as in our available implementation, we always make use of this derivation, operating with the UCB-matched $T’$, rather than a raw $T$.
>
> We have reworked main text section 3.1 to more prominently explain the relation of $T$ to UCB.
>
>
>
> > The completeness of the related work discussion is concerning. This is potential because the summarization lacks high-level extraction of the design of the algorithm, and the focus of the paper is, to some extent, scattered。
>
> Thank you for pointing out the recent Yi et al. paper on MCMC for BO, which we have now included in the related works section. Could you provide further guidance as to what you would find missing from the related works section?
>
> Given the space constraints and the focus of our paper, our intent was to offer a concise summary of batch BO methodology, rather than delving further into MC methods. As suggested by another reviewer, we have now also added TurBO to the related works section as well as the experiments.
>
> The related works section now ends in
> >Eriksson et al. demonstrate that overexploration also can be problematic in higher dimensions, and alleviate this using local trust regions in TuRBO. Maintaining such regions with high precision discretization can be memory-expensive, as indicated by Yi et al., who suggest using MCMC-BO with adaptive local optimization to address this by transitioning a set of candidate points towards more promising positions.

---

> > ### Comment · Reviewer_mXVT · 2024-08-11
> >
> > I appreciate the author's responses to my concerns, especially regarding the interpretability and the related works. Concerning the other potentially related works, I believe my fellow reviewers have proposed sufficient concrete examples. My additional comments on the algorithm's possible advantage over existing theoretical sound algorithms align with the author's recent response. The applicability of the acquisition optimizer is a known problem. Therefore, it is still meaningful to introduce algorithms like BEEBO. I hope the author can address the remaining concerns of other reviewers. At the same time, I maintain my original evaluation.
> >
> > ***Reference:***
> > In section B by Shahriari et al. 2016: "Unfortunately, these auxiliary optimization techniques can be problematic for several reasons." "First, in practice, it is difficult to assess whether the auxiliary optimizer has found the global maximizer of the acquisition function."
> >
> > - Shahriari, Bobak, et al. "Taking the human out of the loop: A review of Bayesian optimization." Proceedings of the IEEE 104.1 (2015): 148-175.

---

### Author Rebuttal · Authors · 2024-08-06

We thank all reviewers for their helpful feedback! We are excited to hear that they find BEEBO
- *Helpful and practical* (mXVT)
- *Novel and with strong context* (x4Ge)
- *Effective with heteroskedastic noise* (CMG8, x4Ge)
- Has a *promising Statistical physics motivation* (4oKm, CMG8)

We have responded to each reviewer separately. In summary, we would like to highlight the following key improvements of the manuscript:
- **TuRBO was added** to the benchmark (x4GE) and the related works (mXVT). as an exemplary method of batched trust region BO methods such as also BALLET (4oKm)
- The **number of replicates** was **doubled** to demonstrate **statistical significance** (x4GE)
- Benchmark experiments at **batch sizes q=5 and q=10** were added (x4GE)
- Highlight the scaling of the T parameter for **UCB-like interpretability** more prominently (mXVT, CMG8)

As also pointed out by many reviewers, we consider it crucial that BO methods are benchmarked thoroughly against existing methods. This is why in our experimental section, we have now recorded more than **12,800 batch BO trajectories** in total, across 33 problems, multiple acquisition strategies, batch sizes and replicates. We believe that this is adequate empirical evidence of BEEBO's performance, which is complemented by Appendix B, which connects BEEBO to the theoretical background of existing work.

Please see the attached PDF for:
- an updated Table A1 including TuRBO, based on 10 replicates
- the q=5 benchmark
- the q=10 benchmark

As the PDF is limited to one page, we include the p-values for the meanBEEBO results in the table here. These were computed using a paired one-sided t-test for each test problem over the 10 replicates, followed by aggregation using Fisher's method.

In the updated manuscript, we include this table in extended form - showing the p-value of each t-test, and the aggregation, for meanBEEBO as well as maxBEEBO. Please just let us know if you wish to see individual p-values that below results are based upon, we will provide them as comments as needed (The full table would be 33x21 cells and poorly suited for the available markdown formatting).

|meanBEEBO |Method | Fisher p-value|
|-----|---------|-----------------|
|$T'$ = 0.05 | $q$-UCB |	2E-31 |
 |  | $q$-EI	|1E-13 |
 |  | TS |	2E-30 |
| | KB |	6E-06|
| | GIBBON |	8E-58 |
| | GIBBON (scaled) |	5E-47 |
| |TuRBO |	4E-29 |
| $T'$ = 0.5 | $q$-UCB |	6E-28
| |$q$-EI$ |	1E-18
||TS|	2E-34
||KB|	6E-10
||GIBBON|	1E-78
||GIBBON (scaled)|	2E-58
||TuRBO|	1E-44
|$T'$=5.0| $q$-UCB |	1E-28
|| $q$-EI |	1E-03
||TS|	4E-20
||KB|	2E-06
||GIBBON|	9E-56
||GIBBON (scaled) |	8E-36
||TuRBO|	2E-46

As for the $T$ parameter scaling, section 3.1 "BEEBO with Gaussian processes" in the updated manuscript ends in

>Using the kernel's learned amplitude $A$, we can relate BEEBO's $T$ parameter to the $\kappa$ of UCB. This allows us to configure BEEBO using a scaled temperature $T'$ that ensures both methods have equal gradients at iso-surfaces, enabling the user to follow existing guidance and intuition from UCB to control the trade-off. A derivation is provided in section B.1

We believe this will make it easier for the reader to find the relevant derivation, as opposed to the previous version where this was only mentioned in the experimental section.

We are looking forward to further discussion.

---

### Decision · Program_Chairs · 2024-09-25

**Decision:**

Accept (poster)

**Comment:**

This work introduces a new acquisition function inspired by statistical physics for batch Bayesian optimisation with GPs. The work includes a theoretical analysis and an extensive set of experiments to support the claims. While reviewers raised concerns regarding missing baselines and other experimental aspects, all were addressed during the rebuttal. Overall, this work is well executed and of interest to the community.